# Three-dimensional chromatin reorganization regulates B cell development during ageing

Fei Ma [1,6], Yaqiang Cao [2,6], Hansen Du[1], Fatima Zohra Braikia [1], Le Zong[3], Noah Ollikainen[1], Marc Bayer[4], Xiang Qiu [1], Bongsoo Park[3], Roshni Roy[1], Satabdi Nandi[1], Dimitra Sarantopoulou [1], Andrew Ziman[5], Aisha Haley Bianchi[1], Isabel Beerman [3], Keji Zhao [2], Rudolf Grosschedl [4] & Ranjan Sen [1]✉

The contribution of three-dimensional genome organization to physiological ageing is not well known. Here we show that large-scale chromatin reorganization distinguishes young and old bone marrow progenitor (pro-) B cells. These changes result in increased interactions at the compartment level and reduced interactions within topologically associated domains (TADs). The gene encoding Ebf1, a key B cell regulator, switches from compartment A to B with age. Genetically reducing Ebf1 recapitulates some features of old pro-B cells. TADs that are most reduced with age contain genes important for B cell development, including the immunoglobulin heavy chain (*Igh*) locus. Weaker intra-TAD interactions at *Igh* correlate with altered variable (V), diversity (D) and joining (J) gene recombination. Our observations implicate three-dimensional chromatin reorganization as a major driver of pro-B cell phenotypes that impair B lymphopoiesis with age.

Structural organization of the genome provides a platform for optimal gene expression[1]. Chromosome conformation capture techniques have revealed several levels of organization such as multi-megabase (Mb)-sized compartments[2], Mb-sized topologically associated domains (TADs)[3,4] and kilobase-sized chromosomal loops[5]. Structural changes at each level alters gene expression associated with cellular differentiation or disease[6]. For example, compartment changes have been noted during embryonic stem (ES) cell differentiation in vitro[7,8] and at specific stages of B lymphocyte differentiation in vivo[9]. TADs are largely invariant across cell types; however, subdomains within TADs vary between cell types and are associated with cell-specific gene expression[5,10–12]. Additionally, chromosomal translocations that disrupt TAD structure have been proposed to contribute to altered gene expression in cancer[13].

By contrast, very little is known about chromatin reorganization during organismal ageing, especially in the immune system. Most studies utilize senescent cells generated in culture as surrogates for physiological ageing[14]. Senescent cells show substantial changes in chromatin structure, including de-repression of heterochromatic compartment B, regional changes from compartment B to A facilitated by loss of facultative heterochromatin, and altered chromosomal looping that activates inflammatory gene expression[15–22]. The extent to which chromatin features of senescent cells reflects organismal ageing remains unclear, especially because these features vary depending upon the mode of senescence induction[18]. In this Article, we studied age-associated chromatin structure changes at a key checkpoint during B lymphocyte development in the bone marrow.

[1]Laboratory of Molecular Biology and Immunology, National Institute on Aging, Baltimore, MD, USA. [2]Laboratory of Epigenome Biology, Systems Biology Center, National Heart, Lung and Blood Institute, Bethesda, MD, USA. [3]Epigenetics and Stem Cell Init, Translational Gerontology Branch, National Institute on Aging, Baltimore, MD, USA. [4]Max Planck Institute of Immunobiology and Epigenetics, Freiburg, Germany. [5]Nikon Instruments Inc., Melville, NY, USA. [6]These authors contributed equally: Fei Ma, Yaqiang Cao. ✉e-mail: senranja@grc.nia.nih.gov

Dysregulated immune function during ageing underlies susceptibility to infectious diseases, autoimmunity and cancers[23–25]. Aged individuals also respond poorly to immunization, thereby precluding them from the most effective intervention against microbial pathogens[23,26–28]. B and T lymphocytes, which are central for responding to infection or vaccination, are reduced with age[23]. For B cells, this is reflected in reduced numbers of pro-B and pre-B cells, which are developmental stages soon after commitment of multipotential precursors to the B lineage[23,29,30]. While ageing affects expression of several transcription factors, mechanisms that attenuate pro-B cell development in old mice have not been identified[31–34]. To reveal cell-intrinsic mechanisms of chromatin organization that account for changes in pro-B cells with age, we assayed chromosome conformation by Hi-C in pro-B cells obtained from young (8–12 weeks) and old (100–110 weeks) recombination activating gene 2 (*Rag2*)-deficient mice (Supplementary Table 1)[35,36]. Absence of variable (V), diversity (D) and joining (J) gene recombination in this strain (or mice lacking Rag1) blocks development at the pro-B cell stage and ensures a homogeneous cell population for epigenetic studies.

## Results

### Compartment changes in old pro-B cells

Pro-B cell numbers in old *Rag2*-deficient mice were approximately twofold lower than in young mice (Extended Data Fig. 1a–c), comparable with previously noted reductions in normal C57Bl/6J mice[37]. Hi-C analysis revealed that compartment level interactions (≥10 Mb) were increased in old pro-B cells whereas those associated with TADs and loops (<10 Mb) were reduced in old pro-B cells (Fig. 1a,b). More than a hundred genomic loci transitioned from euchromatic compartment A to heterochromatic compartment B and vice versa in old pro-B cells (Mahalanobis distance (MD) *P* < 0.01) (Fig. 1c). To probe epigenetic states that accompanied compartment changes, we carried out chromatin immunprecipitation followed by sequencing (ChIP–seq) with anti-H3K27ac and anti-H3K27me3 antibodies. H3K27ac levels were reduced in regions that switched from A to B and increased in those that changed from B to A; however, H3K27me3 levels were unaltered (Fig. 1d).

RNA-seq analysis using young and old pro-B cells showed that genes located within regions that switched from compartment B to A with age were expressed, on average, at higher levels (Fig. 1e). One such gene is *Ncam1* (Extended Data Fig. 1d). Only 6 out of 310 genes annotated to compartments that changed from A to B with age were downregulated in old pro-B cells (fold change >2, *P* < 0.01; Supplementary Table 2), accounting for the absence of a statistically significant difference in average gene expression for A to B compartment changes in old pro-B cells (Fig. 1e).

### Compartmental changes impact *Ebf1*

The most significant A to B transitioning genomic region contained the *Ebf1* gene that encodes a transcription factor that is required for commitment of hematopoietic precursors to B lineage differentiation[38]. *Ebf1* expression was reduced in old pro-B cells (Fig. 1f) accompanied by lower levels of H3K27ac, but unchanged H3K27me3, at the *Ebf1* locus (Fig. 1f). We reasoned that age-associated defects in *Ebf1* expression may underlie some aspects of impaired B cell development in aged mice. Hi-C contact maps of this region in young pro-B cells revealed multiple interactions of the *Ebf1* promoter with intragenic and distal enhancers marked with H3K27ac modifications (Fig. 1g). These interactions were markedly reduced in old pro-B cells and replaced by interactions between intronic and promoter-distal CTCF-bound sites (Fig. 1g). We surmise the latter interactions reflect an 'off' state for *Ebf1* expression. *Ebf1* undergoes radial re-positioning from the nuclear periphery to a more central location concomitant with B cell commitment at the pro-B cell stage[9]. We found that the *Ebf1* locus moved closer to the nuclear periphery in old pro-B cells (Fig. 1h,i). In contrast, the *Foxo1* locus, encoding a co-regulated factor, did not change its radial position or proximity to heterochromatin-associated γ-satellite repeats in old pro-B cells (Fig. 1h and Extended Data Fig. 1e). *Foxo1* compartmentalization was also unaffected by age (Extended Data Fig. 1f). Our observations reveal large-scale conformational changes in pro-B cell genomes during ageing, some of which directly impact key transcriptional regulators of B cell development.

To evaluate the contribution of Ebf1 downregulation to age-associated chromatin and gene expression changes, we first carried out Hi-C and transcriptomic analyses using *Ebf1* heterozygote *Rag2*-deficient pro-B cells (*Ebf1*[+/−]). In these cells Ebf1 expression is reduced but not eliminated (Extended Data Fig. 2a)[39,40], reflecting the state of old pro-B cells. Hi-C analysis revealed changes at compartment (≥10 Mb), TAD and loop (<10 Mb) levels (Fig. 2a). *Ebf1*[+/−] pro-B cells shared some structural features with old pro-B cells, such as reduced intra-TAD interactions, but not others such as accentuated compartments (Fig. 1a and Fig. 2a,b). We infer that, Ebf1 does not lie at the apex of chromatin reorganization in old pro-B cells. Accordingly, *Ebf1* heterozygosity only recapitulated some differences noted between young and old pro-B cells. For example, the *Mmrn1* gene switched from compartment A to B and its expression was reduced in old pro-B cells as well as in *Ebf1*[+/−] pro-B cells (Fig. 2c), whereas at the *Minar1*/*Temed3* locus a new loop coincided with increased *Temed3* RNA in old and *Ebf1*[+/−] pro-B cells (Fig. 2d). Contributions of such loci to the properties of old pro-B cells remain to be determined. Deleting one allele of *Pax5*, a B cell regulator implicated in altering chromatin structure[41], in *Ebf1*[+/−] pro-B cells did not further alter the chromatin state (Extended Data Fig. 2b).

Transcriptomic analyses of *Ebf1*[+/−] pro-B cells also revealed similarities to gene expression changes in old pro-B cells (Fig. 2e). We also found that over 50% of previously identified Ebf1 target genes were altered in old pro-B cells (Supplementary Table 3)[42], and several genes associated with B cell development were downregulated in both *Ebf1*[+/−] cells as well as old pro-B cells. (Fig. 2f). To evaluate the contribution of Ebf1 to gene expression changes in old pro-B cells, we re-expressed

**Fig. 1 | Age-associated chromatin compartment changes in murine bone marrow B cell progenitors. a**, Combined Hi-C contact linear genomic distances density plots from two replicates of young and old pro-B cells. Colour shading refers to compartments (pink), TADs and loops (green), and close interactions (yellow). **b**, Example of Hi-C contact heatmaps, visualized with Juicebox using Coverage (sqrt) normalization. Difference heatmap (right) shows regions with increased (red) and decreased interactions (blue) in old pro-B cells. **c**, PC1 of young and old pro-B cells from Hi-C compartment analysis. Numbers of significantly changed euchromatic (A) and heterochromatic (B) bins of 100 kb are indicated in each quadrant. Significantly changed bins were obtained using a two-pass MD method with a one-sided chi-squared test *P* < 0.01. **d**, Difference of aggregate H3K27ac and H3K27me3 (*n* = 2) signals within significantly switched bins identified in **c**. **e**, Differential expression (*n* = 4) for genes associated with each category identified in **c** from RNA-seq analysis. The top and bottom of each box (in **d** and **e**) represents the 75th and 25th percentile, respectively, with a line at the median. Whiskers extend by 1.5× the interquartile range. *P* values were determined by two-sided Wilcoxon signed-rank test. **f**, Chromatin state of *Ebf1* locus in young and old pro-B cells. Top tracks show Hi-C compartment PC1, with orange indicating compartment A and blue indicating compartment B. Lower tracks show normalized ChIP–seq and RNA-seq tracks. *Clint1* (right) serves as a control gene located in compartment A that is unaffected by age. **g.** Hi-C heatmaps of the *Ebf1* locus. Interactions with greater than twofold change in old pro-B cells are marked by black circles/arcs (reduced) or the blue square/arc (increased). Numbers represent the ratio of Hi-C PETs in old pro-B cells compared with young pro-B cells. **h**, Nuclear positioning assayed by FISH. Representative nuclei from young and old pro-B cells and non-B cells from *Rag2*[−/−] mice are shown. Probe colours are as indicated. Dashed lines delineate the nuclear periphery. **i**, Distance between *Ebf1* locus and nuclear periphery in pro-B cells and non-B cells (*n* = 100) is shown. The red line indicates the median value; *P* values based on unpaired two-sided *t*-tests.

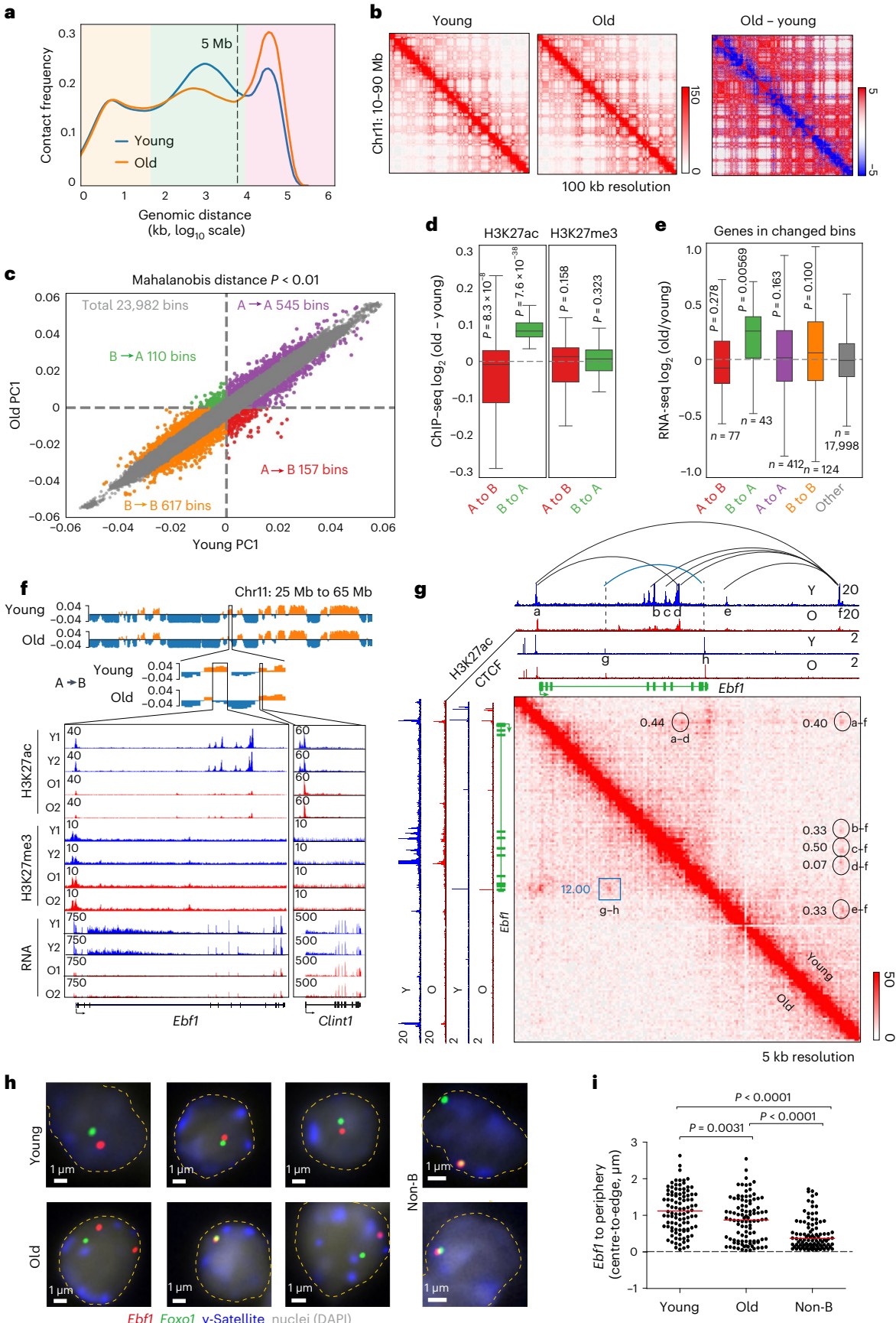

Ebf1 in young and old pro-B cells from *Rag2*[−/−] mice (Extended Data Fig. 2c,d). We found that ex vivo culture of primary pro-B cells in interleukin 7 (IL-7) required for retroviral transduction obliterated most of the differences between freshly isolated young and old pro-B cells (Extended Data Fig. 2e). However, of the 68 age-associated genes that remained differentially expressed, Ebf1 transduction restored expression of 50 (Extended Data Fig. 2f). Taken together, we conclude that reduced expression of Ebf1 associated with relocation of the locus from compartment A to B with age, induces chromatin and gene expression changes that negatively impact B cell development during ageing.

## TAD alterations in old pro-B cells

We identified 2,781 TADs in young pro-B cells. By quantifying the proportion of pair-end tags (PETs) that fell within or outside TAD boundaries as a measure of TAD 'strength' (enrichment score; ES), we found 240 TADs with significantly lower ES (young specific) and 84 TADs with significantly higher ES (old specific) in old pro-B cells (MD *P* < 0.01) (Fig. 3a and Supplementary Table 4). The TAD containing the immunoglobulin heavy chain (*Igh*) gene locus was most reduced in old pro-B cells (Fig. 3a and Extended Data Fig. 3a). Other important B cell regulatory genes, such as *Ebf1* and *Pax5*, were also located within reduced TADs (Fig. 3a,b and Extended Data Fig. 3a,b). The 240 young-specific TADs were associated with reduced H3K27ac within the TAD body (Fig. 3c) and less H3K27me3 at TAD boundaries in old pro-B cells (Fig. 3d). CTCF binding was unchanged; however, Rad21 levels were lower throughout bodies of young-specific TADs in old pro-B cells (Fig. 3e,f). Global reduction of Rad21 recruitment to chromatin in old pro-B cells (Fig. 3g), correlated with increased expression of Wapl, the cohesin off-loader (Fig. 3h and Extended Data Fig. 3c). Increased Wapl expression in old pro-B cells was independent of Ebf1 dysregulation since no changes were evident in *Ebf1*[+/−] pro-B cells (Extended Data Fig. 3d). Several genes associated with B cell development, including *Ebf1*, *Pax5*, *Bach2* and *Bcl11a*, that were located within reduced TADs were expressed at lower levels (Fig. 3h and Extended Data Fig. 3c), although average gene expression was unchanged (Fig. 3i).

We tested the effects of changing Wapl levels in a recombinase deficient pro-B cell line (Extended Data Fig. 4a,b). Twofold higher *Wapl* expression resulted in lower genomic contacts in the 50–300-kb range, typical of intra-TAD interactions (Extended Data Fig. 4c). A total of 341 TADs were significantly reduced, including the TAD encompassing the *Igh* locus (Extended Data Fig. 4d,e). RNA-seq from control and Wapl-overexpressing cells showed that differential gene expression patterns were similar to those seen by reducing Ebf1 levels or in old pro-B cells (Extended Data Fig. 4f). We conclude that altered levels of Wapl induce TAD changes during ageing that impair multiple aspects of pro-B cell differentiation and function.

Reduced levels of H3K27ac within young-specific TADs prompted us to probe H3K27ac dynamics more deeply during ageing. We observed reduced H3K27ac (young selective) at over 10,000 sites genome wide in old pro-B cells. The majority (78%) were annotated as enhancers (Fig. 4a). Conversely, approximately 1,200 sites gained H3K27ac in old pro-B cells (Fig. 4a). Total H3K27ac levels were comparable in young and old pro-B cells (Extended Data Fig. 5a). Young- or old-selective H3K27ac peaks mapped to largely unique sets of genes (Fig. 4b) with opposite expression patterns (Fig. 4c). Examples of genes with young- and old-selective H3K27ac peaks are shown in Fig. 4d. p300, the enzyme that acetylates H3K27ac, had altered recruitment genome-wide in old pro-B cells (Extended Data Fig. 5b), with majority of sites that lost p300 coinciding with sites of reduced H3K27ac in old pro-B cells (Extended Data Fig. 5c). By contrast, recruitment of the chromatin remodeller, Brg1, was very similar in young and old pro-B cells (Extended Data Fig. 5d).

We carried out anti-H3K27ac HiChIP in young and old bone marrow pro-B cells to probe chromatin looping. Interactions involving young- or old-selective H3K27ac sites were on average reduced or increased, respectively, in old pro-B cells compared with interactions of H3K27ac sites that did not change with age (Fig. 4e). This trend was maintained after segregating enhancer- or promoter-associated differential H3K27ac sites (Fig. 4f,g). Enhancer–promoter interactions in several genes critical for B cell development were affected with age (Fig. 4h and Extended Data Fig. 5e).

## Reduced *Igh* TAD strength

The most reduced TAD in old pro-B cells encompassed the 3 Mb *Igh* locus (Fig. 5a,b). Two prominent age-associated differences were obvious at the *Igh* locus. First, a 'stripe' that extends throughout the TAD was reduced in old pro-B cells (Fig. 5c). Location of this stripe coincides with a cluster of CTCF-bound sites (3′ CTCF-binding elements; 3′ CBEs in Fig. 5d) that have been proposed to mark the 3′ end of the *Igh* locus[43,44]. Second, sub-TADs within the *Igh* TAD were reduced in old pro-B cells (Fig. 5c). We have previously proposed that such sub-TADs represent one level of compaction of the part of the locus that encodes variable ($V_H$) gene segments and facilitates VDJ recombination of distally located gene segments[45,46]. H3K27ac and Rad21 levels were lower throughout the 2.5 Mb $V_H$ region (Fig. 5d,e) in old pro-B cells, as was the prominent H3K27ac peak corresponding to the Eμ enhancer (Fig. 5d). By contrast, CTCF binding through the $V_H$ region, at intergenic control region 1 (IGCR1)[47] and at the 3′ CBEs[43] was minimally affected by age (Fig. 5d,e). We also found that a patch of H3K27me3 corresponding to the 3′ $V_H$ gene segments was lower in old pro-B cells (Fig. 5d,e). Fluorescence in situ hybridization (FISH) analysis provided independent evidence for reduced sub-TAD formation in old pro-B cells (Fig. 5f,g). We conclude that widespread loss of H3K27ac and Rad21 and reduced interactions mark the *Igh* TAD in old pro-B cells.

To more deeply probe the interactions within the *Igh* TAD, we carried out capture Hi-C using young and old pro-B cells (Supplementary Table 5). Consistent with whole-genome Hi-C, we found that intra-TAD interactions, as well as interaction of the 3′ *Igh* domain

**Fig. 2 | Reduced Ebf1 expression in pro-B cells partially mimics ageing phenotype. a**, Combined Hi-C contact linear genomic distances density plots from two replicates of *Rag2*-deficient *Ebf1*[+/+] and *Ebf1*[+/−] pro-B cells. Genomic distances are shaded as follows: compartments (pink) TADs and loops (green) and close interactions (yellow). **b**, Similar TAD changes in old versus young and *Ebf1*[+/−] versus *Ebf1*[+/+] pro-B cells (chromosomal region as indicated). Heatmaps in each kind of pro-B cells are shown with difference maps on the right. Blue and red colours represent reduced or increased interactions, respectively, in old or *Ebf1*[+/−] pro-B cells. **c**, Similar compartment switching in old versus young and *Ebf1*[+/−] versus *Ebf1*[+/+] pro-B cells at *Mmrn1* gene locus. Top tracks show Hi-C PC1, with orange indicating compartment A and blue indicating compartment B. Lower tracks show normalized RNA-seq tracks across the *Mmrn1* locus. **d**, Similar loop changes in old versus young and *Ebf1*[+/−] versus *Ebf1*[+/+] pro-B cells. Hi-C heatmaps for a region of chromosome 9 with increased interactions (blue arc) in old and *Ebf1*[+/−] pro-B cells. Numbers adjacent to the blue square represent the ratios of Hi-C PETs in old compared with young (top) pro-B cells and in *Ebf1*[+/−] compared with *Ebf1*[+/+] pro-B cells (bottom). RNA-seq analysis in young/old and *Ebf1*[+/+]/*Ebf1*[+/−] pro-B cells are shown alongside. **e**, CellRadar (https://karlssong.github.io/cellradar/) plot derived from transcriptional changes identified in comparison of young and old (*Rag2*[−/−]) pro-B cells (*n* = 4) and *Ebf1*[+/+] and *Ebf1*[+/−] pro-B cells (*n* = 2). DEGs, differentially expressed genes; FC, fold change; NK, nature killer cells; ProE, proerythroblast; CFUE, colony forming unit-erythroid; Pre CFUE, colony forming unit-erythroid restricted precursor; MkP, megakaryocyte progenitor; MkE, megakaryocyte-erythroid; ETP, early T cell precursor; CLP, common lymphoid progenitor; GMP, granulocyte-monocyte progenitor, pre GM, pre-granulocyte macrophage; LMPP, lympho-myeloid primed progenitor; ST-HSC, short-term hematopoietic stem cell; LT-HSC, long-term hematopoietic stem cell. **f**, Expression levels of key B cell genes in pro-B cells of indicated genotypes and young and old (*Rag2*[−/−]) pro-B cells measured by RNA-seq. Colours indicate *z*-score normalized gene expression levels.

with the rest of the locus, were reduced in old pro-B cells (Fig. 6a). Virtual 4C representation of the Hi-C data showed that interactions of distal regions with the 3′ end of the locus were most prominently reduced in old pro-B (Fig. 6b). To investigate the impact of Ebf1 downregulation on *Igh* structure, we carried out Hi-C analysis of *Ebf1*[+/−] pro-B cells (Fig. 6c). Virtual 4C showed 3′ CBEs contact till the middle of the $V_H$ region in *Ebf1*[+/+], probably due to their proliferative phenotype[48]. These contacts were further reduced in *Ebf1*[+/−] cells whereas interactions with Eμ and IGCR1 were unaffected (Fig. 6c,d).

We conclude that interactions between the 5′ and 3′ ends of the *Igh* locus are reduced in old pro-B cells, in part mediated by reduced Ebf1 expression with age.

To identify functional consequences of altered chromatin interactions with age, we quantified $V_H$ rearrangements in purified pro-B cells from young and old C57BL/6J mice[49,50]. We scored VDJ and DJ junctions in two replicate experiments, each carried out with pro-B cells pooled from four to six mice. We found that the most proximal $V_H$ gene segments were over-represented and the most distal ones were

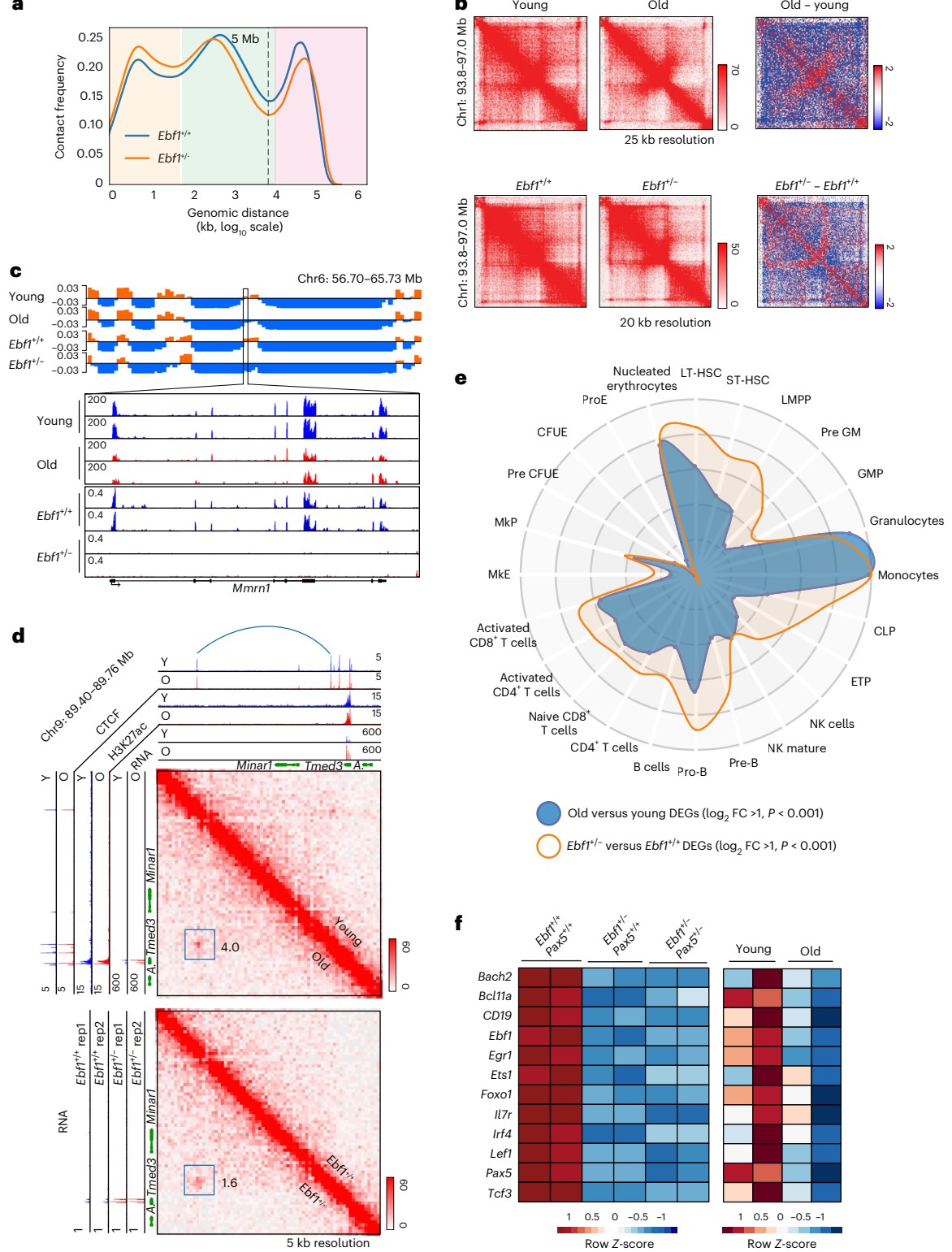

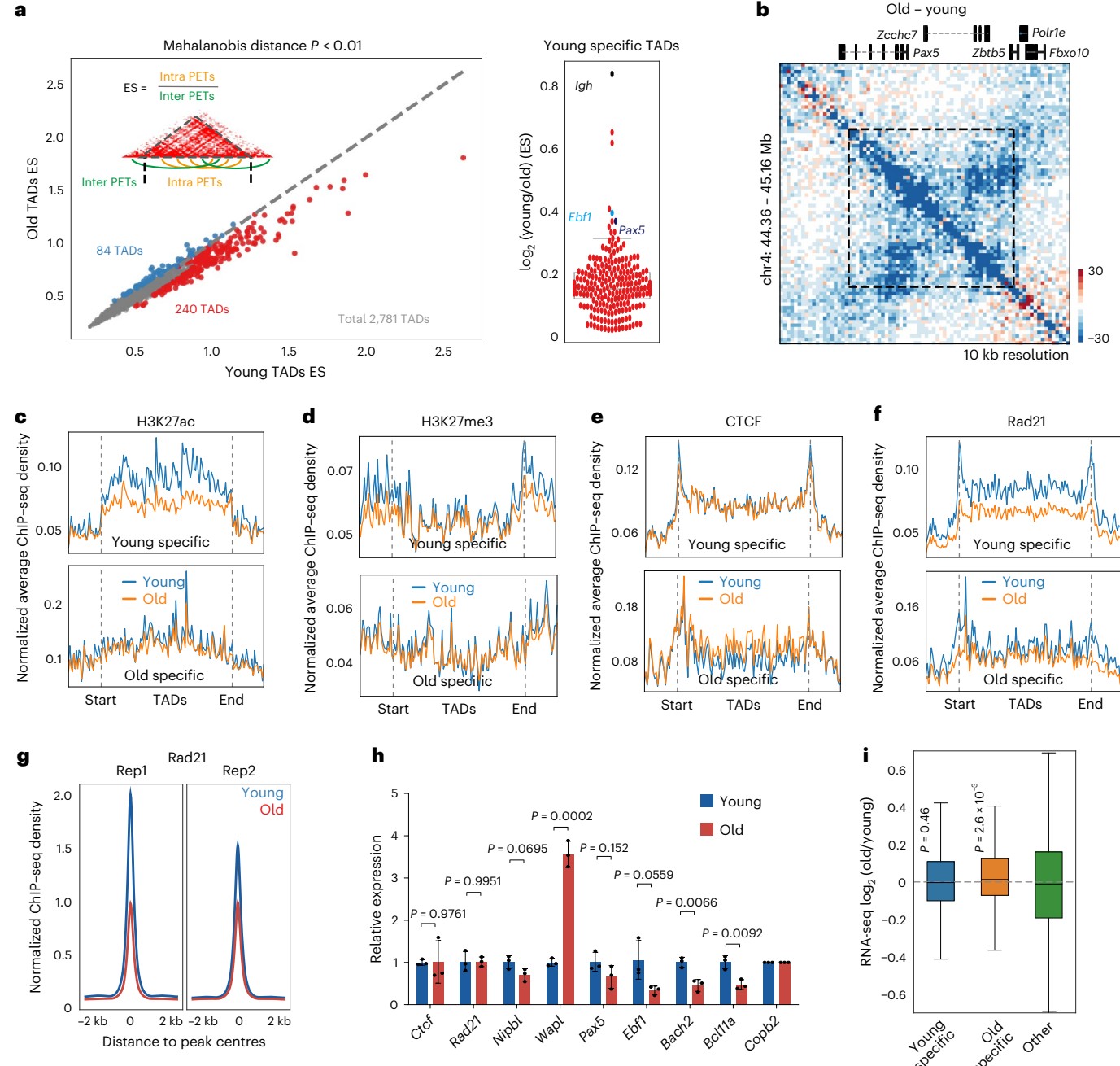

**Fig. 3 | Age-associated changes in TADs in pro-B cells. a**, The scatter plot on the left displays a quantitative comparison of TAD strength derived from Hi-C data of young and old *Rag2⁻/⁻* pro-B cells. Each dot in the plot represents one TAD, and TAD strength is represented as the ES, calculated as indicated. TADs showing increased or decreased interactions with age are coloured blue (old specific) or red (young specific), respectively. Dot plot represents reduced TADs in old pro-B cells. TADs encompassing *Igh*, *Ebf1* and *Pax5* loci are indicated. Significantly changed TADs were obtained using a two-pass MD method with a one-sided chi-squared test, $P < 0.01$. **b**, Difference heatmap showing reduced TAD formation covering the *Pax5* gene locus in aged pro-B cells. **c**–**f**, Aggregation analysis of H3K27ac (**c**), H3K27me3 (**d**), CTCF (**e**) and Rad21 (**f**), ChIP–seq signal densities at

significantly decreased ($n = 240$, young-specific group) and increased ($n = 84$, old-specific group) TADs identified in **a**. **g**, Aggregation analysis of Rad21 ChIP–seq signal densities around peak centre. Rep1 and Rep2 represent two biological experiments. **h**, Quantitative RT–PCR validation of RNA-seq data ($n = 3$). Data are normalized to *Copb2* and presented as mean ± standard error of the mean, with each replicate shown as a dot. Unpaired two-sided *t*-test was used to determine *P* values. **i**, Distribution of expression levels for the genes located in significantly changed TADs ($n = 4$). The top and bottom of each box represents 75th and 25th percentile, respectively, with a line at the median. Whiskers extend by 1.5× the interquartile range. *P* values were determined by two-sided Wilcoxon signed-rank test.

under-represented in old pro-B cells (Fig. 6e,f). $D_H$ gene segment utilization was minimally affected in old pro-B cells (Extended Data Fig. 6a). The proportion of VDJ compared to DJ recombined alleles (Fig. 6g) and levels of productive VDJ junctions were reduced in old pro-B cells (Fig. 6h). Though total number of CDR3 regions were reduced

in old pro-B cells, CDR3 length distribution was not altered with age (Extended Data Fig. 6b,c). We propose that the predominant effect of age on distal $V_H$ gene segment usage and recombination frequency results from reduced interactions between 5′ and 3′ ends of the *Igh* locus in old pro-B cells. Because ageing does not affect chromatin topology

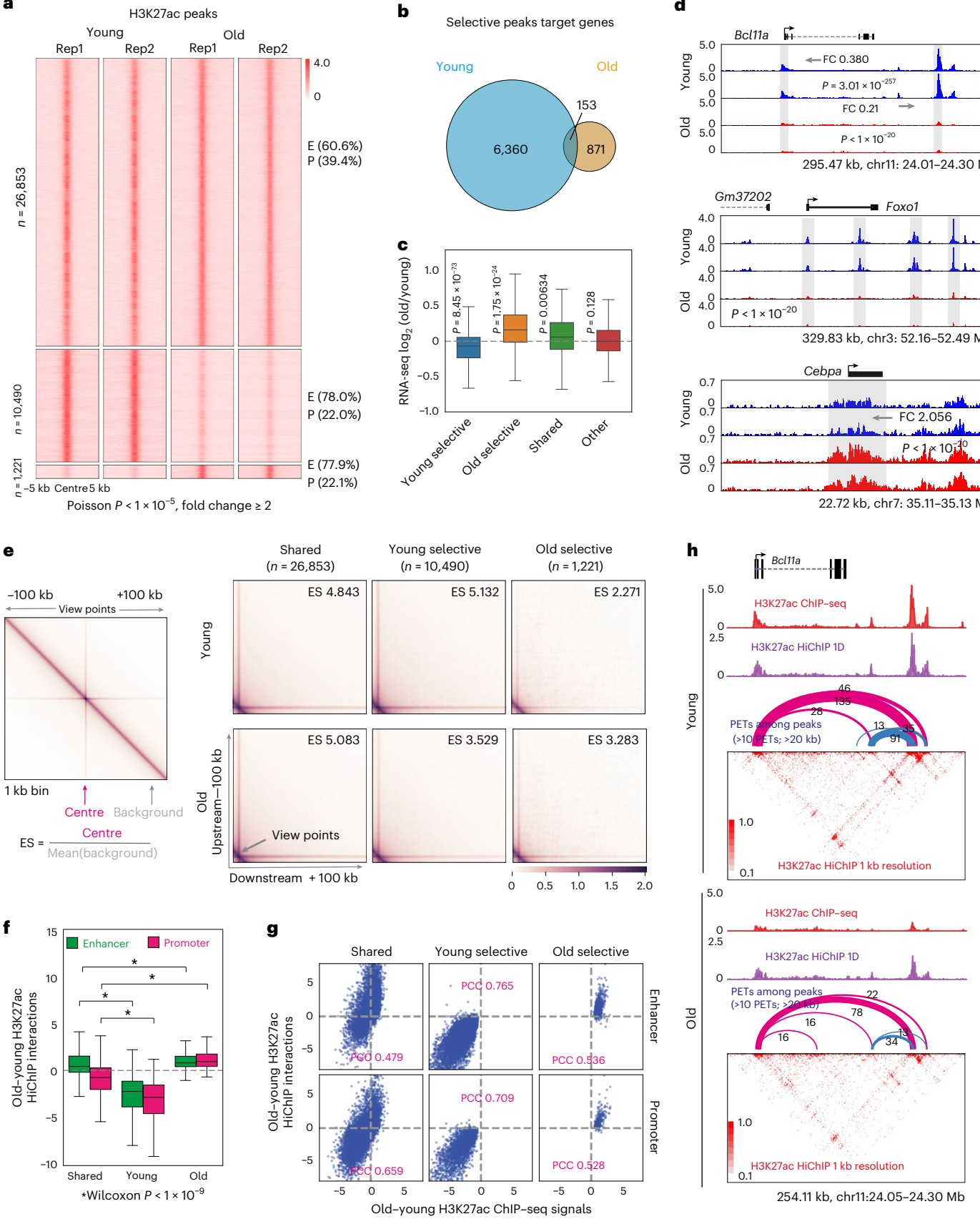

**Fig. 4 | H3K27ac-mediated chromatin reorganization during ageing.**
**a**, Changes in H3K27ac were identified from ChIP–seq in young and old pro-B cells (*n* = 2). Rep1 and Rep2 represent two biological experiments. H3K27ac peaks within 2 kb of the TSS are labelled as promoters (P), others are labelled as enhancers (E). Top, unchanged peaks (shared); middle, peaks that were reduced in old pro-B cells (young selective); and bottom, peaks that were increased in old pro-B cells (old selective). **b**, Age-associated H3K27ac peaks were annotated to genes; Venn diagram shows overlap between genes containing young- or old-selective H3K27ac peaks. **c**, Expression levels (*n* = 4) of genes containing each H3K27ac peak category. The 'Other' category includes genes that are expressed but not annotated to identified H3K27ac peaks. **d**, Genome browser tracks of genes containing young (*Bcl11a* and *Foxo1*) and old-selective (*Cebpa*) H3K27ac peaks. FC, fold change of indicated H3K27ac peak. *P* values were determined by one-sided Poisson test. **e**, Aggregation analysis of H3K27ac HiChIP data

(*n* = 2) with H3K27ac peaks identified in **a**. The ES for each H3K27ac peak was calculated as shown. The right panels display the average ES for each category of H3K27ac peaks. **f**, H3K27ac HiChIP interaction (*n* = 2) differences for categories identified in part 'a' separated by location of H3K27ac at promoter or enhancer. **g**, Correlation between changes in H3K27ac ChIP–seq signal and H3K27ac HiChIP interaction strength for peaks identified in **a**. PCC, Pearson correlation coefficient. **h**, Heatmaps of H3K27ac HiChIP interactions in the *Bcl11a* locus in young (upper) and old (lower) pro-B cells. The pink arches represent interactions from the *Bcl11a* promoter. Strength of interactions are denoted by thickness of arches. Top tracks show H3K27ac peaks obtained from ChIP–seq (red) or HiChIP (purple). The top and bottom of each box (in **c** and **f**) represent 75th and 25th percentile, respectively, with a line at the median. Whiskers extend by 1.5× the interquartile range. *P* values were determined using two-sided Wilcoxon signed-rank test; asterisks signify $P < 1 \times 10^{-9}$ (Methods).

of the 3′ *Igh* domain, within which $D_H$ rearrangements occur, $D_H$ usage and frequency are less affected by age.

## Discussion

Our studies reveal large-scale age-associated chromatin reorganization at a critical juncture of B cell development. These include increased interactions at the compartment level, transitions between compartments and substantially reduced TAD interactions that are accompanied by altered histone modifications and gene expression. The trends we observed during physiological ageing were quantitatively similar to chromatin changes noted for other biological processes, such as ES cell differentiation in vitro[7]. However, each chromatin feature identified here was shared with only a subset of studies of cellular senescence[18] and recent investigations into age-associated chromatin reorganization[51–54]. Our observations highlight the unique chromatin state of old pro-B cells, mechanisms by which it is generated and its functional consequences.

Chromatin structure changes in aged pro-B cells identified here were more extensive than those previously reported for pre-B cells[37]. One possibility is that only a small proportion of pro-B cells in old mice that lack the perturbations reported here, differentiate into pre-B cells. These pre-B cells may therefore reflect the epigenetic state of the 'young-like' precursor pro-B cells. Alternatively, pre-B cells that develop in old mice may acquire a more youthful chromatin state during the transition. Our observations identify the pro-B cell stage as a selective target of epigenetic alterations that negatively impact B cell development during ageing.

Changes stemming from *Ebf1* re-compartmentalization, radial positioning and downregulation of expression impact two key features of old pro-B cells. First, expression of B lineage affirming transcription factors (E2A, Ebf1, Pax5 and Bcl11a) and suppressors of other lineages (Bach2) is reduced, resulting in a gene expression profile that skews towards myeloid and erythroid differentiation[55–59]. This may explain, in part, myeloid skewing of hematopoietic stem cells during ageing[60] and for the previously noted retardation of pro- to pre-B cell differentiation in young *Ebf1*[+/−] mice[39]. Our observations with genetically reduced

Ebf1 expression suggest that chromatin structural changes drive *Ebf1* dysregulation and its downstream consequences during ageing.

Second, immunoglobulin gene assembly and expression of IgH protein in pro-B cells is a key checkpoint during B cell development[55,61]. Pro-B cells that fail to express IgH do not differentiate further and are lost by apoptosis. We found that changes in *Igh* locus structure in old pro-B cells negatively impact this checkpoint by reducing diversity and efficiency of variable ($V_H$) gene segment recombination. Several of these features were recapitulated in *Ebf1*[+/−] pro-B cells, indicating that they resulted from reduced Ebf1 expression in old pro-B cells.

Global weakening of TADs in old pro-B cells was accompanied by reduced genome-wide recruitment of Rad21 and loss of promoter/enhancer interactions. Reduced TADs and increased compartments, as observed in old pro-B cells, were previously noted in cell lines depleted of Rad21 or CTCF[62,63]. Although both these genes were comparably expressed in young and old pro-B cells, Rad21-binding levels were lower in old pro-B cells and may contribute to loss of TAD strength. We also found increased expression of Wapl, the cohesin off-loader, in old pro-B cells[64,65]. This may be the result of reduced Pax5 expression, a proposed repressor of Wapl, in old pro-B cells[41]. De-repression of Wapl by genetic manipulation has been previously shown to adversely affect TADs in pro-B cells, similar to our observation in old pro-B cells[41,48]. Because ectopic expression of Wapl in pro-B cells recapitulated several features of ageing, we hypothesize that increased Wapl and reduced Rad21 binding during physiologic ageing of B lymphocyte progenitors reduces TAD strength via increased disruption of cohesin-dependent chromatin loop extrusion. Age-associated epigenetic dysregulation of B cell development identified here may impact antibody-dependent immunity in older individuals.

## Online content

**Fig. 5 | Age-associated changes in chromatin structure at the *Igh* locus.**
**a**, Schematic representation of the mouse *Igh* locus. Regulatory elements are displayed as ovals. Black and red triangles represent CBEs with opposite orientations. **b**, Difference Hi-C heatmap showing a part of chromosome 12; *Igh* TAD is boxed. **c**, Contact frequency heatmaps of the *Igh* TAD in young and old pro-B cells. A stripe from 3′ CBEs across the *Igh* locus is indicated by dashed box and zoomed in on the left side of each heatmap. Schematics on top represent the locus to scale. Coloured sections represent 3′ *Igh* domain (dark blue), proximal $V_H$ genes (green), middle $V_H$ genes (pink) and distal $V_H$ genes (yellow). A difference map for the same region is shown on the right. The red (RI), green (RII) and blue (RIII) bars above the difference map indicate locations of bacterial artificial chromosome (BAC) probes used for FISH assays in **f** and **g**. RPKM, reads per kilobase per million mapped reads. **d**, Age-dependent histone modifications

and CTCF and Rad21 binding at the *Igh* locus. Tracks from replicate young and old pro-B cell samples are shown. **e**, Quantification of H3K27ac, Rad21, CTCF and H3K27me3 ChIP–seq (*n* = 2) signals in the 2 Mb region of the *Igh* locus that contains $V_H$ gene segments. Proximal, middle and distal refer to locations relative to the 3′ end of the locus (C-J-$D_H$) in part **d**. 'Relative signal' represents read counts per million (CPM) of each region relative to total reads. Bar graph represents the mean of two replicate experiments, with each replicate shown as a dot. **f**,**g**, RI, II and III BAC probes (part **b**) were used in FISH with young and old *Rag2*[−/−] pro-B cells. Representative nuclei and probe colours are as indicated (**f**). Dot plots of spatial distances between indicated probes (*n* = 100) (**g**). Non-B cells represent *Rag2*[−/−] bone marrow cells depleted of CD19[+] pro-B cells. Unpaired two-sided *t*-test was used for **g**, with *P* values indicated.

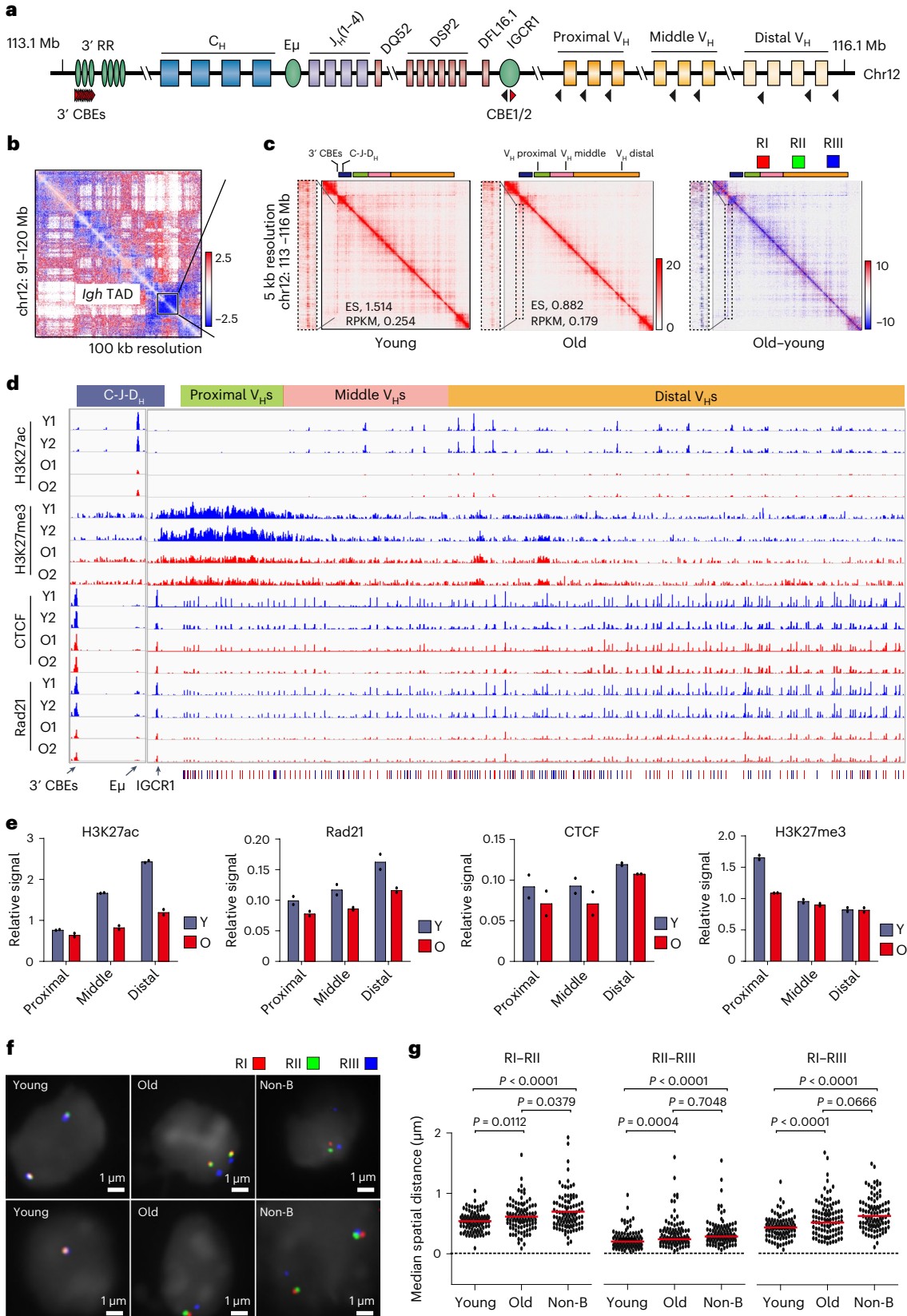

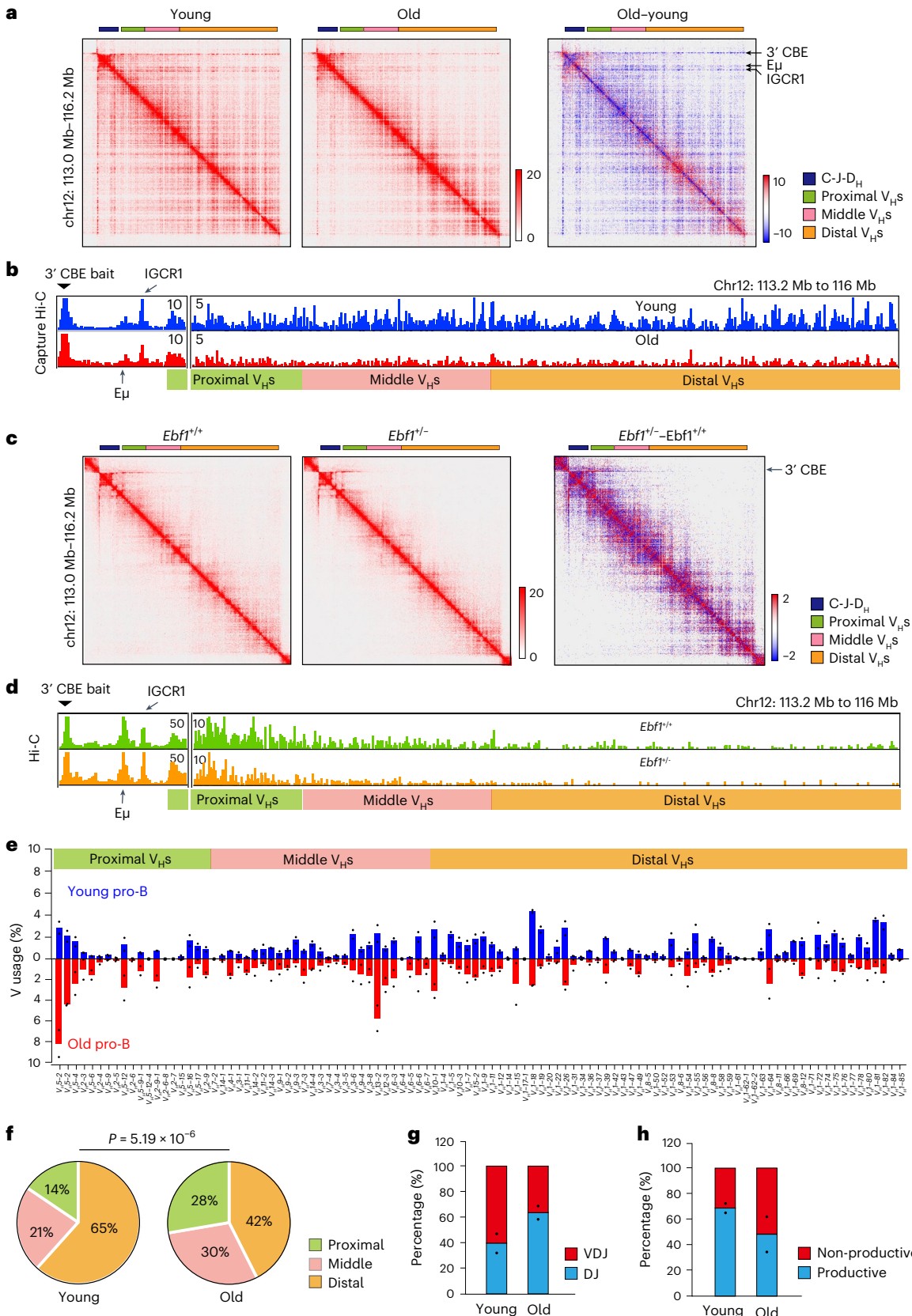

**Fig. 6 | Functional consequences of age-associated impairment of *Igh* locus contraction. a,c**, Contact frequency heatmaps of capture Hi-C (**a**) or Hi-C (**c**) showing the *Igh* TAD in young and old pro-B cells (**a**) or *Ebf1*$^{+/+}$ and *Ebf1*$^{+/-}$ pro-B cells (**c**). Difference maps for the same region are shown on the right; regions with increased and decreased interactions in old or *Ebf1*$^{+/-}$ pro-B cells are coloured red and blue, respectively. Schematics on top represent the *Igh* locus to scale; colours represent different parts of the locus as indicated to the right of the figure. **b,d**, Virtual 4C tracks were derived from capture Hi-C in young and old (**b**) or Hi-C in *Ebf1*$^{+/+}$ and *Ebf1*$^{+/-}$ (**d**) pro-B cells using 3′ CBEs as the bait. Specific interactions with Eµ and IGCR1 appear as peaks (left); widespread interactions with different

parts of the V$_H$ regions are shown. **e**, Rearrangement frequencies ($n = 2$) of all V$_H$ gene segments in young and old pro-B cells from C57BL/6J mice. **f**, Pie graph showing differential proximal, middle and distal V$_H$ gene utilization in young and old pro-B cells. A one-sided chi-squared test was used to determine $P$ value. **g**, Proportion of partial (DJ$_H$) and complete (V$_H$DJ$_H$) *Igh* rearrangements in young and old pro-B cells from C57BL/6J mice ($n = 2$) are shown. **h**, Proportion of productive (those that can encode IgH protein) and non-productive V$_H$DJ$_H$ junctions in young and old pro-B cells from C57BL/6J mice ($n = 2$). Bar graph represents the mean of two replicate experiments, with each replicate shown as a dot for **e**, **g** and **h**.

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

## Methods

This study is compliant with all the relevant ethical regulations. All mouse experiments were performed under protocols approved by National Institute on Aging (NIA) Institutional Animal Care and Use Committees (338-LMBI-2022).

### Mice

Young (8–12 weeks) recombination activating gene 2-deficient ($Rag2^{-/-}$) mice[35,36] were purchased from Jackson Lab (stock no. 008449) and maintained to be old (100–110 weeks) in house by NIA Comparative Medicine Section. Young (8–12 weeks) and old (100–110 weeks) C57BL/6J mice were provided by the NIA Aged Rodent Colonies. The mice used in the experiments comprised both males and females, and for each experiment, the sexes of both young and old mice were matched. In each experiment, two to six mice were pooled on the basis of the required cell count and the actual number of cells obtained.

Mouse husbandry was conducted under standard circadian rhythms of 12 h of darkness followed by 12 h of light, at a constant temperature of $22 \pm 2\,°C$ and humidity ranging between 45% and 70%.

### Cell lines

$Rag2^{-/-}Ebf1^{+/+}Pax5^{+/+}$, $Rag2^{-/-}Ebf1^{+/-}Pax5^{+/+}$ and $Rag2^{-/-}Ebf1^{+/-}Pax5^{+/-}$ pro-B cells are cultured as previous described[40].

The wild-type D345 cell line was generously provided by David G. Schatz of Yale University. These cells were maintained in RPMI1640 (Gibco) supplemented with 10% foetal bovine serum (FBS, Gemini) and a 1× penicillin–streptomycin solution (Gibco).

293T human embryonic kidney cells used for lentiviral expression was purchased from ATCC (cat. no. CRL-3216) and cultured in Dulbecco's modified Eagle medium (Gibco) with 10% FBS (Gemini) and 1× penicillin–streptomycin solution (Gibco).

The Platinum-E (Plat-E) cell line used for retroviral expression was purchased from Cell Biolabs (cat. no. RV-101) and cultured in Dulbecco's modified Eagle medium (Gibco) with 10% FBS (Gemini), 1× penicillin–streptomycin solution (Gibco), 10 mM HEPES and 0.3 µg ml$^{-1}$ L-glutamine. All cells were cultured at 37 °C in a humidified atmosphere with 5% $CO_2$.

### Antibodies

**Flow cytometry.** Phycoerythrin (PE) anti-CD19 (BioLegend, 6D5, cat. no. 115508, 1:100), Brilliant Violet 421 (BV421) anti-B220 (BioLegend, RA3-6B2, cat. no. 103240, 1:100), fluorescein isothiocyanate (FITC) anti-IgM (BioLegend, RMM-1, cat. no. 406506; 1:50), PE anti-CD43 (BD Biosciences, S7, cat. no. 553271, 1:100), H3K27ac-AF488 conjugated (Cell Signaling Technology, cat. no. 15485S. 1:50) and IgG isotype control-AF488 conjugated (Cell Signaling Technology, cat. no. 4340S; 1:50) were used.

**ChIP–seq and HiChIP.** Anti-H3K27ac (Active Motif, cat. no. 39133; 1:500), anti-H3K27me3 (Diagonode, cat. no. C15410195; 1:500), anti-CTCF (Abcam, cat. no. ab70303; 1:250), anti-Rad21 (Abcam, cat. no. ab992; 1:500), anti-Brg1 (Abcam, cat. no. ab110641; 1:200) and anti-p300 (Cell Signaling Technology, cat. no. 57625S; 1:333) were used.

### Pro-B cell purification

For $Rag2^{-/-}$ mice, total bone marrow was extracted from tibia and femurs, and erythrocytes were lysed. Pro-B cells were purified by combining positive selection using CD19$^+$ selective beads (Stem Cell Technology, cat. no. 18954) and sorting by CD19$^+$ and B220$^+$ markers.

For C57BL/6J mice, total bone marrow was extracted from tibia and femurs, and erythrocytes were lysed. Cells were pre-purified using CD19$^+$ selective beads (Stem Cell Technology, cat. no. 18954) and sorted with IgM$^-$B220$^+$CD43$^+$ markers. Flow cytometry experiments were performed on BD FACSAria II Cell Sorter. FlowJo software was used for data analysis.

### In situ Hi-C

Genome-wide in situ Hi-C was performed with $Rag2^{-/-}$ young and old primary pro-B cells, and $Rag2^{-/-}Ebf1^{+/+}Pax5^{+/+}$, $Rag2^{-/-}Ebf1^{+/-}Pax5^{+/+}$, $Rag2^{-/-}Ebf1^{+/-}Pax5^{+/-}$, D345-BFP control and D345-Wapl-BFP pro-B cell lines, using the Arima Hi-C Kit (Arima Genomics), including KAPA Hyper Prep indexing and library amplification (cat. no. KK8500, Roche Molecular Systems Inc) according to the manufacturer's instructions. For each assay, $1 \times 10^6$ cells were used as the input materials and two biological replicates were performed for each group. Samples were sequenced $2 \times 150$ bp on an Illumina NovaSeq instrument at the Single Cell and Transcriptomics Core at Johns Hopkins University.

### RNA-seq

Total RNA was prepared using the Direct-zol RNA Miniprep Kit (Zymo Research, cat. no. R2051) according to manufacturer's instructions. The RNA libraries for young and old $Rag2^{-/-}$ primary pro-B cells (four biological replicates for each group) were prepared using the NEXTFLEX Rapid Directional RNA-Seq Kit (PerkinElmer, cat. no. NOVA-5138-07) and sequenced $2 \times 75$ bp on an Illumina NovaSeq instrument at the Single Cell and Transcriptomics Core at Johns Hopkins University.

RNA libraries for $Rag2^{-/-}Ebf1^{+/+}Pax5^{+/+}$, $Rag2^{-/-}Ebf1^{+/-}Pax5^{+/+}$, $Rag2^{-/-}Ebf1^{+/-}Pax5^{+/-}$ and D345-BFP control and D345-Wapl-BFP pro-B cell lines (two replicates for each group) were prepared using SMARTer Stranded Total RNASeq Kit v2 (Takara, cat. no. 634412) and sequenced $1 \times 100$ bp on an Illumina NovaSeq instrument at the Single Cell and Transcriptomics Core at Johns Hopkins University.

RNA libraries for pro-B cells infected with Ebf1-GFP or GFP-containing retrovirus were prepared using the SMART-Seq mRNA kit (Takara, cat. no. 634773) and sequenced $2 \times 100$ bp on an Illumina NovaSeq instrument at the Computational Biology and Genomics Core in NIA.

### ChIP–seq

For the ChIP of histone modifications, CTCF and Rad21, both young and old primary $Rag2^{-/-}$ pro-B cells underwent crosslinking with 1% formaldehyde (Sigma) for 10 min at room temperature. The reaction was subsequently quenched with 125 mM glycine, and cell lysis was initiated in a buffer containing 1% sodium dodecyl sulfate. Chromatin was then sheared using a Bioruptor (Diagenode) in cycles of 30 s on and 30 s off, totalling 15 min of shearing time. Immunoprecipitation was performed using specific antibodies, with $1 \times 10^6$ cells used for each assay, and antibody dilutions followed established protocols.

As for Brg1 and p300 ChIPs, we followed the protocol by Bossen et al.[66], with some modifications. In this case, two million young and old primary $Rag2^{-/-}$ pro-B cells were first crosslinked with 1.5 mM ethylene glycol bis(succinimidyl succinate) at room temperature with rotation for 15 min. Subsequently, the cells were fixed with 1% formaldehyde (Sigma) for 15 min at room temperature with rotation and quenched with 200 mM glycine. After two phosphate-buffered saline (PBS) washes, the cells were lysed in a buffer containing 1% sodium dodecyl sulfate. Notably, we extended the sonication time to a total of 20 min of shearing time, while the subsequent steps of the protocol mirrored the previously mentioned procedure.

### RT–PCR

Total RNA was isolated as previously described. A total of 100 ng of RNA was used to generate complementary DNA (cDNA) with SuperScript III (Thermo Fisher Scientific, cat. no. 18080051) using random hexamers according to the manufacturer's protocol. Quantitative polymerase chain reaction (RT–PCR) was performed with iTaq Universal SYBR (Bio-Rad, cat. no. 1725125) using primers described in Supplementary Table 6. Three independent experiments were carried out. Cobp2 was used as a control[37]. Data were processed by the delta–delta CT method.

## FISH

FISH was performed as previously described[45,46]. Briefly, cells were fixed on poly-L-lysine coated slides using $1.5 \times 10^6$ cells per slide. Slides were kept for 15 min at 37 °C, followed by fixation with 4% paraformaldehyde in PBS. Cells were further washed with 0.1 M Tris–HCl (pH 7.4), followed by PBS. Fixed cells were treated with 100 µg ml⁻¹ RNase A in PBS, permeabilized using 0.5% saponin/0.5% Triton X-100/PBS for 30 min at room temperature. Probes were listed in Supplementary Table 6.

Labelled probes were denatured at 75 °C for 5 min and applied to cells that were denatured in formamide at 73 °C, followed by incubation in a dark humid chamber. Images were acquired using a Nikon Eclipse Ti2 microscope equipped with a 60× lens. Depending on the size of the nucleus, 40–50 serial optical sections spaced by 0.2 µm were acquired. The datasets were deconvolved using NIS-Elements software (Nikon). For RI, RII and RIII spatial distance measurements, we calculated the centre-to-centre distances between the parent and child, specifically focusing on the child's nearest parent. To measure the distances between the nuclear periphery and the centres of *Foxo1* or *Ebf1* probes[67], we used DAPI as the reference point. The nuclei of individual cells were delineated through DAPI staining, and the measurements were based on the child (*Foxo1* or *Ebf1* probes) intersecting with the parent (DAPI-stained nuclei). For the distances between Foxo1 and γ-satellite signals, we measured the centre-to-centre distances between the parent and child, focusing on the child's nearest parent. We selected the minimum distances for each parent since γ-satellite has multiple signals. While each parent can have more children, each child has only one parent for Nikon measurements.

## Retroviral expression of Ebf1

Retrovirus was produced by transfection of Plat-E cells with retroviral plasmids containing *Ebf1* cDNA and a selective green fluorescent protein (GFP) marker (*pMYs-Ebf1-IRES-EGFP*)[42], or with GFP-only plasmids as the control (*pMSCV-IRES-GFP*, Addgene cat. no. 20672). Supernatants were collected at both 48 and 72 h post-transfection and subsequently concentrated using the Retro-X Concentrator (Takara, cat. no. 631456), following the manufacturer's instructions. Young and old *Rag2⁻/⁻* CD19⁺ cells were cultured with 2 ng ml⁻¹ of IL-7 for three days and then exposed to the resuspended virus with the supplement of 10 µg ml⁻¹ polybrene. The cells were cultured for an additional 72 h and subsequently sorted on the basis of GFP expression before subsequent analysis.

## Lentiviral expression of Wapl

Lentivirus was produced by transfection of 293T cells with the *pHIV-Wapl-IRES-BFP* and *pHIV-IRES-BFP* plasmids. To make these plasmids, the *BFP* DNA fragment was amplified from plasmid *pEJS614_pTetR-P2A-BFPnls/sgNS* (Addgene, cat. no. 108650), and then integrated into *pHIV-IRES-Zsgreen* (Addgene, cat. no. 18121) using the NEBuilder HiFi DNA Assembly Master Mix (New England Biolabs, cat. no. E2621S) as per the manufacturer's instructions, resulting in the generation of *pHIV-IRES-BFP* plasmids. Subsequently, *Wapl* cDNA was introduced into *pHIV-IRES-BFP*, creating the *pHIV-Wapl-IRES-BFP*. Supernatants were collected at both 48 and 72 h post-transfection and subsequently concentrated using the Lenti-X Concentrator (Takara, cat. no. 631232). D345 cells were infected with the resuspended lentivirus with the addition of 8 µg ml⁻¹ polybrene. After infection, the cells were cultured for 72 h and sorted on the basis of BFP expression. The sorted cells were then cultured for 2–3 additional days to obtain an increased number of BFP-positive cells for experimental purpose.

## H3K27ac HiChIP

The H3K27ac HiChIP procedure was executed employing the Arima Hi-C+ Kit (Arima Genetics Inc. cat. no. A101020), strictly adhering to the manufacturer's guidelines outlined in the Arima-HiC+ documents A160168 v00 (HiChIP) and A160169 v00 (library preparation). In each sample, a total of $5 \times 10^6$ cells were utilized. These libraries were individually barcoded and then combined for sequencing on an Illumina NovaSeq instrument.

## Intracellular staining of H3K27ac

Total bone marrow cells were obtained from young and old *Rag2⁻/⁻* mice to evaluate H3K27ac levels through intracellular staining. Cell surface markers were stained with PE anti-CD19 and BV421 anti-B220, washed twice with PBS and then stained in PBS with the eBioscience Fixable Viability Dye eFluor 780 (Invitrogen, cat. no. 65-0865-14). Following this, cells were washed with PBS and fixed and permeabilized using the Fix/Perm buffer provided by the Transcription Factor Buffer Set (BD, cat. no. 562725) according to the manufacturer's instructions. Fixed and permeabilized cells were then washed twice with the Perm/Wash buffer form the Transcription Factor Buffer Set and stained with either anti-H3K27ac or IgG isotype control antibodies. Subsequently, the cells were washed three times with Perm/Wash buffer before being subjected to flow cytometry analysis for H3K27ac quantification.

## Capture Hi-C

Whole-genome Hi-C libraries were first generated as previously described. To enrich the *Igh* locus (mm10, chr12: 113,201,001–116,030,000), SureSelect Target Enrichment probes (Supplementary Table 5) with 2× tiling density were designed and manufactured by Agilent (Agilent Technologies, Inc.). Hi-C libraries were hybridized to probes as specified by the manufacturer. Enriched libraries were sequenced using Illumina NextSeq sequencer to generate 2× 150 bp reads.

## VDJ-seq

VDJ-seq was performed following the HTGTS-Rep-Seq protocol[49,50]. Briefly, 200 ng to 2 µg of genomic DNA isolated from primary pro-B cells sorted from young and old C57BL/6J mice was sonicated with Covaris using the setting of Durations 10 s, Peak power 50, Duty Factor 20%, Cycles/Burst 200. Sheared DNA was linearly amplified using a biotinylated $J_H1$ primer and enriched with streptavidin C1 beads (Thermo Fisher Scientific). A bridge adaptor was ligated to enriched single stranded DNA to the 3′ end and then amplified by a nested primer set. Products were further amplified using P5–I5 and P7–I7 primers to be final libraries and sequenced by Illumina MiSeq sequencer to generate paired-end 250 bp reads. Primers or oligos used are listed in Supplementary Table 6.

## Sequencing data bioinformatics analysis

For all analyses, we utilized the mouse reference genome mm10 and annotations from GENCODE (M21)[68]. The RNA-seq data and ChIP–seq data generated in this study were pre-processed from raw reads to expression level, differentially expressed genes analysis, mapped reads or tracks (bigWig files) following the procedures outlined in our previous studies[69,70].

This manuscript incorporates heatmaps generated from pooled and down-sampled reads, ensuring equal representation from Hi-C or HiChIP data. Related statistical tests were performed with function in Python SciPy[71] package. If not specially mentioned, most plots were generated by the matplotlib[72] and seaborn[73].

## RNA-seq analysis

RNA-seq raw reads of young and old *Rag2⁻/⁻* primary pro-B cells were trimmed and mapped to the mouse reference genome (mm10) using STAR (v2.7.3a)[74] and calculated the raw count using featureCounts package (gene-level) and fragments per kilobase of transcript per million mapped reads (or transcript per million) using RSEM package[75].

For the rest of those RNA-seq experiments, raw reads were trimmed and mapped to mm10 by STAR (v2.7.3a)[74] and quantified into reads per kilobase per million mapped reads with Cufflinks (v2.2.1)[76]. Bigwig tracks were also generated by STAR as signal quantified as reads per kilobase per million mapped reads. RNA-seq tracks were shown by IGV[77].

## ChIP–seq analysis

ChIP–seq raw reads were mapped to the mouse reference genome mm10 by Bowtie2 (v2.3.5)[78]. Only non-redundant reads with mapping quality score ≥10 were saved as BED files for the following analysis. Bigwig tracks were generated by bamCoverage in deepTools (v3.3.0)[79] with parameters of --ignoreDuplicates --minMappingQuality 10 --normalizeUsing CPM for visualization and quantification aggregation analysis. Visualization of ChIP–seq tracks were performed by IGV[77] or the cLoops2 plot module (v0.0.2)[80]. ChIP–seq peaks were identified using the cLoops2 callPeaks module. For H3K27me3, CTCF, Rad21, Brg1 and p300 ChIPs, the key parameters -eps 150 and -minPts 20,50 were employed. For H3K27ac data, the parameters -eps 150,300, -minPts 20,50 and -sen were used. Peak identification was performed on each replicate individually, followed by the compilation of a union set encompassing both conditions (young and old) and overlapping replicates. Differential peaks were identified for young and old conditions using a Poisson test. The test utilized average counts from replicates for each peak and considered normalized total reads of 30 million. A $P$ value cut-off of $<1 \times 10^{-5}$ and a fold change cut-off of ≥2 were used to identify significant differential peaks. ChIP–seq peak or domain regions aggregation analyses, including heatmaps and average profile plots, were generated using the computeMatrix and plotHeatmap commands from the deepTools (v3.3.0) package. Quantification of ChIP–seq signals in Hi-C compartments was performed by the cLoops2 quant module. H3K27ac peak annotations were performed by the anoPeaks.py script in cLoops2 package, which is available at https://github.com/YaqiangCao/cLoops2/blob/master/scripts/anoPeaks.py. H3K27ac peaks within a 2-kb range upstream or downstream of the transcription start site (TSS) are annotated as promoters, while those outside this range are labelled as enhancers for the downstream analysis.

## Hi-C, H3K27ac HiChIP and capture Hi-C analysis

Hi-C, H3K27ac HiChIP and capture Hi-C raw reads were processed to the mouse reference genome mm10 by HiCUP (v0.7.2)[81] with the settings of Arima. High-quality and unique paired-end tags (PETs) from HiCUP were further processed to the HIC file through Juicer (v1.6.0)[82] for visualization with Juicebox[83]. These PETs were also processed with cLoops2 (v0.0.2)[80] for quantifications. Chromosome X was excluded from the analysis. Replicates from H3K27ac HiChIP samples were combined and down-sampled equally to 39 million PETs for all following analysis.

Hi-C compartment analysis of eigenvectors first principal component (PC1) was obtained by hicPCA in HiCExplorer3 package (v3.6)[84] with parameters of -noe 1 at the resolution of 100 kb. We employed a two-pass Mahalanobis distance (MD) calculation, an effective method for outlier detection based on data point distribution, along with a chi-squared test for Hi-C compartment PC1 at the 100-kb bin level to detect PC1 sign flip bins. This approach aligns with the strategy employed in the recent study dcHiC[85]. The MD is calculated as $\mathrm{MD} = \mathrm{diag}((X - X_c) \times C^{-1} \times (X - X_c)^T)$, where $X$ is the matrix for PC1 obtained from the Hi-C compartment analysis. Each row represents a bin, and each column represents a sample (young or old). $C^{-1}$ is the inverse covariance matrix of $X$, $X_c$ is vector of row-wise mean of $X$ and diag is the function used to extract a diagonal array from a matrix. We performed the chi-squared test with the MD, and a $P$ value cut-off of 0.01 was set as the significance cut-off for detecting outliers. For the first pass calculation of MD, outliers were detected with all bins. For the second pass calculation, outliers were removed for calculation $X_c$ and $C^{-1}$. Then, MD distances were calculated for all bins on the basis of the first pass-outlier removed $X_c$ and $C^{-1}$. The code for the analysis to generate the MD distances and $P$ values, the plot, significant changed bins and associated genes is implemented as the comparComp.py script in the cLoops2 package and is available at https://github.com/YaqiangCao/cLoops2/blob/master/scripts/compareComp.py.

Hi-C TADs were called by Juicer arrowhead with parameters of -r 25000 -k KR. We employed ESs to quantify TAD interaction strength.

The ES is calculated by dividing the number of PETs interacting exclusively within a TAD by the number of PETs with one end within the TAD. Like the comparison of compartments, we implemented the two-passes MD calculation and chi-squared test to get the significant changed TADs. The MD is calculated as $\mathrm{MD} = \mathrm{diag}((X - X_c) \times C^{-1} \times (X - X_c)^T)$ where $X$ is the matrix for TAD ESs. Each row represents a TAD, and each column represents a sample (young or old). $C^{-1}$ is the inverse covariance matrix of $X$, $X_c$ is vector of row-wise mean of $X$, and diag is the function used to etract a diagonal array from a matrix. We performed the chi-squared test with the MD, and a $P$ value cut-off of 0.01 was set as the significance cut-off for detecting outliers. For the first pass caculation of MD, outliers were detected with all TADs. For the second pass caculation, outliers were removed for calculation $X_c$ and $C^{-1}$. Then, MDs were caculated for all bins based on the first pass-outlier removed $X_c$ and $C^{-1}$. The TAD ES was calculated by dividing the number of intra PETs (PETs with both ends located in the TAD) by the number of inter PETs (PETs with only one end located in the TAD). We used the ES as a metric because we found that the IgH domain had the highest change, along with domains contain Ebf1 and Pax5 in the top. The code for the analysis to generate the MD distances and $P$ values for the TADs, the plot, significantly changed TADs and associated genes is implemented as the compareDom.py script in the cLoops2 package and is available at https://github.com/YaqiangCao/cLoops2/blob/master/scripts/compareDom.py.

H3K27ac HiChIP data viewpoints analysis was used to examine the changes in interactions for H3K27ac ChIP–seq peaks. For each viewpoint (H3K27ac peak), an ES was calculated using the H3K27ac HiChIP contact matrix at a 1-kb resolution. The ES was centred on the peak and incorporated 100-kb upstream and downstream regions as the background. A higher ES indicates stronger interactions originating from the viewpoint. To visually emphasize the signals, the upper right corner matrices of the ±100-kb contact matrices centred around each viewpoint (H3K27ac peak) were subjected to two rounds of $\log_{10}$ transformation and presented as heatmaps. cLoops2 plot module was utilized to generate visualizations of HiChIP data, including one-dimensional signal profiles, ChIP–seq tracks, arches depicting the number of PETs for combinations of H3K27ac peaks and heatmaps or scatter plots.

## Virtual 4C analysis

Virtual 4C analysis was performed using Cooler2 to extract all contacts involving the 3′ CBE from the Hi-C contact maps[86]. The bin size used for this analysis was 5,000 bp and 'VC_SQRT' was used as the Hi-C normalization method. The bin used for the 3′ CBE bait was chr12:113220000-113225000 (mm10).

## VDJ-seq analysis

VDJ-seq analysis was performed using HTGTS-Rep[49,50]. This computational pipeline determined the frequency of V and D gene usage for each HTGTS experiment. This pipeline was also used to determine the distribution of CDR3 lengths and the percent of productive versus non-productive recombination events.

## Statistics and reproducibility

The presentation of data and the statistical tests utilized are detailed in each figure legend. For datasets with a sample size ($n$) less than three, individual data points are displayed, assuming a normal data distribution, but this was not formally tested. Unless otherwise specified, the statistical tests were performed using GraphPad Prism v.9.5.0. For Fig. 4f, all $P$ values were reported as 0.000000 by scipy.stats.wilcoxon. No statistical method was used to predetermine sample size, but our sample sizes are similar to those reported in previous publications[54]. No data were excluded from the analysis. All biologically independent replicates are explicitly identified in the figure legends. The gating strategy of flow cytometry is shown in Supplementary Fig. 1. Fluorescence-activated cell sorting data and microscopy (FISH) images are representative of at least three independent biological replicates.

Experiments were not randomized, and data collection and analysis were not performed blind to experimental conditions.

## Reporting summary

Further information on research design is available in the Nature Portfolio Reporting Summary linked to this article.

## Data availability

All sequencing data generated in this study, including RNA-seq, ChIP–seq, Hi-C, capture Hi-C, VDJ-seq and HiChIP data, have been deposited in the Gene Expression Omnibus (GEO) under accession codes GSE214438. Source data are provided with this paper.

## Code availability

Hi-C compartment and domain comparison script comparComp.py and comparDom.py are integrated into the cLoops2 package and available via GitHub at https://github.com/YaqiangCao/cLoops2/blob/master/scripts/compareComp.py and https://github.com/YaqiangCao/cLoops2/blob/master/scripts/compareDom.py. The script used to show Fig. 4e is available via GitHub at https://github.com/YaqiangCao/cLoops2/blob/master/scripts/showAggVPs.py.

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

## Acknowledgements

This research was supported by the Intramural Research Program of the National Institute on Aging (R.S. and I.B.) and National Heart, Lung, and Blood Institute (K.Z.), as well as Max Planck for Immunobiology and Epigenetics (R.G.). Data analysis of this work utilized the computational resources of the NIH HPC Biowulf cluster (http://hpc.nih.gov). Special thanks to C. Dunn at the NIA Flow Core for assistance with FACS sorting, J. Fan and S. De at the NIA Computational Biology and Genomics Core, C. Sherman-Baust in the Laboratory of Molecular Biology and Immunology (LMBI) at NIA, and L. Orzolek at the Transcriptomics and Deep Sequencing Core in Johns Hopkins Medicine for their support with Illumina sequencing. Thanks to A. Lenaerts and H.-J. Schwarz at the Max Planck Institute for their assistance in preparing the pMYs-Ebf1-IRES-EGFP plasmid. We also appreciate the contributions of M. Kaileh in LMBI and the Comparative Medicine Section at NIA for animal care, A. Singh for insightful input on B cell development and C. Sasaki for assistance with ChIP–seq quality checks.

## Author contributions

Conceptualization was carried by F.M., Y.C. and R.S. Methodology was created by F.M., Y.C., F.Z.B., M.B. and H.D. Investigation was performed by F.M., F.Z.B., M.B., H.D., L.Z., R.R., S.N., X.Q. and A.H.B. Data analysis and visualization were performed by F.M., Y.C., N.O., B.P., R.R., D.S., A.Z. and A.H.B. Funding acquisition, project administration and supervision were carried out by I.B., K.Z., R.G. and R.S. Writing of the original draft was carried out by F.M., Y.C., N.O., H.D., B.P. and R.S. Writing—review and editing of the text—was performed by F.M., Y.C., F.Z.B., N.O., L.Z., R.R., I.B., K.Z., R.G. and R.S.

## Competing interests

The authors declare no competing interests.

## Additional information

**Extended data** is available for this paper at https://doi.org/10.1038/s41556-024-01424-9.

**Correspondence and requests for materials** should be addressed to Ranjan Sen.

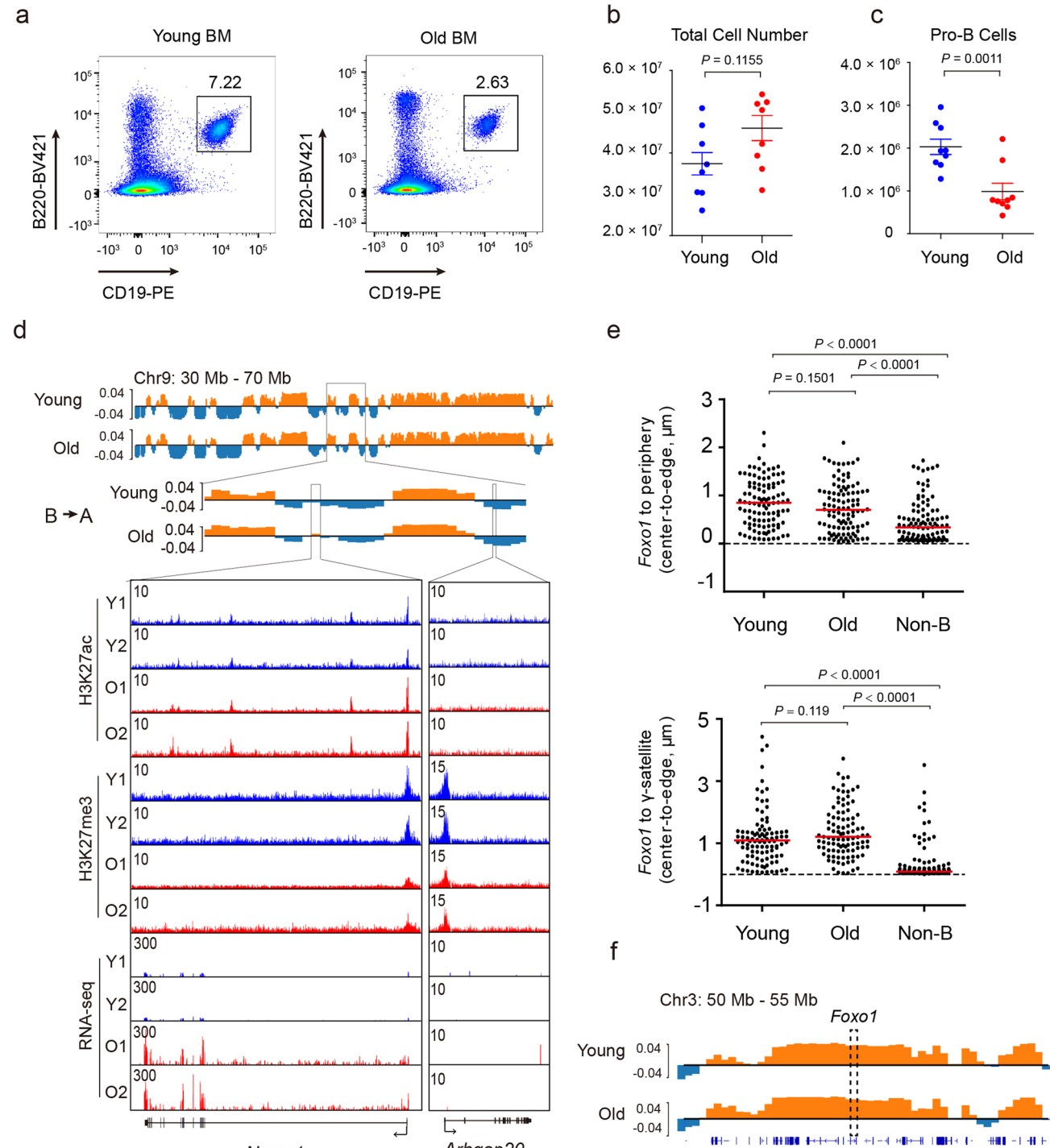

**Extended Data Fig. 1 | Pro-B cell number is reduced in aged mice.**
(**a**) Representative flow cytometry plots showing the population of *Rag2*⁻/⁻ pro-B cells from young and old bone marrow. Numbers indicate percentage of cells in the gates. Gating strategy was shown in Source Data. (**b**) Average total number of bone marrow cells from tibias and femurs of young and old *Rag2*⁻/⁻ mice (n = 8). Data are presented as mean ± SEM. Unpaired two-sided *t* test was used, with *P*-values indicated. (**c**) Number of pro-B cells after CD19⁺ selection and CD19⁺ B220⁺ sorting from young and old *Rag2*⁻/⁻ mice (n = 9). Each point represents an individual mouse. Data are presented as mean ± SEM. Unpaired two-sided *t* test was used, with *P*-values indicated. (**d**) An example locus transitioned from compartment B (young) to compartment A (old) with age. Top tracks show Hi-C PC1 of a segment of chromosome 9 that contains *Ncam1* in pro-B cells at different ages; the region corresponding to the *Ncam1* locus is expanded below. Normalized tracks for H3K27ac and H3K27me3 ChIP-Seq and RNA-Seq across the Ncam1 locus are shown with IGV. *Arhgap20* (right) serves as a control gene located in compartment B that is unaffected by age. (**e**) Dot plots for distance between *Foxo1* locus to the nuclear periphery (upper panel) or to the nearest γ-satellite signal (bottom panel). Median is shown in red line. N = 100. Unpaired two-sided *t* test was used, with *P*-values indicated. (**f**) Hi-C PC1 showing a segment of chromosome 3 containing the *Foxo1* locus at different ages.

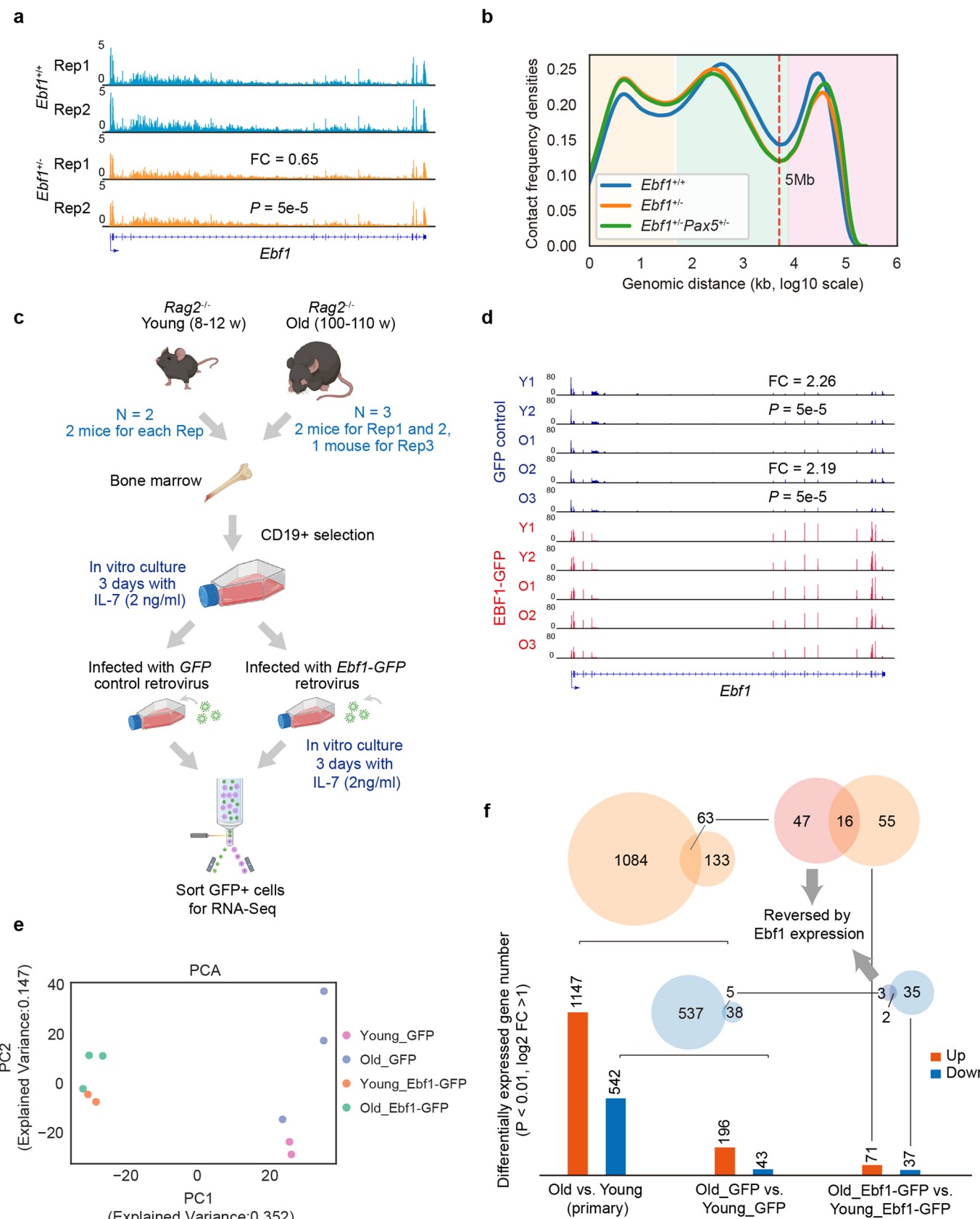

**Extended Data Fig. 2 | See next page for caption.**

**Extended Data Fig. 2 | Ebf1 perturbation causes chromatin structure and expression changes in pro-B cells.** (**a**) Normalized tracks for RNA-Seq of *Ebf1*$^{+/+}$ and *Ebf1*$^{+/-}$ pro-B cells across the *Ebf1* locus are shown with IGV. "FC" represents fold change, and the Poisson test was applied with *P* values indicated (n = 2). (**b**) Average Hi-C contact plots (n = 2) derived from *Ebf1*$^{+/+}$ and *Ebf1*$^{+/-}$ pro-B cells. Color shading refers to: compartments: pink, TADs and loops: green, and close interactions: yellow. (**c**) Experimental scheme for overexpressing Ebf1 in cultured pro-B cells. (**d**) Normalized tracks for RNA-Seq across the *Ebf1* locus of different samples are shown with IGV. "FC" represents fold change, and the Poisson test was applied with P values indicated (n = 2). (**e**) Principal component analysis (PCA) plot depicting the distribution of RNA-Seq from different samples as indicated. (**f**) Schematic representation of differentially expressed gene counts across multiple comparisons.

a

### Top 5 decreasing TADs

| TAD Domain ID | Size* | Old/Young (ES) | Old/Young (RPKM) | Genes or Locus Invovled |
|---|---|---|---|---|
| chr12: 113200000-116000000 | 2.8 Mb | 0.5822296 | 0.703275862 | IgH |
| chr17: 3000000-6025000 | 3.0 Mb | 0.678052 | 0.820445637 | Nox3, Pisds2, Tfb1m, Arid1b, Tmem242, Cldn20, Zdhhc14, Synj2, Tiam2, Snx9, Scaf8, Ldhal6b |
| chr11: 43500000-45925000 | 2.4 Mb | 0.7183095 | 0.799455297 | Ebf1, Clint1, Ublcp1, Pwwp2a, Rnf145, Il12b, Ccnjl, Ttc1, Fabp6, Adra1b |
| chr1: 132375000-135800000 | 3.4 Mb | 0.7212093 | 0.829856172 | Sox13, Nfasc, Ptprv, Mybph, Zbed6, Ube2t, Tnni1, Zc3h11a, Optc, Tmcc2, Rabif, Lgr6, Nav1, Ipo9, Adipor1, Lrrn2, Lax1, Chil1, Btg2, Fmod, Dstyk, Ppp1r12b, Arl8a, Tmem81, Chit1, Ptpn7, Snrpe, Etnk2, Ren1, Atp2b4, Platr1, Syt2, Zc3h11a, Shisa4, Mdm4, Csrp1, Cntn2, Ppp1r15b, Tmem183a, Mgat4e, Gpr37l1, Kdm5b, Phlda3, Rbbp5, Ppfia4, Adora1, Pik3c2b, Prelp, Lmod1, Elf3, Klhl12, Timm17a, Plekha6, Cyb5r1, Kiss1, Myog, Rnpep, Golt1a |
| chr4: 44550000-45000000 | 0.5 Mb | 0.7243236 | 0.725498427 | Pax5, Grhpr, Zbtb5, Zcchc7 |

\* Size < 5 Mb

b

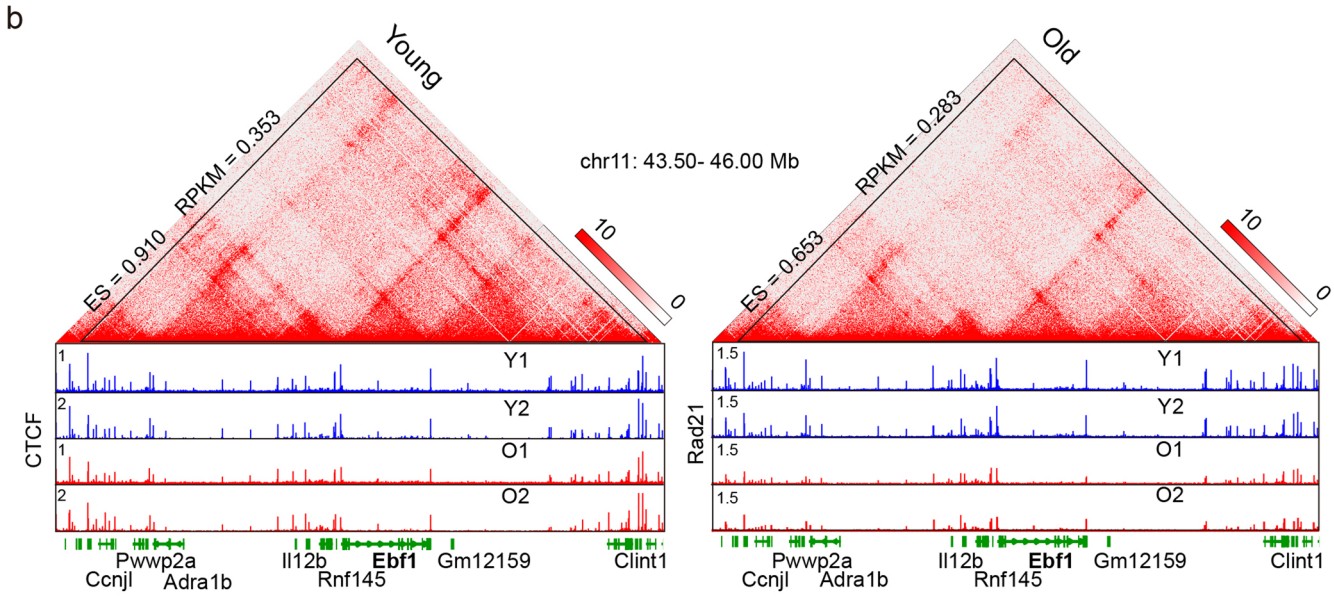

c

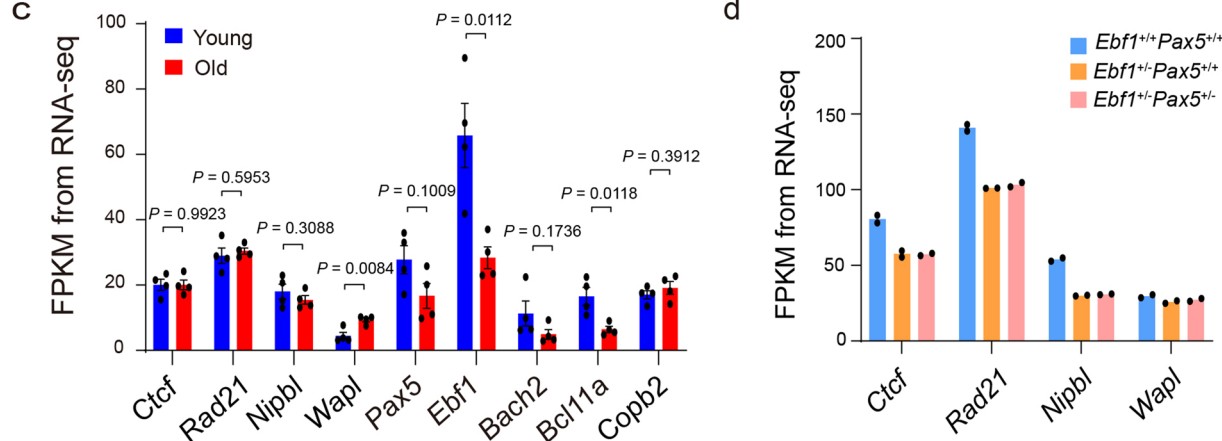

**Extended Data Fig. 3 | Most decreased TADs in aged pro-B cells.** (**a**) A chart showing information of top 5 decreased TADs. TAD size is limited to be less than 5 Mb. B cell associated genes are in blue. (**b**) An example showing one of the top decreased TAD containing Ebf1 locus. Young (left) and old (right) Hi-C contact frequency heat maps are in 5 kb resolution. Normalized tracks for CTCF (left) and Rad21 (right) ChIP-Seq across the TAD are shown with IGV. ES (enrichment score) and RPKM values of the TAD are annotated. (**c**) Normalized expression levels of key genes involved in either chromatin structure maintenance (*Ctcf*, *Rad21*, *Nipbl* and *Wapl*) or B cell development (*Pax5*, *Ebf1*, *Bach2* and *Bcl11a*) in young and old pro-B cells (n = 4). *Copb2* is shown as a control. Data are presented as mean ± SEM. Unpaired two-sided *t* test was used and *P*-values were indicated. (**d**) Normalized expression levels of key genes involved in chromatin structure maintaining in *Ebf1*⁺ᐟ⁺, *Ebf1*⁺ᐟ⁻ and *Ebf1*⁺ᐟ⁻*Pax5*⁺ᐟ⁻ pro-B cells (n = 2). Bar graph represents the mean of two replicate experiments, with each replicate shown as a dot.

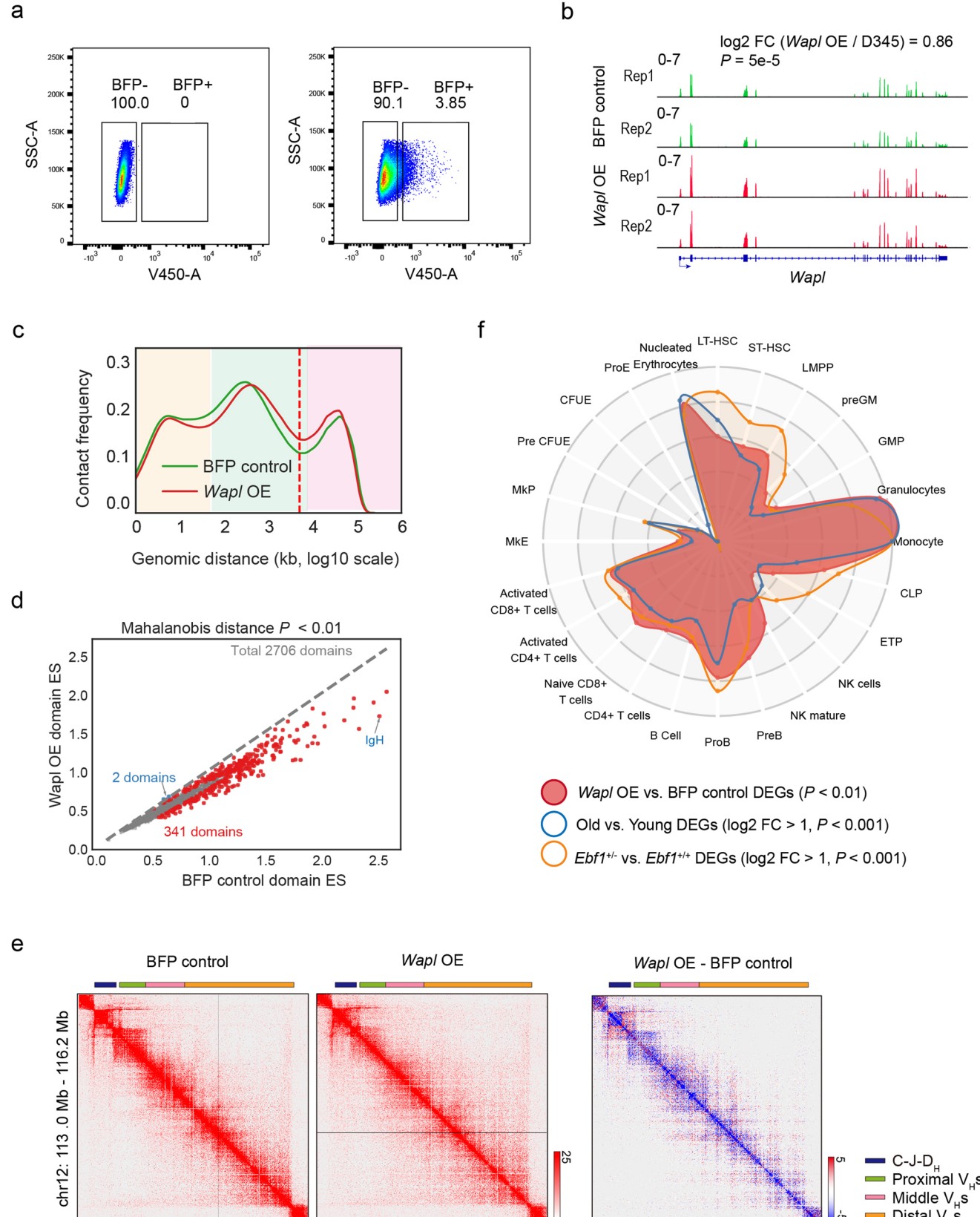

**Extended Data Fig. 4 | See next page for caption.**

**Extended Data Fig. 4 | Wapl overexpression remodels the chromatin structure in the recombinase deficient pro-B cell line (D345). (a)** Flow cytometric analysis showing the D345 pro-B cell line with (left) or without (right) infection with Wapl-BFP lentivirus. The percentages of BFP+ and BFP- cells are indicated. BFP+ cells are sorted for experiments. **(b)** Normalized tracks for RNA-Seq across the *Wapl* locus are shown with IGV. "FC" represents fold change, and the Poisson test was applied with the P value indicated (n = 2). **(c)** Average Hi-C contact plots (n = 2) derived from BFP control and Wapl overexpressed D345 cells. Color shading refers to: compartments: pink, TADs and loops: green, and close interactions: yellow. **(d)** Quantitative comparison of TAD strength derived from Hi-C data of BFP control and Wapl overexpressed D345 cells. Similar analysis and the same cutoffs as in Fig. 3a were used. Representative genes from the top-ranked changed TADs are displayed. **(e)** Hi-C contact frequency heat maps (pooled from 2 biological replicates) showing the Igh TAD in BFP control and Wapl overexpressed D345 pro-B cells. A difference map (*Wapl* OE minus BFP control) for the same region is shown on the right; regions with increased and decreased interactions in old are colored red and blue, respectively. Schematics on top represent the Igh locus to scale; colors represent different parts of the locus as indicated to the right of the figure. **(f)** CellRadar (https://karlssong. github.io/cellradar/) plot derived from transcriptional changes identified in comparison of young and old pro-B cells (blue), *Ebf1*$^{+/+}$ and *Ebf1*$^{+/-}$ pro-B cells (orange), and BFP control and Wapl overexpressed D345 cells. Differentially expressed genes (DEGs) with log2 fold change > 1 and *P*-value < 0.01 were used as input for young and old pro-B cells (blue), and *Ebf1*$^{+/+}$ and *Ebf1*$^{+/-}$ pro-B cells. DEGs with *P*-value < 0.01 were used as input for BFP control and Wapl overexpressed D345 cells because of the limited number of DEGs by a more stringent cutoff.

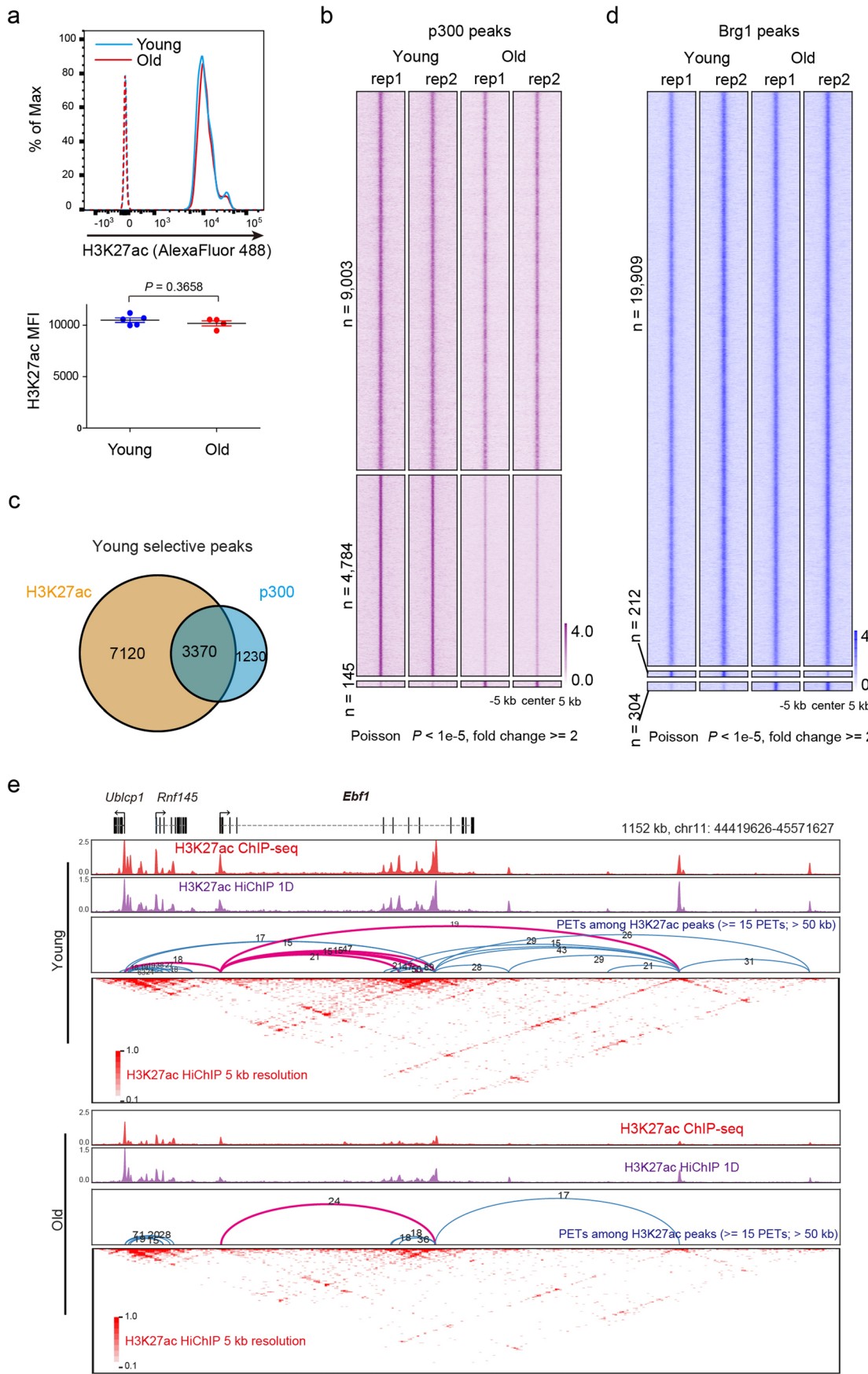

**Extended Data Fig. 5 | See next page for caption.**

**Extended Data Fig. 5 | H3K27ac mediated chromatin reorganization during aging.** (**a**) Measurement of H3K27ac levels in *Rag2*$^{-/-}$ young (blue) and old (red) pro-B cells via flow cytometry. The IgG isotype controls are shown in dotted lines. The dot plot on the bottom indicates the median fluorescence intensities (MFI) of H3K27ac levels in young and old pro-B cells. Data are presented as mean ± SEM. Unpaired two-sided *t* test was used, with the *P*-value indicated. (**b**) Age-selective p300 ChIP-Seq binding sites were identified in young and old pro-B cells.

The analysis and cutoffs used were consistent with those in Fig. 4a. (**c**) Overlaps of young selective H3K27ac and p300 peaks. (**d**) Age-selective Brg1 ChIP-Seq binding sites were identified in young and old pro-B cells. The analysis and cutoffs used were consistent with those in Fig. 4a. (**e**) H3K27ac HiChIP heatmaps of the *Ebf1* locus in young (upper) and old (lower) pro-B cells. The visualization elements are similar to that shown in Fig. 4h.

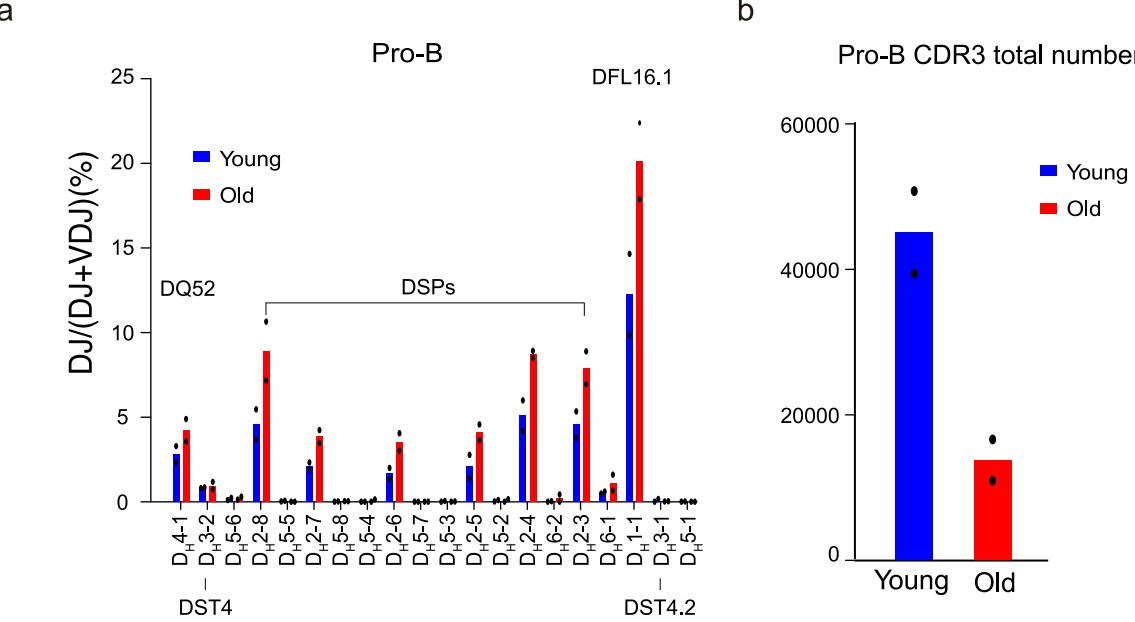

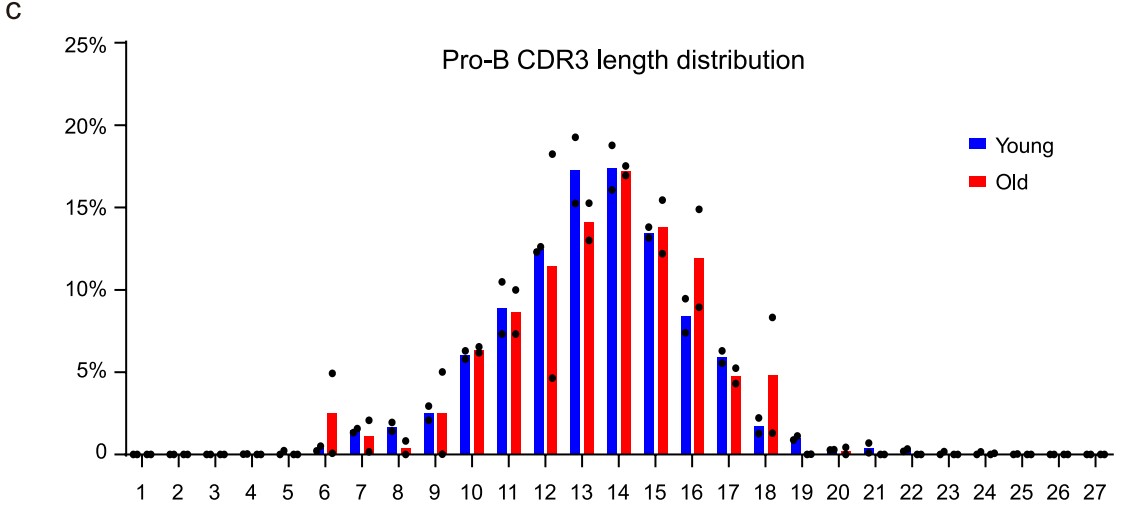

**Extended Data Fig. 6 | D$_H$ utilization and CDR3 length distribution are not affected by age. (a)** Average utilization of D$_H$ gene segments in V$_H$DJ$_H$ and DJ$_H$ joins in young and old C57BL/6J pro-B cells by VDJ-Seq (Hu et al.[49]) using J$_H$1 bait (n = 2). Clone names of D$_H$ gene segments are annotated. **(b)** Total CDR3 number in young and old C57BL/6J pro-B cells (n = 2). **(c)** CDR3 length distribution of productive V$_H$DJ$_H$ exons in young and old C57BL/6J pro-B cells by J$_H$1 bait (n = 2). Bar graph represents the mean of two replicate experiments, with each replicate shown as a dot.

# Reporting Summary

## Statistics

For all statistical analyses, confirm that the following items are present in the figure legend, table legend, main text, or Methods section.

| n/a | Confirmed | |
|---|---|---|
| ☐ | ☒ | The exact sample size (*n*) for each experimental group/condition, given as a discrete number and unit of measurement |
| ☐ | ☒ | A statement on whether measurements were taken from distinct samples or whether the same sample was measured repeatedly |
| ☐ | ☒ | The statistical test(s) used AND whether they are one- or two-sided<br>*Only common tests should be described solely by name; describe more complex techniques in the Methods section.* |
| ☐ | ☒ | A description of all covariates tested |
| ☐ | ☒ | A description of any assumptions or corrections, such as tests of normality and adjustment for multiple comparisons |
| ☐ | ☒ | A full description of the statistical parameters including central tendency (e.g. means) or other basic estimates (e.g. regression coefficient) AND variation (e.g. standard deviation) or associated estimates of uncertainty (e.g. confidence intervals) |
| ☐ | ☒ | For null hypothesis testing, the test statistic (e.g. *F*, *t*, *r*) with confidence intervals, effect sizes, degrees of freedom and *P* value noted<br>*Give P values as exact values whenever suitable.* |
| ☒ | ☐ | For Bayesian analysis, information on the choice of priors and Markov chain Monte Carlo settings |
| ☒ | ☐ | For hierarchical and complex designs, identification of the appropriate level for tests and full reporting of outcomes |
| ☐ | ☒ | Estimates of effect sizes (e.g. Cohen's *d*, Pearson's *r*), indicating how they were calculated |

*Our web collection on statistics for biologists contains articles on many of the points above.*

## Software and code

Policy information about availability of computer code

| Data collection | No softwares were used to collect public data. Sequencing data in FASTQ format were transferred by sequencing core with Google Drive or Globus (v3.1.5.637). |
|---|---|
| Data analysis | cLoops2 (v0.0.2): https://github.com/YaqiangCao/cLoops2<br>Bowtie2 (v2.3.5): http://bowtie-bio.sourceforge.net/bowtie2/index.shtml<br>STAR (v2.7.3a): https://github.com/alexdobin/STAR<br>Cufflinks (v2.2.1): http://cole-trapnell-lab.github.io/cufflinks/<br>HiCUP (v0.7.2): https://www.bioinformatics.babraham.ac.uk/projects/hicup<br>HiCExplorer3 (v3.6): https://hicexplorer.readthedocs.io/en/latest/<br>Juiicer (v1.11.04): https://github.com/aidenlab/juicer<br>deepTools (v3.3.0): https://deeptools.readthedocs.io/en/develop/index.html<br>Flowjo (10.6.0):https://www.flowjo.com/solutions/flowjo/downloads/previous-versions<br>GraphPad Prism (v.9.5.0):https://www.graphpad.com/updates/prism-900-release-notes |

For manuscripts utilizing custom algorithms or software that are central to the research but not yet described in published literature, software must be made available to editors and reviewers. We strongly encourage code deposition in a community repository (e.g. GitHub). See the Nature Portfolio guidelines for submitting code & software for further information.

## Data

Policy information about <u>availability of data</u>

All manuscripts must include a <u>data availability statement</u>. This statement should provide the following information, where applicable:
- Accession codes, unique identifiers, or web links for publicly available datasets
- A description of any restrictions on data availability
- For clinical datasets or third party data, please ensure that the statement adheres to our <u>policy</u>

Hi-C, Capture Hi-C, RNA-Seq, ChIP-seq, HiChIP and VDJ-Seq data generated by this study have been deposited to GEO with public accession of GSE214438. Both raw data and processed data with mouse reference genome mm10 are provided.

## Human research participants

Policy information about <u>studies involving human research participants and Sex and Gender in Research.</u>

| | |
|---|---|
| Reporting on sex and gender | N/A |
| Population characteristics | N/A |
| Recruitment | N/A |
| Ethics oversight | N/A |

Note that full information on the approval of the study protocol must also be provided in the manuscript.

# Field-specific reporting

Please select the one below that is the best fit for your research. If you are not sure, read the appropriate sections before making your selection.

☒ Life sciences　　　☐ Behavioural & social sciences　　　☐ Ecological, evolutionary & environmental sciences

For a reference copy of the document with all sections, see <u>nature.com/documents/nr-reporting-summary-flat.pdf</u>

# Life sciences study design

All studies must disclose on these points even when the disclosure is negative.

| | |
|---|---|
| Sample size | No statistical method was used to pre-determine the sample size. The sample sizes used in this study are indicated in the corresponding legends. Our sample sizes are similar to those reported in previous publications about aged mice. |
| Data exclusions | No data were excluded for analysis. |
| Replication | Biological replicates of each experiments were described in the corresponding figure legends. |
| Randomization | No randomization was needed for data collection. Some presentation examples were randomly selected. |
| Blinding | The sequencing data in this study were processed and analyzed with computer programs. The investigators were not blinded during data collection and analysis. No blinding was required in this study. |

# Reporting for specific materials, systems and methods

We require information from authors about some types of materials, experimental systems and methods used in many studies. Here, indicate whether each material, system or method listed is relevant to your study. If you are not sure if a list item applies to your research, read the appropriate section before selecting a response.

## Materials & experimental systems

| n/a | Involved in the study |
|-----|----------------------|
| ☐ | ☒ Antibodies |
| ☐ | ☒ Eukaryotic cell lines |
| ☒ | ☐ Palaeontology and archaeology |
| ☐ | ☒ Animals and other organisms |
| ☒ | ☐ Clinical data |
| ☒ | ☐ Dual use research of concern |

## Methods

| n/a | Involved in the study |
|-----|----------------------|
| ☐ | ☒ ChIP-seq |
| ☐ | ☒ Flow cytometry |
| ☒ | ☐ MRI-based neuroimaging |

# Antibodies

| | |
|---|---|
| Antibodies used | PE anti-CD19 (BioLegend, 6D5, Cat#115508)<br>BV421 anti-B220 (BioLegend, RA3-6B2, Cat # 103240)<br>FITC anti-IgM (BioLegend, RMM-1, Cat # 4065060)<br>PE anti-CD43 (BD Biosciences, S7, Cat #553271)<br>H3K27ac-AF488 conjugated (Cell Signaling Technology, Cat #15485S)<br>IgG isotype control-AF488 conjugated (Cell Signaling Technology, Cat #4340S)<br>anti-H3K27ac (Active Motif, Cat # 39133),<br>anti-H3K27me3 (Diagonode, Cat #C1541005)<br>anti-CTCF (Abcam, Cat# ab703030)<br>anti-Rad21 (Abcam, Cat# ab992)<br>anti-Brg1 (Abcam, Cat #ab110641)<br>anti-p300 (Cell Signaling Technology, Cat#57625S) |
| Validation | The validation information can be found in the merchandise websites. Each antibody was titrated before use and the dilution ratio is shown below:<br>Flow cytometry:<br>PE anti-CD19 (BioLegend, 6D5, Cat#115508, 1:100. Verified Reactivity: Mouse. Application: FC. https://www.biolegend.com/ja-jp/products/pe-anti-mouse-cd19-antibody-1530)<br>BV421 anti-B220 (BioLegend, RA3-6B2, Cat # 103240, 1:100. Verified Reactivity:Mouse, Human. Application: FC/IHC-F. https://www.biolegend.com/nl-be/products/brilliant-violet-421-anti-mouse-human-cd45r-b220-antibody-7158)<br>FITC anti-IgM (BioLegend, RMM-1, Cat # 406506; 1:50. Verified Reactivity:Mouse. Application: FC. https://www.biolegend.com/de-de/products/fitc-anti-mouse-igm-2334?GroupID=BLG3548)<br>PE anti-CD43 (BD Biosciences, S7, Cat #553271, 1:100. Verified Reactivity: Mouse. Application: FC. https://www.bdbiosciences.com/en-us/products/reagents/flow-cytometry-reagents/research-reagents/single-color-antibodies-ruo/pe-rat-anti-mouse-cd43.553271)<br>H3K27ac-AF488 conjugated (Cell Signaling Technology, Cat #15485S. 1:50. Verified Reactivity:Mouse, Human, Rat, Monkey. Application: FC/IHC-F. https://www.cellsignal.com/products/antibody-conjugates/acetyl-histone-h3-lys27-d5e4-xp-rabbit-mab-alexa-fluor-488-conjugate/15485)<br>IgG isotype control-AF488 conjugated (Cell Signaling Technology, Cat #4340S; 1:50. Verified Reactivity:Mouse, Human. Application: FC. https://www.cellsignal.com/products/antibody-conjugates/rabbit-igg-isotype-control-alexa-fluor-488-conjugate/4340)<br>ChIP-Seq and Hi-ChIP:<br>anti-H3K27ac (Active Motif, Cat # 39133; 1:500. Verified Reactivity:Mouse, Human, Budding Yeast. Application: FC/IHC-F/IP/ChIP/WB/ICC/CUT&RUN/CUT&Tag. https://www.activemotif.com/catalog/details/39133/histone-h3-acetyl-lys27-antibody-pab)<br>anti-H3K27me3 (Diagonode, Cat #C15410195; 1:500. Verified Reactivity:Mouse, Human, Drosophila, C. elegans, Daphnia, Arabidopsis, maize, tomato, poplar, silena latifolia, C. merolae, wide range expected. Application: WB/IHC-Fr/IP/CUT&Tag/ELISA/ChIP. https://www.diagenode.com/en/p/h3k27me3-polyclonal-antibody-premium-50-mg-27-ml)<br>anti-CTCF (Abcam, Cat# ab70303; 1:250. Verified Reactivity:Mouse, Human. Application: WB/IHC-Fr/IP/IHC-P. https://www.abcam.com/products/primary-antibodies/ctcf-antibody-ab70303.html)<br>anti-Rad21 (Abcam, Cat# ab992; 1:500. Verified Reactivity:Mouse, Human. Application: WB/IP. https://www.abcam.com/products/primary-antibodies/rad21-antibody-ab992.html)<br>anti-Brg1 (Abcam, Cat #ab110641; 1:200. Mouse, Human, Rad. Application: FC/IHC-P/IP/ICC/WB/IF/CUT&RUN. https://www.abcam.com/products/primary-antibodies/brg1-antibody-epncir111a-ab110641.html)<br>anti-p300 (Cell Signaling Technology, Cat#57625S; 1:333. Verified Reactivity:Mouse, Human, Rat, Monkey. Application: WB/IP/ChIP/CUT&RUN) |

# Eukaryotic cell lines

Policy information about cell lines and Sex and Gender in Research

| | |
|---|---|
| Cell line source(s) | The D345 cell line was generously provided by David G. Schatz of Yale University.<br>293T human embryonic kidney cells used for lentiviral expression was purchased from ATCC (Cat# CRL-3216).<br>The Platinum-E (Plat-E) cell line used for retroviral expression was purchased from Cell Biolabs (Cat# RV-101).<br>Rag2-/-Ebf1+/+Pax5+/+, Rag2-/-Ebf1+/-Pax5+/+ and Rag2-/-Ebf1+/−Pax5+/− pro-B cells were provided by Rudolf Grosschedl of Max Planck Institute of Immunobiology and Epigenetics. |
| Authentication | Cell lines were authenticated by morphology and genotyping. |
| Mycoplasma contamination | We did not test the mycoplasma contamination for these cells. |

| Commonly misidentified lines (See ICLAC register) | No commonly misidentified cell lines were used in this study. |
|---|---|

# Animals and other research organisms

Policy information about studies involving animals; ARRIVE guidelines recommended for reporting animal research, and Sex and Gender in Research

| Laboratory animals | Young (2-3 months) recombination activating gene 2-deficent (Rag2-/-) mice were purchased from Jackson Lab (Stock# 008449) and maintained to be old (23-24 months) in house by NIA Comparative Medicine Section (CMS). Young (2-3 months) and old (23-24 months) C57BL/6J mice were provided by National Institute on Aging (NIA) Aged Rodent Colonies. |
|---|---|
| Wild animals | No wild animals were involved in this study. |
| Reporting on sex | This study did not specifically focus on sex differences, and for each experiment, the sexes of both young and old mice were matched. |
| Field-collected samples | No field-collected samples were involved in this study. |
| Ethics oversight | All mouse experiments were performed under protocols approved by NIA Institutional Animal Care and Use Committees (338-LMBI-2022,2023,2024,2025) |

Note that full information on the approval of the study protocol must also be provided in the manuscript.

# ChIP-seq

## Data deposition

☒ Confirm that both raw and final processed data have been deposited in a public database such as GEO.

☒ Confirm that you have deposited or provided access to graph files (e.g. BED files) for the called peaks.

| Data access links *May remain private before publication.* | ChIP-seq data generated by this study have been deposited to GEO with public accession of GSE214438 |
|---|---|
| Files in database submission | Processed uniquely mapped high quality reads in BED or BEDPE files were uploaded to GEO for downloading and downstream analysis.  Raw FASTQ files are available corresponding to the GSM accession.<br>GSM6605809_ChIP-seq_Old_CTCF_rep1.bed.gz<br>GSM6605810_ChIP-seq_Old_CTCF_rep2.bed.gz<br>GSM6605811_ChIP-seq_Old_H3K27ac_rep1.bed.gz<br>GSM6605812_ChIP-seq_Old_H3K27ac_rep2.bed.gz<br>GSM6605813_ChIP-seq_Old_H3K27me3_rep1.bed.gz<br>GSM6605814_ChIP-seq_Old_H3K27me3_rep2.bed.gz<br>GSM6605815_ChIP-seq_Old_Rad21_rep1.bed.gz<br>GSM6605816_ChIP-seq_Old_Rad21_rep2.bed.gz<br>GSM6605817_ChIP-seq_Young_CTCF_rep1.bed.gz<br>GSM6605818_ChIP-seq_Young_CTCF_rep2.bed.gz<br>GSM6605819_ChIP-seq_Young_H3K27ac_rep1.bed.gz<br>GSM6605820_ChIP-seq_Young_H3K27ac_rep2.bed.gz<br>GSM6605821_ChIP-seq_Young_H3K27me3_rep1.bed.gz<br>GSM6605822_ChIP-seq_Young_H3K27me3_rep2.bed.gz<br>GSM6605823_ChIP-seq_Young_Rad21_rep1.bed.gz<br>GSM6605824_ChIP-seq_Young_Rad21_rep2.bed.gz<br>GSM7921744 ChIP-seq_Old_Brg1_rep1.bed.gz<br>GSM7921745 ChIP-seq_Old_Brg1_rep2.bed.gz<br>GSM7921746 ChIP-seq_Old_p300_rep1.bed.gz<br>GSM7921747 ChIP-seq_Old_p300_rep2.bed.gz<br>GSM7921748 ChIP-seq_Young_Brg1_rep1.bed.gz<br>GSM7921749 ChIP-seq_Young_Brg1_rep2.bed.gz<br>GSM7921750 ChIP-seq_Young_p300_rep1.bed.gz<br>GSM7921751 ChIP-seq_Young_p300_rep2.bed.gz |
| Genome browser session (e.g. UCSC) | WashU Epigenome Browser (http://epigenomegateway.wustl.edu/browser/) session bundle id: cdc07070-8e21-11ee-946c-99128820ea7d |

## Methodology

| Replicates | ChIP-seq in this study were performed with two biological replicates. |
|---|---|

| Sequencing depth | Single-end sequenced ChIP-seq samples:<br>sample TotalReads MappingRatio(%s) redundancy totalMappedReads uniqueReads<br>ChIP-seq_Old_CTCF_rep1 36152000 99.72 0.174670618 36003382 29714649<br>ChIP-seq_Old_CTCF_rep2 38403295 99.66 0.310768488 38219155 26341846<br>ChIP-seq_Old_H3K27ac_rep1 52414407 99.47 0.137581775 52051887 44890496<br>ChIP-seq_Old_H3K27ac_rep2 68395723 99.65 0.1378693 68048340 58666563<br>ChIP-seq_Old_H3K27me3_rep1 49286106 99.63 0.123891891 49012562 42940303<br>ChIP-seq_Old_H3K27me3_rep2 56635461 99.69 0.116566709 56357935 49788476<br>ChIP-seq_Old_Rad21_rep1 37594299 99.72 0.167368901 37439566 31173347<br>ChIP-seq_Old_Rad21_rep2 32376536 99.76 0.172744361 32258888 26686347<br>ChIP-seq_Young_CTCF_rep1 35139722 99.28 0.172994046 34843176 28815514<br>ChIP-seq_Young_CTCF_rep2 35386844 99.68 0.285618516 35224558 25163772<br>ChIP-seq_Young_H3K27ac_rep1 40703245 99.71 0.146787088 40534260 34584354<br>ChIP-seq_Young_H3K27ac_rep2 42092146 99.73 0.147615562 41922687 35734246<br>ChIP-seq_Young_H3K27me3_rep1 42840595 99.58 0.123732445 42591044 37321150<br>ChIP-seq_Young_H3K27me3_rep2 48840342 99.58 0.125633268 48555499 42455313<br>ChIP-seq_Young_Rad21_rep1 32632147 99.58 0.172958351 32461179 26846747<br>ChIP-seq_Young_Rad21_rep2 44822618 99.69 0.179776658 44629765 36606375<br><br>Paired-end sequenced ChIP-seq samples:<br>sample TotalReads MappingRatio(%s) redundancy totalMappedPETs uniquePETs<br>ChIP-seq_Old_Brg1_rep1 61783501 96.12 0.14960201 52364049 44530282<br>ChIP-seq_Old_Brg1_rep2 69004309 95.68 0.171523419 58220907 48234658<br>ChIP-seq_Old_p300_rep1 73935833 96.56 0.184713159 62098429 50628032<br>ChIP-seq_Old_p300_rep2 68687702 96.76 0.185678715 57910402 47157673<br>ChIP-seq_Young_Brg1_rep1 63872867 97.2 0.380600285 54827857 33960359<br>ChIP-seq_Young_Brg1_rep2 61142512 96.15 0.432857312 52074673 29533770<br>ChIP-seq_Young_p300_rep1 64604824 96.49 0.501838071 53618451 26710671<br>ChIP-seq_Young_p300_rep2 60880153 96.91 0.29125307 51002343 36147754 |
| Antibodies | anti-H3K27ac (Active Motif, Cat # 39133; 1:500)<br>anti-H3K27me3 (Diagonode, Cat #C1541005; 1:500)<br>anti-CTCF (Abcam, Cat# ab70303; 1:250)<br>anti-Rad21 (Abcam, Cat# ab992; 1:500)<br>anti-Brg1 (Abcam, Cat #ab110641; 1:200)<br>anti-p300 (Cell Signaling Technology, Cat#57625S; 1:333) |
| Peak calling parameters | ChIP-seq raw reads were mapped to the mouse reference genome mm10 by Bowtie2 (v2.3.5). Only non-redundant reads with MAPQ >=10 were used for the following analysis. BigWig tracks were generated by bamCoverage in deepTools (v3.3.0) with parameters of --ignoreDuplicates --minMappingQuality 10 --normalizeUsing CPM for visualization and quantification aggregation analysis. Peaks were called by cLoops2 with key parameters of -eps 150 for CTCF and Rad21, -eps 150,300 for H3K27ac and -eps 300,500 for H3K27me3, and -minPts was set to 20,50. |
| Data quality | CTCF motifs were checked for CTCF and RAD21 ChIP-seq data. Visualization through genome browser also confirmed strong peaks of all samples. We also measured the ratio of peaks (show as following) that have higher than 5-folds signal enrichment compared to the mean value of upstream and downstream flanking regions of 5-fold, 10-fold and 20-fold sizes.<br><br>5-fold ES 10-fold ES 20-fold ES<br>ChIP-seq_Old_CTCF_rep1 88.68563422 91.1182034 98.62997704<br>ChIP-seq_Old_CTCF_rep2 84.22470872 85.45398553 99.30708823<br>ChIP-seq_Old_H3K27ac_rep1 43.99613099 47.81677491 67.65925107<br>ChIP-seq_Old_H3K27ac_rep2 35.20046564 37.76834321 58.78042935<br>ChIP-seq_Old_H3K27me3_rep1 17.69499418 22.99185099 34.90104773<br>ChIP-seq_Old_H3K27me3_rep2 15.6460945 19.75650916 29.74927676<br>ChIP-seq_Old_Rad21_rep1 79.41933113 82.10951856 94.63800074<br>ChIP-seq_Old_Rad21_rep2 84.13290535 86.55764726 98.6900321<br>ChIP-seq_Young_CTCF_rep1 89.63096983 91.25931934 98.96434476<br>ChIP-seq_Young_CTCF_rep2 87.59759687 89.30508598 99.61082869<br>ChIP-seq_Young_H3K27ac_rep1 57.53865751 60.89664202 77.03549061<br>ChIP-seq_Young_H3K27ac_rep2 57.23651918 60.08344436 75.92384826<br>ChIP-seq_Young_H3K27me3_rep1 25.87425595 30.74776786 43.34077381<br>ChIP-seq_Young_H3K27me3_rep2 26.41387582 30.27954608 40.62182858<br>ChIP-seq_Young_Rad21_rep1 85.96245307 87.34918648 99.1864831<br>ChIP-seq_Young_Rad21_rep2 79.14553486 80.6296305 91.30577044<br>ChIP-seq_Old_Brg1_rep1 16.21013424 23.16287732 30.34841313<br>ChIP-seq_Old_Brg1_rep2 19.78908583 26.4595642 33.73356204<br>ChIP-seq_Old_p300_rep1 43.5253268 45.93545752 36.90767974<br>ChIP-seq_Old_p300_rep2 43.48846304 46.71490027 36.8596011<br>ChIP-seq_Young_Brg1_rep1 32.89585819 44.02590762 56.84336117<br>ChIP-seq_Young_Brg1_rep2 49.37111265 62.44644091 86.49619903<br>ChIP-seq_Young_p300_rep1 65.48011426 71.02834542 78.78488244<br>ChIP-seq_Young_p300_rep2 74.39927733 79.90966576 91.36404697 |

| Software | Bowtie2 (v2.3.5): http://bowtie-bio.sourceforge.net/bowtie2/index.shtml<br>deepTools (v3.3.0): https://deeptools.readthedocs.io/en/develop/index.html<br>cLoops2 (v0.0.2): https://github.com/YaqiangCao/cLoops2 |
|---|---|

# Flow Cytometry

## Plots

Confirm that:

☒ The axis labels state the marker and fluorochrome used (e.g. CD4-FITC).

☒ The axis scales are clearly visible. Include numbers along axes only for bottom left plot of group (a 'group' is an analysis of identical markers).

☒ All plots are contour plots with outliers or pseudocolor plots.

☒ A numerical value for number of cells or percentage (with statistics) is provided.

## Methodology

| Sample preparation | For Rag2-/- mice, total bone marrow was extracted from tibia and femurs, and erythrocytes are lysed. Pro-B cells were purified by combining positive selection using CD19+ selective beads (Stem Cell Technology, Cat#18954) and sorting by CD19 + and B220+ markers. For C57BL/6J mice, total bone marrow was extracted from tibia and femurs and erythrocytes are lysed. Cells were pre-purified using CD19+ selective beads (Stem Cell Technology, Cat#18954) and sorted with IgM-B220+CD43+ markers. |
|---|---|
| Instrument | BD FACSAria II |
| Software | BD FACSDiva and Flowjo_v10.6.0 software were used for data acquisition and analysis. |
| Cell population abundance | Cell purity was assessed after sorting, with more than 95% purity achieved after each sorting, ensuring suitability for subsequent experiments. |
| Gating strategy | For the Rag2-/- mice, cells were gated on FSC-A/SSC-A to select lymphocytes, FSC-H/FSC-A to gate on single cells. Then CD19 +B220+ cells were sorted for experiment.<br>For C57BL/6J mice, cells were gated on FSC-A/SSC-A to select lymphocytes, FSC-H/FSC-A to gate on single cells. SSC-A/IgM-FITC to gate on IgM- cells, then sort B220+CD43+ cells for experiment. |

☒ Tick this box to confirm that a figure exemplifying the gating strategy is provided in the Supplementary Information.