## [Peer Review File · Nature Cell Biology]

Peer Review Information

Journal: Nature Cell Biology

Manuscript Title: Three-dimensional chromatin re-organization regulates B cell development during aging

Corresponding author name(s): Ranjan Sen

Editorial Notes:

Reviewer Comments & Decisions:

Decision Letter, initial version:
--

Dear Dr. Sen,

Please first accept our sincere apology for the delay getting back to you with a decision due to unusual difficulties in retrieving reviewer comments.

Your manuscript, "Age-associated chromatin re-organization in progenitor B cells", has now been seen by 3 referees, who are experts in lymphocyte biology and chromatin structure (referee 1); chromatin structure, epigenetics (referee 2); and epigenetics in aging (referee 3). As you will see from their comments (attached below) they find this work of potential interest, but have raised substantial concerns, which in our view would need to be addressed with considerable revisions before we can consider publication in Nature Cell Biology.

Nature Cell Biology editors discuss the referee reports in detail within the editorial team, including the chief editor, to identify key referee points that should be addressed with priority, and requests that are overruled as being beyond the scope of the current study. To guide the scope of the revisions, I have listed these points below. We are committed to providing a fair and constructive peer-review process, so please feel free to contact me if you would like to discuss any of the referee comments further.

In particular, it would be essential to:

A) Provide additional data to support the proposed mechanism and functional outcome:

Reviewer 1

"...Could the authors please validate the change in nuclear location using 3D-FISH? What fraction of

aged pro-B cells show a change in nuclear location? Are the EBF1 alleles repositioning to the nuclear lamina? Alternatively, they might relocate to the peri-centromeric chromatin in aged cells as described for plasma cells. Could the authors please check this out?"

"...Could the authors also check for nuclear repositioning of FOXO1 in young versus aged pro-B cells using the data that presented here? One interesting question that comes to mind is whether forced EBF1 and/or FOXO1 expression in aged pro-B cells rescues an aged-specific gene program?..."

"How is H3K27Ac abundance altered in aged-versus young pro-B cells? Is E2A and P300 occupancy altered? Likewise, since occupancy by chromatin remodelers, including BRG1, is instructed by EBF1 it would be nice to examine for BRG1 occupancy in young and aged pro-B cells."

Reviewer 2

"Can overexpression of Ebf1 revert some of the aging-related chromatin reorganization features in old pro-B cells? This experiment would offer more compelling evidence supporting the role of reduced levels of Ebf1 in chromatin re-organization in old pro-B cells."

"Similarly, can knockdown of Wapl reverse the age associated chromatin organization back to that in the young pro-B cells?"

Reviewer 3

"...First, the focus on Ebf1 is not well justified in the study, and this finding is central to the broad conclusions about genomic 3D organization changes. See below for detailed suggestions. Second, the driving changes may be loss (mostly) and gain of enhancers and enhancer-promoter interactions, and this is not explicitly tested in the datasets. The major conclusion (186) of "large scale age-associated chromatin re-organization at a critical juncture of B cell development" may be an emergent computational property of smaller looping changes between enhancer-promoters due to enhancer loss. The authors should fully evaluate enhancers and enhancer-promoter interactions which may then necessitate tempering conclusions."

" Most changes are compartment A (euchromatin) to B (heterochromatin). Fig 1 and 2 very quickly focus on one gene, Ebf1, that switches from A to B. How was this gene chosen? What is the molecular function of Ebf1? What are the overall statistics of changes (amplitude of compartment changes, RNA seq, ChIP seq, etc) and where does this gene Ebf1 rank in these changes? What are the gene categories of changes and is this a key gene for B cells in comparison to other genes. It is not clear why CTCF would be increased rather than decreased during gene repression and no increase in K27me3 (perhaps its K27me2?)."

"Then further deep analysis of Ebf1 is shown in Fig 2 (95-115) and again in Fig 4, looking at chromatin and gene expression in Ebf1+/- cells. How were these cells derived? From individual humans? How are repeats carried out? Why would similar chromatin and gene expression changes be expected in Ebf1+/- compared to old B cells? The downstream focus on Ebf1 regulation of specific genes Mmnr1 and Minar1/Temed3 again seems very specific (frankly, rather cherry-picked). Is Ebf1 a transcription factor – does it ChIP to Mmnr1 or to Minar1/Temed3? (And later, to IgH?) Where do these genes rank in genes regulated by Ebf1? The conclusion (113-115) seems very strong about Ebf1's role without demonstration of robust genome-wide involvement."

"It is not clear (at least to this reviewer) that old and Ebf1+/- can be directly compared to deduce the

contribution of Ebf1 to old phenotype. What if heterozygous loss of Ebf1 in young cells leads to other changes due to developmental timing? For example, Wapl could be changed in PTMs. This conclusion should be carefully stated and discussed."

"...This is, to this reviewer, side-stepping the likely major change driving all the interactions changes, which is specific loss of enhancers in old cells leading to loss of enhancer-promoter interactions. The authors should evaluate enhancers genome-wide and enhancer-promoter interactions relative to gene expression to determine if enhancer alterations explain the loss of cell identity."

B) Improve the statistical analysis and data interpretation:

Reviewer 2

"In Figure 1C, the authors classified bins into A to B or B to A transitioning regions based on (0,0) coordinate. However, many regions called as switching compartments near the border could be artifacts. A more stringent cutoff should be determined. Much of consequential analysis is based on the A->B and B->A definitions in this figure."

"Similar concern to Figure 3A. The authors classified differential domains based on $\log_2(0)$. However, there is no threshold used to identify significantly differential domains in the MA plot. The authors may need to apply some cutoff such as FDR and \log_2FC . there should be a statistical significance cutoff. Also figure 3 legend title has an extra "changes"."

Reviewer 3

"Several hundred loci transitioned from compartment A to B and visa versa. What statistics were applied to reach this conclusion, and how do the authors know that these are bona fide age-related differences rather than technical (or animal to animal) variation? The majority of the called compartment changes in 1C appear along the X=Y line, with only a handful of genomic loci seeming to separate. With statistics and biological replicates taken into account, hundreds of called changes along X=Y line may prove to be insignificant, undercutting the author's claims of "large-scale conformational changes" (93)."

"Calling altered TADs is critical to the study. The authors describe one method in main text (lowered intraTAD interactions) but not the second method – but then use the intersection as their set of altered TADs. It is not clear to this reviewer how the two methods differ. As this intersection is key to the further focus on IgH (most reduced TAD), the authors should put both methods into main text."

C) All other referee concerns pertaining to strengthening existing data, providing controls, methodological details, clarifications and textual changes as applicable should also be addressed.

D) Finally please pay close attention to our guidelines on statistical and methodological reporting (listed below) as failure to do so may delay the reconsideration of the revised manuscript. In particular please provide:

- a Supplementary Table including all numerical source data in Excel format, with data for different

figures provided as different sheets within a single Excel file. The file should include source data giving rise to graphical representations and statistical descriptions in the paper and for all instances where the figures present representative experiments of multiple independent repeats, the source data of all repeats should be provided.

We would be happy to consider a revised manuscript that would satisfactorily address these points, unless a similar paper is published elsewhere, or is accepted for publication in Nature Cell Biology in the meantime.

- ensure that it conforms to our format instructions and publication policies (see below and www.nature.com/nature/authors/).
- provide a point-by-point rebuttal to the full referee reports verbatim, as provided at the end of this letter.
- provide the completed Editorial Policy Checklist (found here <https://www.nature.com/authors/policies/Policy.pdf>), and Reporting Summary (found here <https://www.nature.com/authors/policies/ReportingSummary.pdf>). This is essential for reconsideration of the manuscript and these documents will be available to editors and referees in the event of peer review. For more information see <http://www.nature.com/authors/policies/availability.html> or contact me.

Nature Cell Biology is committed to improving transparency in authorship. As part of our efforts in this direction, we are now requesting that all authors identified as 'corresponding author' on published papers create and link their Open Researcher and Contributor Identifier (ORCID) with their account on the Manuscript Tracking System (MTS), prior to acceptance. ORCID helps the scientific community achieve unambiguous attribution of all scholarly contributions. You can create and link your ORCID from the home page of the MTS by clicking on 'Modify my Springer Nature account'. For more information please visit www.springernature.com/orcid.

[REDACTED]

We would like to receive a revised submission within six months. We would be happy to consider a revision even after this timeframe, however if the resubmission deadline is missed and the paper is eventually published, the submission date will be the date when the revised manuscript was received.

We hope that you will find our referees' comments, and editorial guidance helpful. Please do not hesitate to contact me if there is anything you would like to discuss.

Best wishes,

Zhe Wang

Zhe Wang, PhD
Senior Editor
Nature Cell Biology

Tel: +44 (0) 207 843 4924
email: zhe.wang@nature.com

Reviewers' Comments:

Reviewer #1:

Remarks to the Author:

Age-associated chromatin re-organization in progenitor B cells

Whether and how aging affects genome topology remains a lingering enigma. In this interesting manuscript the authors examine how nuclear architecture is altered in young versus aged pro-B cells. The authors find that during the course of aging compartments display an enrichment for genomic interactions at the levels of compartmental domains. On the other hand, genomic interactions within topological domains declined in aged pro-B cells. As it relates to B-lineage specific gene programs, the authors found that the gene encoding Ebf1, a key B cell regulator, repositions from the euchromatic to the heterochromatic compartment in old pro-B cells. Depleting Ebf1 expression in young pro-B cells recapitulates some features of aged pro-B cells. Finally, the authors found that depletion of intra-TAD interactions that spans the IgH locus correlates well with altered VDJ recombination. Collectively, these data show how alterations in nuclear architecture during the course of aging modulates B cell development and somatic recombination.

This is an interesting paper. It provides new insight into molecular mechanism that underpin the aging process of B-lineage cells. This is particularly important as it relates to the decline in efficiency of vaccination and antibody response to pathogens in the elderly. The data are overall compelling and well presented. The paper is also well written. An important question come to mind however. Could the authors please validate the change in nuclear location using 3D-FISH? What fraction of aged pro-B cells show a change in nuclear location? Are the EBF1 alleles repositioning to the nuclear lamina? Alternatively, they might relocate to the peri-centromeric chromatin in aged cells as described for plasma cells. Could the authors please check this out?

Additional comments:

1. Since FOXO1 acts in a feed-forward loop with EBF1 it may very well be that this circuitry is altered in aged pro-B cells. From the data presented here it seems that indeed FOXO1 expression is substantially altered. Could the authors also check for nuclear repositioning of FOXO1 in young versus aged pro-B cells using the data that presented here? One interesting question that comes to mind is whether forced EBF1 and/or FOXO1 expression in aged pro-B cells rescues an aged-specific gene program? I realize that it might not be straightforward to perform this experiment since EBF1 and

FOXO1 dosage might be key.

2. How is H3K27Ac abundance altered in aged-versus young pro-B cells? Is E2A and P300 occupancy altered? Likewise, since occupancy by chromatin remodelers, including BRG1, is instructed by EBF1 it would be nice to examine for BRG1 occupancy in young and aged pro-B cells.

Reviewer #2:

Remarks to the Author:

In this manuscript, Ma and Cao et al report the finding of chromatin re-organization in progenitor B cells during aging. They performed Hi-C using pro-B cells from young and old recombination activating gene 2 (Rag2)-deficient mice. Their analysis revealed decreased intra-TAD interactions and increased inter TAD interactions in old pro-B cells. RNA-seq analysis showed that relatively more expressed genes within regions that transition from compartment A to B with age were associated with immune cell proliferation and differentiation. They found Ebf1, a key B cell developmental regulator, was one of the genes within the A to B transitioning regions. Their Hi-C and RNA-seq analyses using Ebf1 heterozygote Rag2-deficient pro-B cells showed that Ebf1^{+/-} pro-B cells shared similar features with old pro-B cells, suggesting that reduced Ebf1 in old pro-B cells might contribute the chromatin re-organization. Interestingly, they also observed increased expression of cohesin off-loader Wapl (by nearly 4 fold) in old pro-B cells, which correlates with reduced cohesin occupancy along TAD bodies, explaining the loss of chromatin interactions within TADs in old pro-B cells. Among TADs reduced in old pro-B cells, they found immunoglobulin heavy chain gene (IgH) locus was included. Their FISH and capture Hi-C analyses using young and old pro-B cells showed reduced intra-TAD interactions and interactions of the 3' IgH domain. Lastly, their VDJ sequencing on purified pro-B cell genomic DNA from young and old C57BL/6J mice showed reduced VH recombination, which may result from reduced interactions between 5' and 3' ends of the IgH locus in old pro-B cells.

Overall, this is a comprehensive analysis of age associated chromatin re-organization in the progenitor B cells, which highlight a potential role for chromatin reorganization in age associated changes in gene expression and pro-B cell function. The experiments were rigorously carried out, and the data analysis in general is clearly described. The study is of broad interest to the biomedical research field. On the other hand, there are considerable concerns as detailed below that the authors need to address before this manuscript can be deemed suitable for publication:

Main critiques:

1. Can overexpression of Ebf1 revert some of the aging-related chromatin reorganization features in old pro-B cells? This experiment would offer more compelling evidence supporting the role of reduced levels of Ebf1 in chromatin re-organization in old pro-B cells.
2. Similarly, can knockdown of Wapl reverse the age associated chromatin organization back to that in the young pro-B cells?
3. Authors point out a reduction in TAD strength in age at 1477 TADs (Figure 3A). They note that these TADs have lower Rad21 binding and hypothesize it is due to increased Wapl expression (Figure 3C,D). Authors should comment on the potential causes of increased Wapl expression in old pro-B cells and whether there is a global reduction of Rad21. If not, they could explore why some TADs are losing Rad21 while others are able to maintain it in aging.
4. In Figure 1C, the authors classified bins into A to B or B to A transitioning regions based on (0,0)

coordinate. However, many regions called as switching compartments near the border could be artifacts. A more stringent cutoff should be determined. Much of consequential analysis is based on the A->B and B->A definitions in this figure.

5. Similar concern to Figure 3A. The authors classified differential domains based on $\log_2(0)$. However, there is no threshold used to identify significantly differential domains in the MA plot. The authors may need to apply some cutoff such as FDR and \log_2FC . there should be a statistical significance cutoff. Also figure 3 legend title has an extra "changes".

Additional concerns:

6. For some people who do not know the function of Rag2, I suggest including a sentence to explain how Rag2 deficiency affects VDJ recombination in this strain, and especially, its association with key B-cell development transcription factor such as Ebf1.

7. The authors used Rag2-deficient mice. RAG1 and RAG2 together initiate recombination and IgH gene assembly by introducing double-strand DNA breaks in developing lymphocytes. The authors did not explain the possible impact of RAG1 in this study.

8. In line 44-52, the authors mentioned that very little is known about chromatin re-organization during aging. However, there is some age-related 3D chromatin reorganization study done in, for example, muscle stem cell aging. I suggest that the authors either discuss openly the similarities and differences of their study with previous work, or modify their statement limited to immune cells or B cell development.

9. In Figure 1B, there are some regions with increased short-range contacts (red color). Can the authors elaborate more on what those regions are?

10. In Figure 1E, the enriched GO terms for genes within compartments switching from A to B in aging seem to be the same as those enriched in immune cell/B cell gene expression generally, which could be inherent to those genes within compartment A of pro-B cells. It may be necessary to perform this enrichment analysis with a background set of only genes in compartment A of the young pro-B cells rather than all genes to avoid any bias in sampling.

11. The enriched GO terms for regions switching from compartment B to compartment A in aging (Fig S1.D) are far more significant (despite having fewer regions total). It would be great if authors could comment on them as well.

12. In line 213-215, the author proposed the shift from B cell lineage to myeloid and erythroid differentiation. However, the authors did not measure the proportion of cell populations or did not provide direct evidence related to this in this study. I suggest referring previous studies that showed shift from the lymphoid to the myeloid lineage of in aged hematopoietic stem cells to support their statement.

13. Scale bar is not defined in figures or not explained in captions for Figure 1B, 2B, 2D, 3G, 4A, and 4C.

14. Any statistical test for Figure S4?

15. Authors should further discuss potential causes for the compartmental changes seen in aging pro-B cells and leading to important changes in B cell developmental regulators.

Reviewer #3:

Remarks to the Author:

The question of the driving role of 3D changes in development and aging remains an outstanding question. This manuscript describes an extensive study of three dimensional (3D) interactions in

young vs old pro-B cells. The authors find many 3D changes. They characterize two local changes (Ebf1 and IgH). They conclude that (abstract) "3D chromatin reorganization is a major driver" of aging in B cells.

The study documents certain changes in 3D organization with age, and this is convincing. There are two major problems (and other issues) with the study that must be addressed before further consideration of the manuscript for publication. First, the focus on Ebf1 is not well justified in the study, and this finding is central to the broad conclusions about genomic 3D organization changes. See below for detailed suggestions. Second, the driving changes may be loss (mostly) and gain of enhancers and enhancer-promoter interactions, and this is not explicitly tested in the datasets. The major conclusion (186) of "large scale age-associated chromatin re-organization at a critical juncture of B cell development" may be an emergent computational property of smaller looping changes between enhancer-promoters due to enhancer loss. The authors should fully evaluate enhancers and enhancer-promoter interactions which may then necessitate tempering conclusions.

Specific points

1.64-69. Technical questions with study: How homogeneous is the pro-B cell population (referred to as "relatively homogeneous")? If not homogeneous, changes observed could be due to shifts in cell population. Also, how many samples were used and were individual samples mixed? If mixed, changes could be due to individual mouse variation.

2.73-75. Several hundred loci transitioned from compartment A to B and visa versa. What statistics were applied to reach this conclusion, and how do the authors know that these are bona fide age-related differences rather than technical (or animal to animal) variation? The majority of the called compartment changes in 1C appear along the X=Y line, with only a handful of genomic loci seeming to separate. With statistics and biological replicates taken into account, hundreds of called changes along X=Y line may prove to be insignificant, undercutting the author's claims of "large-scale conformational changes" (93).

3.84-94. Most changes are compartment A (euchromatin) to B (heterochromatin). Fig 1 and 2 very quickly focus on one gene, Ebf1, that switches from A to B. How was this gene chosen? What is the molecular function of Ebf1? What are the overall statistics of changes (amplitude of compartment changes, RNA seq, ChIP seq, etc) and where does this gene Ebf1 rank in these changes? What are the gene categories of changes and is this a key gene for B cells in comparison to other genes. It is not clear why CTCF would be increased rather than decreased during gene repression and no increase in K27me3 (perhaps its K27me2?).

4. Then further deep analysis of Ebf1 is shown in Fig 2 (95-115) and again in Fig 4, looking at chromatin and gene expression in Ebf1+/- cells. How were these cells derived? From individual humans? How are repeats carried out? Why would similar chromatin and gene expression changes be expected in Ebf1+/- compared to old B cells? The downstream focus on Ebf1 regulation of specific genes Mmrn1 and Minar1/Temed3 again seems very specific (frankly, rather cherry-picked). Is Ebf1 a transcription factor – does it ChIP to Mmrn1 or to Minar1/Temed3? (And later, to IgH?) Where do these genes rank in genes regulated by Ebf1? The conclusion (113-115) seems very strong about Ebf1's role without demonstration of robust genome-wide involvement.

5.128. It is not clear (at least to this reviewer) that old and Ebf1+/- can be directly compared to deduce the contribution of Ebf1 to old phenotype. What if heterozygous loss of Ebf1 in young cells leads to other changes due to developmental timing? For example, Wapl could be changed in PTMs. This conclusion should be carefully stated and discussed.

6.120-121. Calling altered TADs is critical to the study. The authors describe one method in main text (lowered intraTAD interactions) but not the second method – but then use the intersection as their set

of altered TADs. It is not clear to this reviewer how the two methods differ. As this intersection is key to the further focus on IgH (most reduced TAD), the authors should put both methods into main text. 7. Fig 3B and 3C. The authors should show heatmaps (ie using Deeptools) under the metaplots or another indication demonstrating how many changing TADs have differential histone modifications, CTCF, or Rad21. The signal differences observed in these metaplots could be due to small differences across many TADs or large differences in a few TADs, which informs how these results are interpreted. 8. 152-153. It appears that most of the documented changes that are occurring locally to IgH and Ebf1 loci are loss of K27ac. The authors interpret this rather broadly as "reduced interactions locus-wide and widespread loss of H3K27ac and Rad21 mark the IgH TAD in old pro-B cells". This is, to this reviewer, side-stepping the likely major change driving all the interactions changes, which is specific loss of enhancers in old cells leading to loss of enhancer-promoter interactions. The authors should evaluate enhancers genome-wide and enhancer-promoter interactions relative to gene expression to determine if enhancer alterations explain the loss of cell identity. 9. Figure 4. Can the authors explain the "virtual 4D map"- what is this and its interpretation.

ABSTRACT AND MAIN TEXT – please follow the guidelines that are specific to the format of your manuscript, as listed in our Guide to Authors (http://www.nature.com/ncb/pdf/ncb_gta.pdf) Briefly,

Nature Cell Biology Articles, Resources and Technical Reports have 3500 words, including a 150 word abstract, and the main text is subdivided in Introduction, Results, and Discussion sections. Nature Cell Biology Letters have up to 2500 words, including a 180 word introductory paragraph (abstract), and the text is not subdivided in sections.

Methods should be written concisely, but should contain all elements necessary to allow interpretation and replication of the results. As a guideline, Methods sections typically do not exceed 3,000 words. The Methods should be divided into subsections listing reagents and techniques. When citing previous methods, accurate references should be provided and any alterations should be noted. Information must be provided about: antibody dilutions, company names, catalogue numbers and clone numbers for monoclonal antibodies; sequences of RNAi and cDNA probes/primers or company names and catalogue numbers if reagents are commercial; cell line names, sources and information on cell line identity and authentication. Animal studies and experiments involving human subjects must be reported in detail, identifying the committees approving the protocols. For studies involving human subjects/samples, a statement must be included confirming that informed consent was obtained. Statistical analyses and information on the reproducibility of experimental results should be provided in a section titled "Statistics and Reproducibility".

All Nature Cell Biology manuscripts submitted on or after March 21 2016 must include a Data

availability statement at the end of the Methods section. For Springer Nature policies on data availability see <http://www.nature.com/authors/policies/availability.html>; for more information on this particular policy see <http://www.nature.com/authors/policies/data/data-availability-statements-data-citations.pdf>. The Data availability statement should include:

- Accession codes for primary datasets (generated during the study under consideration and designated as "primary accessions") and secondary datasets (published datasets reanalysed during the study under consideration, designated as "referenced accessions"). For primary accessions data should be made public to coincide with publication of the manuscript. A list of data types for which submission to community-endorsed public repositories is mandated (including sequence, structure, microarray, deep sequencing data) can be found here <http://www.nature.com/authors/policies/availability.html#data>.
- Unique identifiers (accession codes, DOIs or other unique persistent identifier) and hyperlinks for datasets deposited in an approved repository, but for which data deposition is not mandated (see here for details <http://www.nature.com/sdata/data-policies/repositories>).
- At a minimum, please include a statement confirming that all relevant data are available from the authors, and/or are included with the manuscript (e.g. as source data or supplementary information), listing which data are included (e.g. by figure panels and data types) and mentioning any restrictions on availability.
- If a dataset has a Digital Object Identifier (DOI) as its unique identifier, we strongly encourage including this in the Reference list and citing the dataset in the Methods.

We recommend that you upload the step-by-step protocols used in this manuscript to the Protocol Exchange. More details can found at www.nature.com/protocolexchange/about.

All imaging data should be accompanied by scale bars, which should be defined in the legend. Cropped images of gels/blots are acceptable, but need to be accompanied by size markers, and to retain visible background signal within the linear range (i.e. should not be saturated). The boundaries of panels with low background have to be demarked with black lines. Splicing of panels should only be considered if unavoidable, and must be clearly marked on the figure, and noted in the legend with a statement on whether the samples were obtained and processed simultaneously. Quantitative comparisons between samples on different gels/blots are discouraged; if this is unavoidable, it should only be performed for samples derived from the same experiment with gels/blots were processed in parallel, which needs to be stated in the legend.

TABLES – main tables should be provided as individual Word files, together with a brief title and

legend. For supplementary tables see below.

The total number of Supplementary Figures (not including the “unprocessed scans” Supplementary Figure) should not exceed the number of main display items (figures and/or tables (see our Guide to Authors and March 2012 editorial <http://www.nature.com/ncb/authors/submit/index.html#suppinfo>; <http://www.nature.com/ncb/journal/v14/n3/index.html#ed>). No restrictions apply to Supplementary Tables or Videos, but we advise authors to be selective in including supplemental data.

GUIDELINES FOR EXPERIMENTAL AND STATISTICAL REPORTING

REPORTING REQUIREMENTS – To improve the quality of methods and statistics reporting in our papers we have recently revised the reporting checklist we introduced in 2013. We are now asking all life sciences authors to complete two items: an Editorial Policy Checklist (found here <https://www.nature.com/authors/policies/Policy.pdf>) that verifies compliance with all required editorial policies and a reporting summary (found here <https://www.nature.com/authors/policies/ReportingSummary.pdf>) that collects information on experimental design and reagents. These documents are available to referees to aid the evaluation of the manuscript. Please note that these forms are dynamic ‘smart pdfs’ and must therefore be

downloaded and completed in Adobe Reader. We will then flatten them for ease of use by the reviewers. If you would like to reference the guidance text as you complete the template, please access these flattened versions at <http://www.nature.com/authors/policies/availability.html>.

Author Rebuttal to Initial comments

December 14, 2023

Dr. Zhe Wang

Senior Editor

Nature Cell Biology

One New York Plaza, Suite 4600

New York, NY 10004-1562

Dear Dr. Wang:

Thank you for coordinating review of our manuscript by Ma et al. (NCB-LE49860). We were encouraged by the positive responses of reviewers and *NCB* editors and have taken pains (and time) to address all comments that were raised during review. We are now submitting a revised manuscript that includes new experimental data (Fig.1h and i, Fig 3b, g and i, Fig 4, Fig.5f and g, Extended Data Fig. 1e and f, Extended Data Fig.2, Extended Data Fig.4 and Extended Data Fig.5), improved statistical analyses (regarding Fig.1c-e and Fig. 3a) and conceptual clarifications requested by reviewers. We believe that the manuscript and the underlying message are greatly enhanced by these changes and hope you will find it suitable for publication in *Nature Cell Biology*. Below we provide point-by-point responses first to editorial prioritization of key issues and, second, to comments from each reviewer. Editor and reviewer comments are underlined and noted within quotes, and all substantive changes in the text are marked by red font for easy evaluation.

EDITORIAL PRIORITIES

A ‘Provide additional data to support the proposed mechanism and functional outcome.’

Reviewer 1

“...Could the authors please validate the change in nuclear location using 3D-FISH? What fraction of aged pro-B cells show a change in nuclear location? Are the EBF1 alleles repositioning to the nuclear lamina? Alternatively, they might relocate to the peri-centromeric chromatin in aged cells as described for plasma cells. Could the authors please check this out?”

This comment pertains to our observation that the *Ebf1* locus shifts from euchromatic compartment A to heterochromatic compartment B in old pro-B cells. The reviewer is interested in knowing whether compartment changes are accompanied by changes in nuclear location. To address this, we carried out 3D-FISH with *Ebf1* and *Foxo1* probes in young and old pro-B cells and non-B lineage bone marrow cells from RAG2-deficient mice. Cells for each experiment were pooled from 2-3 mice. In new Figure 1h and i, we show that the *Ebf1* locus is located closer to the nuclear periphery in old pro-B cells compared to

young pro-B cells, with the average displacement lying between that seen in non-B cells (closest to the periphery) and young pro-B cells (furthest from the periphery) (discussed on page 5).

“...Could the authors check for nuclear repositioning of FOXO1 in young versus aged pro-B cells using the data that presented here? One interesting question that comes to mind is whether forced EBF1 and/or FOXO1 expression in aged pro-B cells rescues an aged-specific gene program?”

We carried out FISH with *Foxo1* probes in young and old RAG2-deficient pro-B cells and found no statistical difference in location relative to either the nuclear periphery or centromeric heterochromatin (new Figure 1h and Extended Data Fig. 1e, page 5). The *Foxo1* locus was located squarely within compartment A in both young and old pro-B cells (new Extended Data Fig. 1f), suggesting that it was down-regulated by a mechanism distinct from that of *Ebf1* (discussed on page 5).

“How is H3K27ac abundance altered in aged-versus young pro-B cells? Is E2A and P300 occupancy altered? Likewise, since occupancy by chromatin remodelers, including BRG1, is instructed by EBF1 it would be nice to examine for BRG1 occupancy in young and old pro-B cells.”

In response, we carried out ChIP-Seq with anti-p300 and anti-Brg1 antibodies, and re-evaluated our H3K27ac ChIP-Seq to obtain a global view of changes that occur during aging. We observed no difference in levels of H3K27ac in young and old pro-B cells (new Extended Data Fig. 5a). However, approximately 27% H3K27ac peaks were reduced in intensity in old pro-B cells; conversely, only 3% H3K27ac peaks were increased in old pro-B cells (new Figure 4a, b; discussed on page 8).

Anti-p300 ChIP-seq revealed about 14,000 binding sites genome-wide in fresh *ex vivo* primary pro-B cells. Approximately 30% of sites lost p300 binding in old pro-B cells whereas only 1% sites were gained (new Extended Data Fig. 5b). Importantly, 74% sites that lost p300 were also reduced for H3K27ac (new Extended Data Fig. 5c). In sharp contrast, very few of approximately 20,000 Brg1-bound sites changed with age (new Extended Data Fig. 5d). These observations are discussed on page 8 of the revised manuscript.

Reviewer 2

“Can over-expression of Ebf1 revert some of the aging-related chromatin reorganization features in old pro-B cells? This experiment would offer more compelling evidence supporting the role of reduced levels of Ebf1 in chromatin re-organization in old pro-B cells. Similarly, can knockdown of Wapl reverse the age associated chromatin organization back to that in the young pro-B cell?”

The reviewer correctly noted that rejuvenation of old pro-B cells by Ebf1 over-expression or knock-down of Wapl would provide “more compelling evidence supporting the role (of these proteins) in chromatin re-organization in old pro-B cells”. We spent most of the revision period addressing this gold standard. To obtain meaningful results, this involves manipulating Ebf1 or Wapl levels in old pro-B cells without changing their aged phenotype. We found that *ex vivo* culture of primary pro-B cells, that is essential for altering Ebf1 or Wapl levels, changed the transcriptomes of old pro-B cells (new Extended Data Fig. 2c, e). Though relatively few age-associated genes remained differentially expressed after culture of old pro-B cells, we found that Ebf1 transduction changed the expression profile of these genes towards that of young pro-B cells (new Extended Data Fig. 2d, f). These observations support the conclusion that Ebf1 down-regulation during aging contributes to differential gene expression in old pro-B cells. Additionally, our observations strongly reinforce the notion that physiological aging is an organismic property that is virtually impossible to study in tissue culture.

We limited the Ebf1 transduction studies to gene expression analyses rather than carry out a full chromatin study because results from Ebf1 heterozygote pro-B cells indicated that reduced Ebf1 levels are not at the apex of large-scale chromatin re-organization. Rather, we propose that chromatin re-organization leads to Ebf1 down-regulation and consequent changes in the pro-B cell transcriptome (discussed on pages 5, 6).

Wapl knockdown was not feasible in old primary pro-B cells. We attempted this via a) lentiviral transduction of shRNAs and b) siRNA transfection into old pro-B cells. As noted above, *ex vivo* culture of pro-B cells altered the aging phenotype sufficiently that we did not obtain interpretable results. Instead, we provide two lines of evidence in the revised manuscript that Wapl impacts pro-B cell chromatin structure. First, we draw attention to the published study in which Wapl expression was increased in pro-B cells by mutating a repressive element in the gene promoter. Hi-C analyses showed reduced interactions in the TAD range (300 kb - 10 Mb) and increased interactions in the compartment range (> 10 Mb). This pattern was similar to what we observed in old pro-B cells (new Figure 1a). We had previously quoted this paper¹ (ref #41 in both old and new versions). Second, we increased Wapl expression in a pro-B cell line (new Extended Data Fig. 4). Hi-C analyses of Wapl expressing cells showed lower genomic contacts in the 50-300 kb range, typical of intra-TAD interactions, and higher genomic contacts in > 10 Mb range, which are compartmental interactions, as seen in old pro-B cells (new

Extended Data Fig. 4c). The *Igh* locus was amongst those that were most affected by Wapl expression (new Extended Data Fig 4d, e). Finally, RNA-Seq analyses showed that Wapl expression skewed the transcriptomic profile towards myeloid lineage cells, like what we had observed in old pro-B cells and *Ebf1*^{+/-} pro-B cells (new Extended Data Fig. 4f). Our observations demonstrate that altering Wapl levels recapitulates many chromatin and gene expression features of old pro-B cells. These results are discussed on page 7 of the revised manuscript.

In summary, we made good faith effort to alter Ebf1 and Wapl levels in primary old pro-B cells. We found that *ex vivo* culture altered the phenotype of old pro-B cells, making it impossible to do the perfect experiment. Instead, we manipulated Ebf1 and Wapl levels in pro-B cells and demonstrated that the changes induced are completely consistent with our hypothesis regarding the effects of age on B cell development.

Reviewer 3

“...First, the focus on Ebf1 is not well justified in the study, and this finding is central to the broad conclusions about genomic 3D organization changes. See below for detailed suggestions.”

We have included additional justification for focusing on Ebf1 (pages 4, 5). Briefly, the *Ebf1* locus is the top region (out of 157) that transitioned from A to B compartment in old pro-B cells. Not only is *Ebf1* the most strongly affected loci, but the gene also caught our attention because it is a well-known regulator of B cell development. Additionally, *Ebf1* has been previously shown to undergo nuclear re-localization in pro-B cells, a change that could well be reflected in altered compartmentalization.

“Second, the driving changes may be loss (mostly) and gain of enhancers and enhancer-promoter interactions, and this is not explicitly tested in the datasets. The major conclusion (186) of “large scale age-associated chromatin re-organization at a critical juncture of B cell development” may be an emergent computational property of smaller looping changes between enhancer-promoters due to enhancer loss. The authors should fully evaluate enhancers and enhancer-promoter interactions which may then necessitate tempering conclusions.the driving changes may be loss (mostly) and gain of enhancers and enhancer-promoter interactions, and this is not explicitly tested in the datasets.”

The reviewer correctly noted that we had not thoroughly investigated enhancer-promoter interactions in the previous version of the manuscript. In the revised manuscript we provide such analyses (new Figure 4). We carried out H3K27ac ChIP-Seq as a means of identifying active enhancers and H3K27ac Hi-ChIP to evaluate interactions between enhancers and promoters. We found that approximately 27% of H3K27ac-marked sites were reduced in old pro-B cells (new Figure 4a); most of these were located at enhancers. By contrast, only 3% H3K27ac-marked sites were increased in old pro-B cells, again mostly at enhancers (new Figure 4a). Genes annotated to H3K27ac peaks that reduced with age were expressed at significantly lower levels in old pro-B cells and vice versa (new Figure 4b-d). Hi-ChIP analyses showed that enhancers and promoters with reduced H3K27ac were involved in weaker interactions in old pro-B cells (new Figure 4e-h). These observations are exactly in accordance with the predictions made by the reviewer (discussed on pages 8, 9).

“Most changes are compartment A (euchromatin) to compartment B (heterochromatin). Fig. 1 and 2 quickly focus on one gene, Ebf1, that switched from A to B. How was this gene chosen? What is the molecular function of Ebf1? What are the overall statistics of changes (amplitude of compartment changes, RNA seq, ChIP seq, etc) and where does this gene Ebf1 rank in these changes? What are the gene categories of changes and is this a key gene for B cells in comparison to other genes?”

As described in the first response to Reviewer 3, we provide additional justification for our focus on *Ebf1* (pages 4, 5). The statistics of changes are provided in new Figure 1c legend and the Source Data.

“Then further deep analysis of Ebf1 is shown in Fig 2 (95-115) and again in Fig 4, looking at chromatin and gene expression in Ebf1^{+/-} cells. How were these cells derived? From individual humans? How are repeats carried out? Why would similar chromatin and gene expression changes be expected in Ebf1^{+/-} compared to old B cells? The downstream focus on Ebf1 regulation of specific genes Mmnr1 and Minar1/Temed3 again seems very specific (frankly, rather cherry-picked). Is Ebf1 a transcription factor – does it ChIP to Mmnr1 or to Minar1/Temed3? (And later, to IgH?) Where do these genes rank in genes regulated by Ebf1? The conclusion (113-115) seems very strong about Ebf1’s role without demonstration of robust genome-wide involvement.”

The rationale for detailed analysis of *Ebf1*^{+/-} pro-B cells (which are derived from a mouse fetal liver, with two biological replicates) was the following. Our attention had been drawn to Ebf1 for reasons summarized in the first response to Reviewer 3. However, despite the known importance of Ebf1 for B cell development, our observations did not reveal the extent to which changes noted in old pro-B cells

were caused by alterations in Ebf1 expression. We hypothesized that alterations in Ebf1 were likely to be responsible for only a subset of changes seen in old pro-B cells. One way to test this was to examine chromatin state and gene expression in pro-B cells in which Ebf1 levels were lowered by a means other than age. We chose to do this using pro-B cells that lacked one of two *Ebf1* alleles. These cells had been previously shown to express half the level of Ebf1 compared to pro-B cells with two *Ebf1* alleles. We reasoned that *Ebf1*^{+/-} pro-B cells represented pro-B cells with reduced Ebf1 expression but none of the other (Ebf1-independent) characteristics of old pro-B cells.

As explained more extensively in the revised manuscript (pages 5, 6), we found that Ebf1 down-regulation did not recapitulate increased compartment level interactions found in old pro-B cells. We interpret this to mean that Ebf1 does not induce compartment changes at the global level as observed in old pro-B cells. However, some loci changed comparably in *Ebf1*^{+/-} and old pro-B cells. One such locus is *Minar1/Tmed3* (Figure 2d). This locus has been previously shown to bind Ebf1 in pro-B cells². We infer that changes at this locus may be the result of reduced Ebf1 expression in old pro-B cells and *Ebf1*^{+/-} cells. Remarkably, Ebf1 heterozygosity led to changes in gene expression that had many features of old pro-B cells, especially genes relevant for B cell development (Figure 2e, f). We conclude that compartmental changes in old pro-B cells reduce Ebf1 expression that, in turn, alter gene expression patterns in keeping with its role as a transcription factor. Identification of the initiator of compartment level alterations during aging remains a goal for the future.

“It is not clear (at least to this reviewer) that old and Ebf1+/- pro-B cells can be directly compared to deduce the contribution of Ebf1 to old phenotype. What if heterozygous loss of Ebf1 in young cells leads to other changes due to developmental timing? For example, Wapl could be changed in PTMs. This conclusion should be carefully stated and discussed.”

We appreciate the point made by the reviewer. In the revised version we have elaborated our reasons for choosing this experimental approach, while cautioning the reader about the possible developmental caveat (pages 5, 6).

“...This is, to this reviewer, side-stepping the likely major change driving all the interaction changes, which is specific loss of enhancers in old cells leading to loss of enhancer-promoter interactions. The authors should evaluate enhancers genome-wide and enhancer-promoter interactions relative to gene expression to determine if enhancer alterations explain the loss of cell identity”.

We applaud the reviewer's perceptiveness about the possible role of enhancer-promoter interactions in driving interaction changes in old pro-B cells. As discussed in the second response point to Reviewer 3, we provide new experimental data and analysis (new Figure 4) investigating the role of enhancers and enhancer-promoter interactions in young and old pro-B cells. By combining H3K27ac ChIP-Seq (to score for differences between young and old pro-B cells) and H3K27ac Hi-ChIP (to score for interactions between H3K27ac-marked enhancers and promoters) we found that loss of enhancer-promoter interactions greatly exceeded gain of such interactions in old pro-B cells (discussed on page 8).

B 'Improve the statistical analysis and data interpretation'

Reviewer 2

"In Figure 1C, the authors classified bins into A to B or B to A transitioning regions based on (0,0) coordinate. However, many regions called as switching compartments near the border could be artifacts. A more stringent cutoff should be determined. Much of consequential analysis is based on the A->B and B->A definitions in this figure."

Both reviewers 2 and 3 make this point, which is both relevant and important. Our analysis of the Hi-C data began in early 2020, at which time compartment analysis relied primarily on the sign of PCA analysis.³⁻⁵ Furthermore, comparisons between compartments were limited to flips/switches.^{6,7} This was also our strategy for Figure 1c (similar to, for example, Figures 1a and d in ⁷, and Figure S8 in ⁶). This kind of flip/switch analysis had also been used in more recent studies⁸ close to the time when we submitted our manuscript. However, we completely understand the need to stay state-of-the-art to publish in *Nature* and have therefore addressed reviewers' concerns as follows. We implemented a two-pass Mahalanobis distance calculation (an effective measurement for detecting outliers based on the distribution pattern of data points), as well as a Chi-square test, which is the same strategy used in a recent dcHi-C analysis⁸. As expected, application of these criteria reduced the number of significant compartment flips/switches compared to our previous analysis; in addition, we also detected some changes within the same compartments (new Figure 1c). Importantly, loci that we had focused on as undergoing compartment switching in the original submission, such as *Ebf1*, remained highly statistically significant even after applying the criteria noted above.

The Mahalanobis distance (MD) was calculated as $MD = \text{diag}((X - X_c) \times C^{-1} \times (X - X_c)^T)$, where X is the matrix for PC1 obtained from the Hi-C compartment analysis. Each row represents a bin,

and each column represents a sample (young or old). C^{-1} is the inverse covariance matrix of X , X_c is vector of row-wise mean of X , and *diag* is the function used to extract a diagonal array from a matrix. We performed the Chi-Square test with the MD and a P-value cutoff 0.01 was set as the significance cutoff for detecting outliers. For the first pass calculation of MD, outliers were detected with all bins. For the second pass calculation, outliers were removed for calculating X_c and C^{-1} . Then, MD distances were re-calculated for all bins based on first pass-outlier removed X_c and C^{-1} . Code for the analysis to generate MD distances and p-values, the plot, significant changed bins and associated genes are available at: <https://github.com/YaqiangCao/cLoops2/blob/master/scripts/compareComp.py>

“Similar concern to Figure 3A. The authors classified differential domains based on $\log_2(O)$. However, there is no threshold used to identify significantly differential domains in the MA plot. The authors may need to apply some cutoff such as FDR and \log_2FC . there should be a statistical significance cutoff. Also figure 3 legend title has an extra ‘changes’.”

This is also an important point raised by reviewers 2 and 3. Accordingly, we have revised our comparison of TADs between young and old samples with additional criteria (new Figure 3a). Like revised compartment comparisons, we implemented two-pass Mahalanobis distance calculation and Chi-square test to get the significant changed TADs. The Mahalanobis distance (MD) was calculated as $MD = \text{diag}((X - X_c) \times C^{-1} \times (X - X_c)^T)$, where X is the matrix for TAD enrichment scores (ES). Each row represents a TAD, and each column represents a sample (young or old). C^{-1} is the inverse covariance matrix of X , X_c is vector of row-wise mean of X , and *diag* is the function used to extract a diagonal array from a matrix. We performed the Chi-Square test with the MD and a P-value cutoff 0.01 was set as the significance cutoff for detecting outliers. For the first pass calculation of MD, outliers were detected with all bins. For the second pass calculation, outliers were removed for calculating X_c and C^{-1} . Thereafter, MD distances were calculated for all bins based on first pass-outlier removed X_c and C^{-1} . The TAD enrichment score (ES) was calculated by dividing the number of intra PETs (paired-end tags with both ends located in the TAD) by the number of inter PETs (paired-end tags with only one end located in the TAD). Using ES as the metric we found that the *IgH* TAD had the greatest change between young and old pro-B cells. Additionally, domains containing *Ebf1* and *Pax5* were amongst the top 5 (Extended Data Fig. 3a). The code for the analysis to generate MD distances and p-values for TADs, the plot, significant changed TADs and associated genes are available at: <https://github.com/YaqiangCao/cLoops2/blob/master/scripts/compareDom.py>.

Reviewer 3

“Several hundred loci transitioned from compartment A to B and visa versa. What statistics were applied to reach this conclusion, and how do the authors know that these are bona fide age-related differences rather than technical (or animal to animal) variation? The majority of the called compartment changes in 1C appear along the X=Y line, with only a handful of genomic loci seeming to separate. With statistics

and biological replicates taken into account, hundreds of called changes along X=Y line may prove to be insignificant, undercutting the author's claims of "large-scale conformational changes"

See first response under Reviewer 2 (Section B) for statistics applied towards identifying compartment-level transitions.

"Calling altered TADs is critical to the study. The authors describe one method in main text (lowered intraTAD interactions) but not the second method – but then use the intersection as their set of altered TADs. It is not clear to this reviewer how the two methods differ. As this intersection is key to the further focus on IgH (most reduced TAD). "

In the original submission we had considered that calling differential TADs by two methods and focusing only on those called by both pipelines to add rigor to the analysis. Reviewer 3 found our description of the two methods to be inadequate and recommended both be included in the main text. As described in the second response to Reviewer 2 (Section B), we have imposed greater rigor on differential TAD identification using two-stage statistical analyses. This computationally more stringent approach still reveals the *Igh* TAD as most changed in old pro-B cells, and TADs encompassing *Ebf1* and *Pax5* amongst the top 5 most changed. Thus, our new analyses do not change any of the biological implications explicated in the original submission. These results are summarized in a new Figure 3a (discussed on pages 6 and 7).

C 'All other referee concerns'

Reviewer 1

We thank the reviewer for their supportive comments. He/she found this to be an 'interesting paper' that provides 'new insight' and is 'particularly important as it relates to reduced efficiency of vaccination in the elderly'. The manuscript was considered to be 'well written' with 'overall compelling' data that was 'well presented'.

Specific comments:

'Could the authors please validate the change in nuclear location using 3D-FISH? What fraction of aged pro-B cells show a change in nuclear location? Are the EBF1 alleles repositioning to the nuclear lamina? Alternatively, they might relocate to the peri-centromeric chromatin in aged cells as described for plasma cells. Could the authors please check this out?'

This comment pertains to our observation that the *Ebf1* locus shifts from euchromatic compartment A to heterochromatic compartment B in old pro-B cells. The reviewer is interested in knowing whether compartment changes are accompanied by changes in nuclear location. To address this, we carried out

3D-FISH with *Ebf1* and *Foxo1* probes in young and old pro-B cells and non-B lineage bone marrow cells from RAG2-deficient mice. Cells for each experiment were pooled from 2-3 mice. In new Figure 1h and i, we show that the *Ebf1* locus is located closer to the nuclear periphery in old pro-B cells compared to young pro-B cells, with the average displacement lying between that seen in non-B cells (closest to the periphery) and young pro-B cells (furthest from the periphery) (discussed on page 5).

Additional comments:

“Since FOXO1 acts in a feed-forward loop with EBF1 it may very well be that this circuitry is altered in aged pro-B cells. From the data presented here it seems that indeed FOXO1 expression is substantially altered. Could the authors also check for nuclear repositioning of FOXO1 in young versus aged pro-B cells using the data that presented here?”

The reviewer is likely querying *Foxo1* compartmentalization since they refer to data already present in the original submission. We show in the revised manuscript that *Foxo1* is located squarely in compartment A in both young and old pro-B cells (Extended Data Fig. 1f, discussed on page 5). In addition, we provide new 3D-FISH data that shows that *Foxo1* nuclear localization also does not significantly change with age (Extended Data Fig. 1e, discussed on page 5). We conclude that *Foxo1* down-regulation in old pro-B cells occurs by mechanisms distinct from nuclear re-compartmentalization.

“One interesting question that comes to mind is whether forced EBF1 and/or FOXO1 expression in aged pro-B cells rescues an aged-specific gene program? I realize that it might not be straightforward to perform this experiment since EBF1 and FOXO1 dosage might be key.”

A related question was also raised by Reviewer 2. The reviewer correctly noted that rejuvenation of old pro-B cells by *Ebf1* (and/or *Foxo1*) over-expression. We spent most of the revision period addressing this gold standard. To obtain meaningful results, this involves manipulating the *Ebf1* level in old pro-B cells without changing their aged phenotype. We found that *ex vivo* culture of primary pro-B cells, that is essential for altering the *Ebf1* level, changed the transcriptomes of old pro-B cells (new Extended Data Fig. 2c, e). Though relatively few age-associated genes remained differentially expressed after culture of old pro-B cells, we found that *Ebf1* transduction changed the expression profile of these genes towards that of young pro-B cells (new Extended Data Fig. 2d, f). These observations support the conclusion that *Ebf1* down-regulation during aging contributes to differential gene expression in old pro-B cells. Additionally, our observations strongly reinforce the notion that physiological aging is an organismic property that is virtually impossible to study in tissue culture. Because of complications related to phenotypic changes caused by *ex vivo* culture, we did not carry out similar experiments with *Foxo1*.

“How is H3K27ac abundance altered in aged-versus young pro-B cells?”

To response this question, we carried out a flow-based measurement of H3K27ac and observed no difference in levels of H3K27ac in young and old pro-B cells (new Extended Data Fig. 5a). We also re-evaluated our H3K27ac ChIP-Seq to obtain a global view of changes that occur during aging. However, approximately 27% H3K27ac peaks were reduced in intensity in old pro-B cells; conversely, only 3% H3K27ac peaks were increased in old pro-B cells (new Figure 4a, b; discussed on page 8).

“Is E2A and p300 occupancy altered?”

We carried out anti-P300 ChIP-Seq using freshly isolated pro-B cells from *Rag2*-deficient mice (new Extended Data Fig.5b, discussed on page 8). The results revealed about 14,000 binding sites genome-wide in fresh *ex vivo* primary pro-B cells. Approximately 30% of sites lost p300 binding in old pro-B cells whereas only 1% sites were gained (new Extended Data Fig. 5b). Importantly, 74% sites that lost p300 were also reduced for H3K27ac (new Extended Data Fig. 5c). Due to difficulty obtaining large numbers of aged *Rag2*-deficient mice for isolation of pro-B cells, we had to select some experiments over others. Because of the known connection between H3K27ac and p300, we opted in favor of p300 ChIP-Seq.

“Likewise, since occupancy by chromatin remodelers, including BRG1, it would be nice to examine for BRG1 occupancy in young and old pro-B cells.”

We carried out anti-Brg1 ChIP-Seq in freshly isolated pro-B cells from young and old RAG2-deficient mice. We found very few of approximately 20,000 Brg1-bound sites changed with age (new Extended Data Fig. 5d, discussed on page 8). Our interpretation is that reduced Ebf1 levels may be compensated by other transcription factors with respect to Brg1 recruitment. However, Brg1 recruitment is clearly insufficient to direct young-like gene expression in old pro-B cells.

Reviewer 2

This reviewer found our studies were ‘rigorously carried out’, the data analysis to be ‘clearly described’ and, overall, that study would be of ‘broad interest to the biomedical research field’. We sincerely appreciate these comments. However, he/she also noted ‘considerable concerns’, which we have addressed as described below.

Main critiques

1. “Can over-expression of Ebf1 revert some of the aging-related chromatin reorganization features in old pro-B cells? This experiment would offer more compelling evidence supporting the role of reduced levels of Ebf1 in chromatin re-organization in old pro-B cells.”

The reviewer correctly noted that rejuvenation of old pro-B cells by Ebf1 over-expression (or knock-down of Wapl) would provide “more compelling evidence supporting the role (of these proteins) in chromatin re-organization in old pro-B cells”. We spent most of the revision period addressing this gold

standard. To obtain meaningful results, this involves manipulating Ebf1 (or Wapl) levels in old pro-B cells without changing their aged phenotype. We found that *ex vivo* culture of primary pro-B cells, that is essential for altering Ebf1 or Wapl levels, changed the transcriptomes of old pro-B cells (new Extended Data Fig. 2c, e). Though relatively few age-associated genes remained differentially expressed after culture of old pro-B cells, we found that Ebf1 transduction changed the expression profile of these genes towards that of young pro-B cells (new Extended Data Fig. 2d, f). These observations support the conclusion that Ebf1 down-regulation during aging contributes to differential gene expression in old pro-B cells. Additionally, our observations strongly reinforce the notion that physiological aging is an organismic property that is virtually impossible to study in tissue culture.

We limited the Ebf1 transduction studies to gene expression analyses rather than carry out a full chromatin study because results from Ebf1 heterozygote pro-B cells indicated that reduced Ebf1 levels are not at the apex of large-scale chromatin re-organization. Rather, we propose that chromatin re-organization leads to Ebf1 down-regulation and consequent changes in the pro-B cell transcriptome (discussed on pages 5, 6).

2. “Similarly, can knockdown of Wapl reverse age-associated chromatin re-organization?”

Wapl knockdown was not feasible in old primary pro-B cells. We attempted this via a) lentiviral transduction of shRNAs and b) siRNA transfection into old pro-B cells. As noted above, *ex vivo* culture of pro-B cells altered the aging phenotype sufficiently that we did not obtain interpretable results. Instead, we provide two lines of evidence in the revised manuscript that Wapl impacts pro-B cell chromatin structure. First, we draw attention to the published study in which Wapl expression was increased in pro-B cells by mutating a repressive element in the gene promoter. Hi-C analyses showed reduced interactions in the TAD range (300 kb - 10 Mb) and increased interactions in the compartment range (> 10 Mb). This pattern was similar to what we observed in old pro-B cells (new Figure 1a). We had previously quoted this paper¹ (ref #41 in both old and new versions). Second, we increased Wapl expression in a pro-B cell line (new Extended Data Fig. 4). Hi-C analyses of Wapl expressing cells showed lower genomic contacts in the 50-300 kb range, typical of intra-TAD interactions, and higher genomic contacts in > 10 Mb range, which are compartmental interactions, as seen in old pro-B cells (new Extended Data Fig. 4c). The *Igh* locus was amongst those that were most affected by Wapl expression (new Extended Data Fig 4d, e). Finally, RNA-Seq analyses showed that Wapl expression skewed the transcriptomic profile towards myeloid lineage cells, like what we had observed in old pro-B cells and *Ebf1*^{+/-} pro-B cells (new Extended Data Fig. 4f). Our observations demonstrate that altering Wapl levels recapitulates many chromatin and gene expression features of old pro-B cells. These results are discussed on page 7 of the revised manuscript.

In summary, we made good faith effort to alter Ebf1 and Wapl levels in primary old pro-B cells. We found that ex vivo culture altered the phenotype of old pro-B cells, making it impossible to do the perfect experiment. Instead, we manipulated Ebf1 and Wapl levels in pro-B cells and demonstrated that the changes induced are completely consistent with our hypothesis regarding the effects of age on B cell development.

3. “Authors point out a reduction in TAD strength in age at 1477 TADs (Figure 3A). They note that these TADs have lower Rad21 binding and hypothesize it is due to increased Wapl expression (Figure 3C,D). Authors should comment on the potential causes of increased Wapl expression in old pro-B cells and whether there is a global reduction of Rad21. If not, they could explore why some TADs are losing Rad21 while others are able to maintain it in aging”

We hypothesize that increased Wapl expression in old pro-B cells may be caused by reduced expression of the putative *Wapl* repressor, Pax5 (page 13). Indeed, we observed global reduction of Rad21 (new Figure 3g, discussed in page 7).

4. “In Figure 1C, the authors classified bins into A to B or B to A transitioning regions based on (0,0) coordinate. However, many regions called as switching compartments near the border could be artifacts. A more stringent cutoff should be determined. Much of consequential analysis is based on the A->B and B->A definitions in this figure. ”

Both reviewers 2 and 3 make this point, which is both relevant and important. Our analysis of the Hi-C data began in early 2020, at which time compartment analysis relied primarily on the sign of PCA analysis.³⁻⁵ Furthermore, comparisons between compartments were limited to flips/switches.^{6,7} This was also our strategy for Figure 1c (similar to, for example, Figures 1a and d in ⁷, and Figure S8 in ⁶). This kind of flip/switch analysis had also been used in more recent studies⁸ close to the time when we submitted our manuscript. However, we completely understand the need to stay state-of-the-art to publish in NCB and have therefore addressed reviewers’ concerns as follows. We implemented a two-pass Mahalanobis distance calculation (an effective measurement for detecting outliers based on the distribution pattern of data points), as well as a Chi-square test, which is the same strategy used in a recent dcHi-C analysis⁸. As expected, application of these criteria reduced the number of significant compartment flips/switches compared to our previous analysis; in addition, we also detected some changes within the same compartments (new Figure 1c). Importantly, loci that we had focused on as undergoing compartment switching in the original submission, such as Ebf1, remained highly statistically significant even after applying the criteria noted above.

The Mahalanobis distance (MD) was calculated as $MD = \text{diag}((X-X_c) \times C^{-1} \times (X-X_c)^T)$, where X is the matrix for PC1 obtained from the Hi-C compartment analysis. Each row represents a bin, and each column represents a sample (young or old). C^{-1} is the inverse covariance matrix of X, X_c is vector of row-wise mean of X, and diag is the function used to extract a diagonal array from a matrix. We performed the Chi-Square test with the MD and a P-value cutoff 0.01 was set as the significance cutoff

for detecting outliers. For the first pass calculation of MD, outliers were detected with all bins. For the second pass calculation, outliers were removed for calculating X_c and C^{-1} . Then, MD distances were re-calculated for all bins based on first pass-outlier removed X_c and C^{-1} . Code for the analysis to generate MD distances and p-values, the plot, significant changed bins and associated genes are available at: <https://github.com/YaqiangCao/cLoops2/blob/master/scripts/compareComp.py>

5. “Similar concern to Figure 3A. The authors classified differential domains based on $\log_2(0)$. However, there is no threshold used to identify significantly differential domains in the MA plot. The authors may need to apply some cutoff such as FDR and \log_2FC . there should be a statistical significance cutoff.”

This is also an important point raised by reviewers 2 and 3. Accordingly, we have revised our comparison of TADs between young and old samples with additional criteria (new Figure 3a). Like revised compartment comparisons, we implemented two-pass Mahalanobis distance calculation and Chi-square test to get the significant changed TADs. The Mahalanobis distance (MD) was calculated as $MD = \text{diag}((X - X_c) \times C^{-1} \times (X - X_c)^T)$, where X is the matrix for TAD enrichment scores (ES). Each row represents a TAD, and each column represents a sample (young or old). C^{-1} is the inverse covariance matrix of X , X_c is vector of row-wise mean of X , and diag is the function used to extract a diagonal array from a matrix. We performed the Chi-Square test with the MD and a P-value cutoff 0.01 was set as the significance cutoff for detecting outliers. For the first pass calculation of MD, outliers were detected with all bins. For the second pass calculation, outliers were removed for calculating X_c and C^{-1} . Thereafter, MD distances were calculated for all bins based on first pass-outlier removed X_c and C^{-1} . The TAD enrichment score (ES) was calculated by dividing the number of intra PETs (paired-end tags with both ends located in the TAD) by the number of inter PETs (paired-end tags with only one end located in the TAD). Using ES as the metric we found that the IgH TAD had the greatest change between young and old pro-B cells. Additionally, domains containing Ebf1 and Pax5 were amongst the top 5 (Extended Data Fig. 3a). The code for the analysis to generate MD distances and p-values for TADs, the plot, significant changed TADs and associated genes are available at: <https://github.com/YaqiangCao/cLoops2/blob/master/scripts/compareDom.py>.

Additional concerns

6. “For some people who do not know the function of Rag2, I suggest including a sentence to explain how Rag2 deficiency affects VDJ recombination in this strain, and especially, its association with key B-cell development transcription factor such as Ebf1”

We have included a brief description of the phenotype of Rag2-deficient pro-B cells (page 3). To the best of our knowledge, there is no evidence of functional interactions between Rag2 and Ebf1.

7. “The authors used Rag2-deficient mice. RAG1 and RAG2 together initiate recombination and IgH gene assembly by introducing double-strand DNA breaks in developing lymphocytes. The authors did not explain the possible impact of RAG1 in this study.”

In the revised version, we noted that Rag1- or Rag2-deficiency has the same effect on B cell development, that of halting development at the pro-B cell stage (page 3).

8. “In line 44-52, the authors mentioned that very little is known about chromatin re-organization during aging. However, there is some age-related 3D chromatin reorganization study done in, for example, muscle stem cell aging. I suggest that the authors either discuss openly the similarities and differences of their study with previous work, or modify their statement limited to immune cells or B cell development.”

We note that the muscle stem cell aging papers mentioned by the reviewer had not been published at the time when our manuscript was submitted. The results in these papers are not directly comparable to our studies. However, we have quoted these papers in the revised manuscript (page 11) and modified our statement to emphasize our use of immune cells for this aging study (page 2).

9. “In Figure 1B, there are some regions with increased short-range contacts. In Figure 1B, there are some regions with increased short-range contacts (red color). Can the authors elaborate more on what those regions are?”

We believe the reviewer was referring to the dashed line square-highlighted region in the left panel of Figure R1 (only for reviewers, at the end of this letter). Upon closer examination, we have zoomed in on this region, and it appears to be a 7.5 Mb-sized TAD showing smaller increased interactions amount compared to nearby region long-distance distance interactions, as shown in the right panel of the following figure. We consider this observation to be within the normal range since some TADs exhibit increased interactions, as demonstrated in the revised new Figure 3a.

10. “In Figure 1E, the enriched GO terms for genes within compartments switching from A to B in aging seem to be the same as those enriched in immune cell/B cell gene expression generally, which could be inherent to those genes within compartment A of pro-B cells. It may be necessary to perform this enrichment analysis with a background set of only genes in compartment A of the young pro-B cells rather than all genes to avoid any bias in sampling.”

We agree that interpretation of GO terms is complicated. We revised the compartment analysis with more stringent cutoffs as suggested by the reviewer (point 4), but we did not find any enriched GO terms adding much mechanistic information, so we have deleted this figure from the revised manuscript.

11. “The enriched GO terms for regions switching from compartment B to compartment A in aging (Fig S1.D) are far more significant (despite having fewer regions total). It would be great if authors could comment on them as well.”

We revised the compartment analysis with more stringent cutoffs as suggested by the reviewer (point 4), but we did not find any enriched terms revealing mechanistic information. Therefore, we have removed this part of the result in the revision. All compartment changes are provided in Source Data for Figure 1, permitting the interested reader to carry out such analysis.

12. “In line 213-215, the author proposed the shift from B cell lineage to myeloid and erythroid differentiation. However, the authors did not measure the proportion of cell populations or did not provide direct evidence related to this in this study. I suggest referring previous studies that showed shift from the lymphoid to the myeloid lineage of in aged hematopoietic stem cells to support their statement.”

We thank the reviewer for this suggestion. We have referred to a paper about shift from the lymphoid to the myeloid lineage in aged hematopoietic stem cells in the revised manuscript (page 12).

13. “Scale bar is not defined in figures or not explained in captions for Figure 1B, 2B, 2D, 3G, 4A, and 4C.”

We thank the reviewer for noticing this. All these Hi-C heatmaps were visualized by Juicebox⁹, and the scale bars represent the normalized interaction strength as determined by Juicebox. This information was included in the corresponding figure legends.

14. “Any statistical test for Figure S4?”

We appreciate the reviewer’s keen observation regarding Figure S4 (new Extended Data Fig. 6) and apologize for the omission of a statistical test. We have incorporated the statistical test in the revised version of our work.

15. “Authors should further discuss potential causes for the compartment changes.”

This is a very interesting question. We concluded from our studies in *Ebf1*^{+/-} pro-B cells that compartment changes were upstream of changes in *Ebf1* expression. That is, change in *Ebf1* expression was the consequence of compartment changes in old pro-B cells, not *vice versa* (page 5). We also hypothesize that the global increase in compartment-level interactions in the Hi-C assay may, in part, result from diminished TAD strength in old pro-B cells which, in turn, is due to reduced enhancer-promoter interactions in old pro-B cells (page 8).

Reviewer 3

While the reviewer acknowledged the significance of the inquiry into chromatin organization during aging as an "outstanding question" and found our study to be "convincing," there were notable concerns raised in the original submission. The reviewer also highlighted "two major problems (and other issues)." We have addressed these comments as described below.

Major problems

“First, the focus on Ebf1 is not well justified in the study.”

We have included additional justification for the focus on Ebf1 (pages 5, 6). There are three reasons for our continued emphasis on this transcription factor. First, the *Ebf1* locus is the top (out of 157) region that transitions from A to B compartment in old pro-B cells, with associated reduction in mRNA expression. Second, we were prompted to experimentally test the consequences of reduced *Ebf1* expression because it is a key regulator of B cell development. In its absence, B cell development halts prior to commitment of multipotential precursor cells to the B lineage. Third, *Ebf1* was previously shown to undergo radial re-localization in pro-B cells nuclei, a change that could well be related to different compartments.

“Second, the driving changes may be loss (mostly) and gain of enhancers and enhancer-promoter interactions, and this is not explicitly tested in the datasets. The major conclusion of “large scale age-associated chromatin re-organization at a critical juncture of B cell development” may be an emergent computational property of smaller looping changes between enhancer-promoters due to enhancer loss. The authors should fully evaluate enhancers and enhancer-promoter interactions”

The reviewer correctly noted that we had not thoroughly investigated enhancer-promoter interactions in the previous version of the manuscript. In the revised manuscript we provide such analyses (new Figure 4). We carried out H3K27ac ChIP-Seq as a means of identifying active enhancers and H3K27ac Hi-ChIP to evaluate interactions between enhancers and promoters. We found that approximately 27% of H3K27ac-marked sites were reduced in old pro-B cells (new Figure 4a); most of these were located at enhancers. By contrast, only 3% H3K27ac-marked sites were increased in old pro-B cells, again mostly at enhancers (new Figure 4a). Genes annotated to H3K27ac peaks that reduced with age were expressed at significantly lower levels in old pro-B cells and vice versa (new Figure 4b-d). Hi-ChIP analyses showed that enhancers and promoters with reduced H3K27ac were involved in weaker interactions in old pro-B cells (new Figure 4e-h). These observations are exactly in accordance with the predictions made by the reviewer (discussed on pages 8, 9).

Specific points

1. (64-69) “Technical questions with study: How homogenous is the pro-B cells population (referred to as “relatively” homogenous”). If not homogeneous, changes observed could be due to shifts in cell population. Also, how many samples were used and were individual samples mixed? If mixed, changes could be due to individual mouse variation.”

The benefit to using Rag2-deficient mice for these studies is that B cell development is completely blocked at the pro-B cell stage. Thus, all chromatin, transcription and ChIP assays were carried with developmentally homogenous pro-B cells. We had previously used the phrase ‘relatively homogenous’ to reflect the fact that even in a developmentally homogenous population some gene expression heterogeneity can be detected at the single cell level (our unpublished data). To clarify, without hampering scientific accuracy, we have removed the word ‘relatively’ in the revised manuscript (page 3). We pooled purified pro-B cells from 4-6 mice for each biological replicate experiment (Figure 1 legend) to reduce the individual variation.

2. 73-75 “Several hundred loci transitioned from compartment A to B and visa versa. What statistics were applied to reach this conclusion, and how do the authors know that these are bona fide age-related differences rather than technical (or animal to animal) variation? The majority of the called compartment changes in 1C appear along the X=Y line, with only a handful of genomic loci seeming to separate. With statistics and biological replicates taken into account, hundreds of called changes along X=Y line may prove to be insignificant, undercutting the author’s claims of “large-scale conformational changes”

Both reviewers 2 and 3 make this point, which is both relevant and important. Our analysis of the Hi-C data began in early 2020, at which time compartment analysis relied primarily on the sign of PCA analysis.³⁻⁵ Furthermore, comparisons between compartments were limited to flips/switches.^{6,7} This was also our strategy for Figure 1c (similar to, for example, Figures 1a and d in ⁷, and Figure S8 in ⁶). This kind of flip/switch analysis had also been used in more recent studies⁸ close to the time when we submitted our manuscript. However, we completely understand the need to stay state-of-the-art to publish in NCB and have therefore addressed reviewers’ concerns as follows. We implemented a two-pass Mahalanobis distance calculation (an effective measurement for detecting outliers based on the distribution pattern of data points), as well as a Chi-square test, which is the same strategy used in a recent dcHi-C analysis⁸. As expected, application of these criteria reduced the number of significant compartment flips/switches compared to our previous analysis; in addition, we also detected some changes within the same compartments (new Figure 1c). Importantly, loci that we had focused on as undergoing compartment switching in the original submission, such as Ebf1, remained highly statistically significant even after applying the criteria noted above.

The Mahalanobis distance (MD) was calculated as $MD = \text{diag}((X - X_c) \times C^{-1} \times (X - X_c)^T)$, where X is the matrix for PC1 obtained from the Hi-C compartment analysis. Each row represents a bin, and each column represents a sample (young or old). C^{-1} is the inverse covariance matrix of X, X_c is vector of row-wise mean of X, and diag is the function used to extract a diagonal array from a matrix. We performed the Chi-Square test with the MD and a P-value cutoff 0.01 was set as the significance cutoff for detecting outliers. For the first pass calculation of MD, outliers were detected with all bins. For the second pass calculation, outliers were removed for calculating X_c and C^{-1} . Then, MD distances were re-calculated for all bins based on first pass-outlier removed X_c and C^{-1} . Code for the analysis to

generate MD distances and p-values, the plot, significant changed bins and associated genes are available at: <https://github.com/YaquiCao/cLoops2/blob/master/scripts/compareComp.py>

3. (84-94) “Most changes are compartment A (euchromatin) to B (heterochromatin). Fig 1 and 2 very quickly focus on one gene, Ebf1, that switches from A to B. How was this gene chosen? What is the molecular function of Ebf1? What are the overall statistics of changes (amplitude of compartment changes, RNA seq, ChIP seq, etc) and where does this gene Ebf1 rank in these changes? What are the gene categories of changes and is this a key gene for B cells in comparison to other genes. It is not clear why CTCF would be increased rather than decreased during gene repression and no increase in K27me3”

The reviewer queried our focus on Ebf1 as well as requested quantitative details of changes at the *Ebf1* locus and its expression in old pro-B cells. As we discussed in the response to first major problem raised by Reviewer #3, we have included additional justification for focusing on Ebf1 (pages 4, 5). Briefly, the *Ebf1* locus is the top region (out of 157) that transitioned from A to B compartment in old pro-B cells. Not only is *Ebf1* the most strongly affected loci, but the gene also caught our attention because it is a well-known regulator of B cell development. Additionally, *Ebf1* has been previously shown to undergo nuclear re-localization in pro-B cells, a change that could well be reflected in altered compartmentalization. Overall statistics of changes in *Ebf1* are summarized in the Source Data.

4. “Then further deep analysis of Ebf1 is shown in Fig 2 (95-115) and again in Fig 4, looking at chromatin and gene expression in Ebf1+/- cells. How were these cells derived? From individual humans? How are repeats carried out? Why would similar chromatin and gene expression changes be expected in Ebf1+/- compared to old B cells? The downstream focus on Ebf1 regulation of specific genes Mmrn1 and Minar1/Temed3 again seems very specific (frankly, rather cherry-picked). Is Ebf1 a transcription factor – does it ChIP to Mmrn1 or to Minar1/Temed3? (And later, to IgH?) Where do these genes rank in genes regulated by Ebf1? The conclusion (113-115) seems very strong about Ebf1’s role without demonstration of robust genome-wide involvement.”

The rationale for detailed analysis of *Ebf1*^{+/-} pro-B cells (which are derived from a mouse fetal liver, with two biological replicates) was the following. Our attention had been drawn to Ebf1 for reasons summarized in the first response to Reviewer 3. However, despite the known importance of Ebf1 for B cell development, our observations did not reveal the extent to which changes noted in old pro-B cells were caused by alterations in Ebf1 expression. We hypothesized that alterations in Ebf1 were likely to be responsible for only a subset of changes seen in old pro-B cells. One way to test this was to examine chromatin state and gene expression in pro-B cells in which Ebf1 levels were lowered by a means other than age. We chose to do this using pro-B cells that lacked one of two *Ebf1* alleles. These cells had been previously shown to express half the level of Ebf1 compared to pro-B cells with two *Ebf1* alleles. We reasoned that *Ebf1*^{+/-} pro-B cells represented pro-B cells with reduced Ebf1 expression but none of the other (Ebf1-independent) characteristics of old pro-B cells.

As explained more extensively in the revised manuscript (pages 5, 6), we found that Ebf1 down-regulation did not recapitulate increased compartment level interactions found in old pro-B cells. We interpret this to mean that Ebf1 does not induce compartment changes at the global level as observed in old pro-B cells. However, some loci changed comparably in *Ebf1*^{+/-} and old pro-B cells. One such locus is *Minar1/Tmed3* (Figure 2d). This locus has been previously shown to bind Ebf1 in pro-B cells². We infer that changes at this locus may be the result of reduced Ebf1 expression in old pro-B cells and *Ebf1*^{+/-} cells. Remarkably, Ebf1 heterozygosity led to changes in gene expression that had many features of old pro-B cells, especially genes relevant for B cell development (Figure 2e, f). We conclude that compartmental changes in old pro-B cells reduce Ebf1 expression that, in turn, alter gene expression patterns in keeping with its role as a transcription factor. Identification of the initiator of compartment level alterations during aging remains a goal for the future.

5. (128) “It is not clear (at least to this reviewer) that old and *Ebf1*^{+/-} can be directly compared to deduce the contribution of Ebf1 to old phenotype. What if heterozygous loss of Ebf1 in young cells leads to other changes due to developmental timing? For example, *Wapl* could be changed in PTMs. This conclusion should be carefully stated and discussed.”

We appreciate the point made by the reviewer. In the revised version we have elaborated our reasons for choosing this experimental approach, while cautioning the reader about the possible developmental caveat (pages 5, 6).

6. (120-121) “Calling altered TADs is critical to the study. The authors describe one method in main text (lowered intraTAD interactions) but not the second method – but then use the intersection as their set of altered TADs. It is not clear to this reviewer how the two methods differ. As this intersection is key to the further focus on IgH (most reduced TAD), the authors should put both methods into main text.”

In the original submission we had considered that calling differential TADs by two methods and focusing only on those called by both pipelines to add rigor to the analysis. Reviewer 3 found our description of the two methods to be inadequate and recommended both be included in the main text. In the revised version, we have imposed greater rigor on differential TAD identification using two-stage statistical analyses. we implemented two-pass Mahalanobis distance calculation and Chi-square test to get the significant changed TADs. The Mahalanobis distance (MD) was calculated as $MD = \text{diag}((X - X_c) \times C^{-1} \times (X - X_c)^T)$, where X is the matrix for TAD enrichment scores (ES). Each row represents a TAD, and each column represents a sample (young or old). C^{-1} is the inverse covariance matrix of X , X_c is vector of row-wise mean of X , and diag is the function used to extract a diagonal array from a matrix. We performed the Chi-Square test with the MD and a P-value cutoff 0.01 was set as the significance cutoff for detecting outliers. For the first pass calculation of MD, outliers were detected with all bins. For the second pass calculation, outliers were removed for calculating X_c and C^{-1} . Thereafter,

MD distances were calculated for all bins based on first pass-outlier removed X_c and C^{-1} . The TAD enrichment score (ES) was calculated by dividing the number of intra PETs (paired-end tags with both ends located in the TAD) by the number of inter PETs (paired-end tags with only one end located in the TAD). Using ES as the metric we found that the *IgH* TAD had the greatest change between young and old pro-B cells. Additionally, domains containing *Ebf1* and *Pax5* were amongst the top 5 (Extended Data Fig. 3a). The code for the analysis to generate MD distances and p-values for TADs, the plot, significant changed TADs and associated genes are available at:

<https://github.com/YaqiangCao/cLoops2/blob/master/scripts/compareDom.py>

This computationally more stringent approach still reveals the *IgH* TAD as most changed in old pro-B cells, and TADs encompassing *Ebf1* and *Pax5* amongst the top 5 most changed. Thus, our new analyses do not change any of the biological implications explicated in the original submission. These results are summarized in a new Figure 3a (discussed on pages 6 and 7).

7. “Fig 3B and 3C. The authors should show heatmaps (ie using Deeptools) under the metaplots or another indication demonstrating how many changing TADs have differential histone modifications, CTCF, or Rad21. The signal differences observed in these metaplots could be due to small differences across many TADs or large differences in a few TADs, which informs how these results are interpreted.”

We have now revised the analysis of significantly different TADs and updated the aggregation analysis for ChIP-seq signals on TADs. The heatmaps are presented in Figure R2 (available only to reviewers, at the end of this letter), illustrating that the signal differences were attributed to small variations across multiple TADs.

8. (152-153) “It appears that most of the documented changes that are occurring locally to IgH and Ebf1 loci are loss of K27ac. The authors interpret this rather broadly as “reduced interactions locus-wide and widespread loss of H3K27ac and Rad21 mark the IgH TAD in old pro-B cells”. This is, to this reviewer, _____ The authors should evaluate enhancers genome-wide and enhancer-promoter interactions relative to gene expression to determine if enhancer alterations explain the loss of cell identity.”

We applaud the reviewer’s perceptiveness about the possible role of enhancer-promoter interactions in driving interaction changes in old pro-B cells. In the revised manuscript, we provide new experimental data and analysis (new Figure 4) investigating the role of enhancers and enhancer-promoter interactions in young and old pro-B cells. By combining H3K27ac ChIP-Seq (to score for differences between young and old pro-B cells) and H3K27ac Hi-ChIP (to score for interactions between H3K27ac-marked enhancers and promoters) we found that loss of enhancer-promoter interactions greatly exceeded gain of such interactions in old pro-B cells, exactly in accordance with the predictions made by the reviewer (discussed on pages 8, 9).

9. “Figure 4. Can the authors explain the “virtual 4D map”... what is this and its interpretation?”

We believe that the reviewer is referring to "virtual 4C tracks." A virtual 4C track serves as a method to assess the interactions of a specific genomic region with others. In this approach, we identify paired-end tags (PETs) involved in interactions, where at least one end is located within the bait (the target-specific region). Subsequently, we generate 1D signals depicting the density of these interactions, and this graphical representation is termed "virtual 4C." The methodology mirrors that of 4C (Circularized Chromosome Conformation Capture) and is elucidated in the revised manuscript (pages 10 and 36).

In summary, we have addressed all Reviewer queries with new experiments, new analysis and conceptual clarifications as described above. We appreciate the time and efforts of the Reviewers and Editors in improving the rigor and quality of our manuscript. We hope that with these substantive changes, you will find the manuscript acceptable for publication in *Nature Cell Biology*.

Sincerely,

Ranjan Sen, Ph.D.

Chief, Laboratory of Molecular Biology and Immunology

References

1. Hill, L. *et al.* Wapl repression by Pax5 promotes V gene recombination by Igh loop extrusion. *Nature* **584**, 142-147 (2020).
2. Wang, Y. *et al.* A Prion-like Domain in Transcription Factor EBF1 Promotes Phase Separation and Enables B Cell Programming of Progenitor Chromatin. *Immunity* **53**, 1151-1167 e1156 (2020).
3. Lieberman-Aiden, E. *et al.* Comprehensive mapping of long-range interactions reveals folding principles of the human genome. *Science* **326**, 289-293 (2009).
4. Rao, S.S. *et al.* A 3D map of the human genome at kilobase resolution reveals principles of chromatin looping. *Cell* **159**, 1665-1680 (2014).
5. Zheng, X. & Zheng, Y. CscoreTool: fast Hi-C compartment analysis at high resolution. *Bioinformatics* **34**, 1568-1570 (2018).
6. Bonev, B. *et al.* Multiscale 3D Genome Rewiring during Mouse Neural Development. *Cell* **171**, 557-572 e524 (2017).
7. Dixon, J.R. *et al.* Chromatin architecture reorganization during stem cell differentiation. *Nature* **518**, 331-336 (2015).
8. Chakraborty, A., Wang, J.G. & Ay, F. dcHiC detects differential compartments across multiple Hi-C datasets. *Nat Commun* **13**, 6827 (2022).
9. Durand, N.C. *et al.* Juicebox Provides a Visualization System for Hi-C Contact Maps with Unlimited Zoom. *Cell Syst* **3**, 99-101 (2016).

Figure R1

Figure R1. Response to point 9 from Reviewer #2. The difference Hi-C contact frequency heatmap of an 80Mb region (left) of mouse chromosome 11. Regions with increased and decreased interactions in old are colored red and blue, respectively. The highlighted region within the black dashed line box is magnified in the right panel.

Figure R2

Figure R2. Response to point 7 from Reviewer #3. All significantly changed TADs are shown as heatmaps generated with deepTools. Color bars indicate normalized ChIP-Seq signals densities shown in the heatmaps.

Decision Letter, first revision:

Our ref: NCB-LE49860A

16th February 2024

Dear Dr. Sen,

Thank you for submitting your revised manuscript "Age-associated chromatin re-organization in progenitor B cells" (NCB-LE49860A). It has now been seen by two of the original referees and their comments are below. Please kindly note that we have been unable to receive a report from Reviewer 3 and thus asked Reviewer 2 to cross-comment on your response to Reviewer 3, which you may also find below. Together, the reviewers find that the paper has improved in revision, and therefore we'll be happy in principle to publish it in Nature Cell Biology, pending minor revisions to satisfy the referees' final requests and to comply with our editorial and formatting guidelines.

Thank you again for your interest in Nature Cell Biology Please do not hesitate to contact me if you have any questions.

Sincerely,

Zhe Wang, PhD
Senior Editor
Nature Cell Biology

Tel: +44 (0) 207 843 4924
email: zhe.wang@nature.com

Reviewer #1 (Remarks to the Author):

The authors have address the majority of my comments. In my view this is an interesting piece of work with new insights into the important problem of mechanisms that underpin the aging process of immune cells.

Reviewer #2 (Remarks to the Author):

Overall, the authors have effectively addressed my initial concerns. They addressed the lack of statistical tests to determine the changing compartments and domains across age by applying Mahalanobis distance + Chi-square based tests to determine statistically significant changes. (points 4+5). The second concern I had was about the GO analysis, specifically that the input gene list did not undergo a statistical test (as mentioned in 1) and the analysis lacked an appropriate background set of genes. They responded to this by removing the GO analysis section completely as the mechanistic changes they reported were no longer significant after running the GO analysis only on genes within statistically significantly changing TADs. (point 10). They added statistics (scale bars, etc) to figures that were missing them. These changes drastically improved the manuscript.

Below are two minor comments -

1. (comment 1) In recognizing the inherent experimental constraints associated with ex vivo cell culture, it is acknowledged that this method may not entirely replicate the intricacies of the aging phenotype. Despite these limitations, the authors have astutely discerned changes in the transcription profile of old-B cells following the expression of Ebf1. In the new Extended Data Fig. 2f, the 68 genes and 108 genes in Venn diagram include both up and down regulated genes. The potential for upregulated genes to be downregulated introduces ambiguity. To improve result interpretation, I suggest the authors consider segregating upregulated and downregulated genes in the Venn diagram.

2. Typo on the Y-axis label in Extended Data Fig. 2.: "densitiies"

*** FURTHER COMMENTS WHEN ASKED TO CROSS-COMMENT ON YOUR REBUTTAL TO REVIEWER 3:

I have gone through the revised manuscript once again and the responses to reviewer #3 comments. I think that the authors have adequately addressed the reviewer #3's comments. I do not have much to add.

Author Rebuttal, first revision:

Reviewer #1 (Remarks to the Author):

The authors have address the majority of my comments. In my view this is an interesting piece of work with new insights into the important problem of mechanisms that underpin the aging process of

immune cells.

We really appreciate your thorough review of our manuscript and are very pleased to hear that you found our work interesting!

Reviewer #2 (Remarks to the Author):

Overall, the authors have effectively addressed my initial concerns. They addressed the lack of statistical tests to determine the changing compartments and domains across age by applying Mahalanobis distance + Chi-square based tests to determine statistically significant changes. (points 4+5). The second concern I had was about the GO analysis, specifically that the input gene list did not undergo a statistical test (as mentioned in 1) and the analysis lacked an appropriate background set of genes. They responded to this by removing the GO analysis section completely as the mechanistic changes they reported were no longer significant after running the GO analysis only on genes within statistically significantly changing TADs. (point 10). They added statistics (scale bars, etc) to figures that were missing them. These changes drastically improved the manuscript.

Thank you for your thoughtful and constructive feedback on our manuscript. We are pleased to hear that most of your concerns have been effectively addressed.

Below are two minor comments -

1. (comment 1) In recognizing the inherent experimental constraints associated with ex vivo cell culture, it is acknowledged that this method may not entirely replicate the intricacies of the aging phenotype. Despite these limitations, the authors have astutely discerned changes in the transcription profile of old-B cells following the expression of Ebf1. In the new Extended Data Fig. 2f, the 68 genes and 108 genes in Venn diagram include both up and down regulated genes. The potential for upregulated genes to be downregulated introduces ambiguity. To improve result interpretation, I suggest the authors consider segregating upregulated and downregulated genes in the Venn diagram.

It is a good suggestion and we have modified the figure accordingly to improve result interpretation, providing a clearer representation of the changes in gene expression.

2. Typo on the Y-axis label in Extended Data Fig. 2.: "densitiies"

Sorry for the typo and we have corrected it.

*** FURTHER COMMENTS WHEN ASKED TO CROSS-COMMENT ON YOUR REBUTTAL TO REVIEWER 3:

I have gone through the revised manuscript once again and the responses to reviewer #3 comments. I think that the authors have adequately addressed the reviewer #3's comments. I do not have much to add.

We appreciate the time and effort you have dedicated to reviewing our work. Thank you again for your valuable feedback.

Final Decision Letter:

Dear Dr Sen,

I am pleased to inform you that your manuscript, "Three-dimensional chromatin re-organization regulates B cell development during aging", has now been accepted for publication in Nature Cell Biology.

Please note that *Nature Cell Biology* is a Transformative Journal (TJ). Authors may publish their research with us through the traditional subscription access route or make their paper immediately open access through payment of an article-processing charge (APC). Authors will not be required to make a final decision about access to their article until it has been accepted. Find out more about Transformative Journals

If you have not already done so, we strongly recommend that you upload the step-by-step protocols used in this manuscript to the Protocol Exchange (www.nature.com/protocolexchange), an open online resource established by Nature Protocols that allows researchers to share their detailed experimental know-how. All uploaded protocols are made freely available, assigned DOIs for ease of citation and are fully searchable through nature.com. Protocols and Nature Portfolio journal papers in which they are used can be linked to one another, and this link is clearly and prominently visible in the online versions of both papers. Authors who performed the specific experiments can act as primary authors for the Protocol as they will be best placed to share the methodology details, but the Corresponding Author of the present research paper should be included as one of the authors. By uploading your Protocols to Protocol Exchange, you are enabling researchers to more readily reproduce or adapt the methodology you use, as well as increasing the visibility of your protocols and papers. You can also establish a dedicated page to collect your lab Protocols. Further information can be found at www.nature.com/protocolexchange/about

With kind regards,

Zhe Wang, PhD
Senior Editor
Nature Cell Biology

Tel: +44 (0) 207 843 4924
email: zhe.wang@nature.com
